# Spectra of the Conjugate Kernel and Neural Tangent Kernel for Linear-Width Neural Networks

**Zhou Fan**
Department of Statistics and Data Science
Yale University
zhou.fan@yale.edu

**Zhichao Wang**
Department of Mathematics
University of California, San Diego
zhw036@ucsd.edu

## Abstract

We study the eigenvalue distributions of the Conjugate Kernel and Neural Tangent Kernel associated to multi-layer feedforward neural networks. In an asymptotic regime where network width is increasing linearly in sample size, under random initialization of the weights, and for input samples satisfying a notion of approximate pairwise orthogonality, we show that the eigenvalue distributions of the CK and NTK converge to deterministic limits. The limit for the CK is described by iterating the Marcenko-Pastur map across the hidden layers. The limit for the NTK is equivalent to that of a linear combination of the CK matrices across layers, and may be described by recursive fixed-point equations that extend this Marcenko-Pastur map. We demonstrate the agreement of these asymptotic predictions with the observed spectra for both synthetic and CIFAR-10 training data, and we perform a small simulation to investigate the evolutions of these spectra over training.

## 1 Introduction

Recent progress in our theoretical understanding of neural networks has connected their training and generalization to two associated kernel matrices. The first is the *Conjugate Kernel (CK)* or the equivalent Gaussian process kernel [43, 56, 12, 14, 48, 52, 33, 41]. This is the gram matrix of the derived features produced by the final hidden layer of the network. The network predictions are linear in these derived features, and the CK governs training and generalization in this linear model.

The second is the *Neural Tangent Kernel (NTK)* [27, 19, 6]. This is the gram matrix of the Jacobian of in-sample predictions with respect to the network weights, and was introduced to study full network training. Under gradient-flow training dynamics, the in-sample predictions follow a differential equation governed by the NTK. We provide a brief review of these matrices in Section 2.1.

The spectral decompositions of these kernel matrices are related to training and generalization properties of the underlying network. Training occurs most rapidly along the eigenvectors of the largest eigenvalues [5], and the eigenvalue distribution may determine the trainability of the model and the extent of implicit bias towards simpler functions [57, 59]. It is thus of interest to understand the spectral properties of these matrices, both at random initialization and over the course of training.

### 1.1 Summary of contributions

In this work, we apply techniques of random matrix theory to derive an exact asymptotic characterization of the eigenvalue distributions of the CK and NTK at random initialization, in a multi-layer feedforward network architecture. We study a "linear-width" asymptotic regime, where each hidden layer has width proportional to the training sample size. We impose an assumption of approximate pairwise orthogonality for the training samples, which encompasses general settings of independent samples that need not have independent entries.

We show that the eigenvalue distributions for both the CK and the NTK converge to deterministic limits, depending on the limiting eigenvalue distribution of the training data. The limit distribution for the CK at each intermediate hidden layer is a Marcenko-Pastur map of a linear transformation of that of the previous layer. The NTK can be approximated by a linear combination of CK matrices, and its limiting eigenvalue distribution can be described by a recursively defined sequence of fixed-point equations that extend this Marcenko-Pastur map. We demonstrate the agreement of these asymptotic limits with the observed spectra on both synthetic and CIFAR-10 training data of moderate size.

In this linear-width asymptotic regime, feature learning occurs, and both the CK and NTK evolve over training. Although our theory pertains only to their spectra at random initialization of the weights, we conclude with an empirical examination of their spectral evolutions during training, on simple examples of learning a single neuron and learning a binary classifier for two classes in CIFAR-10. In these examples, the bulk eigenvalue distributions of the CK and NTK undergo elongations, and isolated principal components emerge that are highly predictive of the training labels. Recent theoretical work has studied the evolution of the NTK in an entrywise sense [25, 20], and we believe it is an interesting open question to translate this understanding to a more spectral perspective.

## 1.2   Related literature

Many properties of the CK and NTK have been established in the limit of infinite width and fixed sample size $n$. In this limit, both the CK [43, 56, 14, 33, 41] and the NTK [27, 34, 58] at random initialization converge to fixed $n \times n$ kernel matrices. The associated random features regression models converge to kernel linear regression in the RKHS of these limit kernels. Furthermore, network training occurs in a "lazy" regime [11], where the NTK remains constant throughout training [27, 19, 18, 6, 34, 7]. Spectral properties of the CK, NTK, and Hessian of the training loss have been previously studied in this infinite-width limit in [48, 51, 57, 30, 21, 28]. Limitations of lazy training and these equivalent kernel regression models have been studied theoretically and empirically in [11, 7, 60, 22, 23, 35], suggesting that trained neural networks of practical width are not fully described by this type of infinite-width kernel equivalence. The asymptotic behavior is different in the linear-width regime of this work: For example, for a linear activation $\sigma(x) = x$, the infinite-width limit of the CK for random weights is the input Gram matrix $X^\top X$, whereas its limit spectrum under linear-width asymptotics has an additional noise component from iterating the Marcenko-Pastur map.

Under linear-width asymptotics, the limit CK spectrum for one hidden layer was characterized in [46] for training data with i.i.d. Gaussian entries. For activations satisfying $\mathbb{E}_{\xi \sim \mathcal{N}(0,1)}[\sigma'(\xi)] = 0$, [46] conjectured that this limit is a Marcenko-Pastur law also in multi-layer networks, and this was proven under a subgaussian assumption as part of the results of [9]. [39] studied the one-hidden-layer CK with general training data, and [37] specialized this to Gaussian mixture models. These works [39, 37] showed that the limit spectrum is a Marcenko-Pastur map of the inter-neuron covariance. We build on this insight by analyzing this covariance across multiple layers, under approximate orthogonality of the training samples. This orthogonality condition is similar to that of [3], which recently studied the one-hidden-layer CK with a bias term. This condition is also more general than the assumption of i.i.d. entries, and we describe in Appendix I the reduction to the one-hidden-layer result of [46] for i.i.d. Gaussian inputs, as this reduction is not immediately clear. [44] provides another form of the limit distribution in [46], which is equivalent to our form in Appendix I via the relation described in [8].

The limit NTK spectrum for a one-hidden-layer network with i.i.d. Gaussian inputs was recently characterized in parallel work of [4]. In particular, [4] applied the same idea as in Lemma 3.5 below to study the Hadamard product arising in the NTK. [45, 47] previously studied the equivalent spectrum of a sample covariance matrix derived from the network Jacobian, which is one of two components of the Hessian of the training loss, in a slightly different setting and also for one hidden layer.

The spectra of the kernel matrices $X^\top X$ that we study are equivalent (up to the addition/removal of 0's) to the spectra of the sample covariance matrices in linear regression using the features $X$. As developed in a line of recent literature including [16, 46, 17, 39, 36, 24, 42, 4, 15], this spectrum and the associated Stieltjes transform and resolvent are closely related to the training and generalization errors in this linear regression model. These works have collectively provided an asymptotic understanding of training and generalization error for random features regression models derived from the CK and NTK of one-hidden-layer neural networks, and related qualitative phenomena of double and multiple descent in the generalization error curves.

## 2 Background

### 2.1 Neural network model and kernel matrices

We consider a fully-connected, feedforward neural network with input dimension $d_0$, hidden layers of dimensions $d_1, \ldots, d_L$, and a scalar output. For an input $\mathbf{x} \in \mathbb{R}^{d_0}$, we parametrize the network as

$$f_\theta(\mathbf{x}) = \mathbf{w}^\top \frac{1}{\sqrt{d_L}} \sigma \left( W_L \frac{1}{\sqrt{d_{L-1}}} \sigma \left( \ldots \frac{1}{\sqrt{d_2}} \sigma \left( W_2 \frac{1}{\sqrt{d_1}} \sigma(W_1 \mathbf{x}) \right) \right) \right) \in \mathbb{R}. \tag{1}$$

Here, $\sigma : \mathbb{R} \to \mathbb{R}$ is the activation function (applied entrywise) and

$$W_\ell \in \mathbb{R}^{d_\ell \times d_{\ell-1}} \quad \text{for } 1 \le \ell \le L, \qquad \mathbf{w} \in \mathbb{R}^{d_L}$$

are the network weights. We denote by $\theta = (\text{vec}(W_1), \ldots, \text{vec}(W_L), \mathbf{w})$ the weights across all layers. The scalings by $1/\sqrt{d_\ell}$ reflect the "NTK-parametrization" of the network [27]. We discuss alternative scalings and an extension to multi-dimensional outputs in Section 3.4.

Given $n$ training samples $\mathbf{x}_1, \ldots, \mathbf{x}_n \in \mathbb{R}^{d_0}$, we denote the matrices of inputs and post-activations by

$$X \equiv X_0 = (\mathbf{x}_1 \quad \ldots \quad \mathbf{x}_n) \in \mathbb{R}^{d_0 \times n}, \quad X_\ell = \frac{1}{\sqrt{d_\ell}} \sigma\left(W_\ell X_{\ell-1}\right) \in \mathbb{R}^{d_\ell \times n} \quad \text{for } 1 \le \ell \le L.$$

Then the in-sample predictions of the network are given by $f_\theta(X) = (f_\theta(\mathbf{x}_1), \ldots, f_\theta(\mathbf{x}_n)) = \mathbf{w}^\top X_L \in \mathbb{R}^{1 \times n}$. The **Conjugate Kernel (CK)** is the matrix

$$K^{\text{CK}} = X_L^\top X_L \in \mathbb{R}^{n \times n}.$$

More generally, we will call $X_\ell^\top X_\ell$ the conjugate kernel at the intermediate layer $\ell$. Fixing the matrix $X_L$, the CK governs training and generalization in the linear regression model $\mathbf{y} = \mathbf{w}^\top X_L$. For very wide networks, $K^{\text{CK}}$ may be viewed as an approximation of its infinite-width limit,[1] and regression using $X_L$ is an approximation of regression in the RKHS defined by this limit kernel [49].

We denote the Jacobian matrix of the network predictions with respect to the weights $\theta$ as

$$J = \nabla_\theta f_\theta(X) = (\nabla_\theta f(\mathbf{x}_1) \quad \cdots \quad \nabla_\theta f(\mathbf{x}_n)) \in \mathbb{R}^{\dim(\theta) \times n}.$$

The **Neural Tangent Kernel (NTK)** is the matrix

$$K^{\text{NTK}} = J^\top J = \left(\nabla_\theta f_\theta(X)\right)^\top \left(\nabla_\theta f_\theta(X)\right) \in \mathbb{R}^{n \times n}. \tag{2}$$

Under gradient-flow training of the network weights $\theta$ with training loss $\|\mathbf{y} - f_\theta(X)\|^2/2$, the time evolutions of residual errors and in-sample predictions are given by

$$\frac{d}{dt}\left(\mathbf{y} - f_{\theta(t)}(X)\right) = -K^{\text{NTK}}(t) \cdot \left(\mathbf{y} - f_{\theta(t)}(X)\right), \quad \frac{d}{dt} f_{\theta(t)}(X) = K^{\text{NTK}}(t) \cdot \left(\mathbf{y} - f_{\theta(t)}(X)\right) \tag{3}$$

where $\theta(t)$ and $K^{\text{NTK}}(t)$ are the parameters and NTK at training time $t$ [27, 19]. Denoting the eigenvalues and eigenvectors of $K^{\text{NTK}}(t)$ by $(\lambda_\alpha(t), \mathbf{v}_\alpha(t))_{\alpha=1}^n$, and the spectral components of the residual error by $r_\alpha(t) = \mathbf{v}_\alpha(t)^\top (\mathbf{y} - f_{\theta(t)}(X))$, these training dynamics are expressed spectrally as

$$\mathbf{v}_\alpha(t)^\top \frac{d}{dt}\left(\mathbf{y} - f_{\theta(t)}(X)\right) = -\lambda_\alpha(t) r_\alpha(t), \qquad \frac{d}{dt} f_{\theta(t)}(X) = \sum_{\alpha=1}^n \lambda_\alpha(t) r_\alpha(t) \cdot \mathbf{v}_\alpha(t).$$

Note that these relations hold instantaneously at each training time $t$, regardless of whether $K^{\text{NTK}}(t)$ evolves or remains approximately constant over training. Hence, $\lambda_\alpha(t)$ controls the instantaneous rate of decay of the residual error in the direction of $\mathbf{v}_\alpha(t)$.

For very wide networks, $K^{\text{NTK}}$, $\lambda_\alpha$, and $\mathbf{v}_\alpha$ are all approximately constant over the entirety of training [27, 19, 18, 6, 11]. This yields the closed-form solution $r_\alpha(t) \approx r_\alpha(0) e^{-t\lambda_\alpha}$, so that the in-sample predictions $f_{\theta(t)}(X)$ converge exponentially fast to the observed training labels $\mathbf{y}$, with a different exponential rate $\lambda_\alpha$ along each eigenvector $\mathbf{v}_\alpha$ of $K^{\text{NTK}}$.

## 2.2 Eigenvalue distributions, Stieltjes transforms, and the Marcenko-Pastur map

We will derive almost-sure weak limits for the empirical eigenvalue distributions of random symmetric kernel matrices $K \in \mathbb{R}^{n \times n}$ as $n \to \infty$. Throughout this paper, we will denote this as

$$\lim \operatorname{spec} K = \mu$$

where $\mu$ is the limit probability distribution on $\mathbb{R}$. Letting $\{\lambda_\alpha\}_{\alpha=1}^n$ be the eigenvalues of $K$, this means

$$\frac{1}{n} \sum_{\alpha=1}^n f(\lambda_\alpha) \to \mathbb{E}_{\lambda \sim \mu}[f(\lambda)] \tag{4}$$

a.s. as $n \to \infty$, for any continuous bounded function $f : \mathbb{R} \to \mathbb{R}$. Intuitively, this may be understood as the convergence of the "bulk" of the eigenvalue distribution of $K$.[2] We will also show that $\|K\| \leq C$ a.s., for a constant $C > 0$ and all large $n$. Then (4) in fact holds for any continuous function $f : \mathbb{R} \to \mathbb{R}$, as such a function must be bounded on $[-C, C]$.

We will characterize the probability distribution $\mu$ and the empirical eigenvalue distribution of $K$ by their Stieltjes transforms. These are defined, respectively, for a spectral argument $z \in \mathbb{C}^+$ as[3]

$$m_\mu(z) = \int \frac{1}{x - z} d\mu(x), \qquad m_K(z) = \frac{1}{n} \sum_{\alpha=1}^n \frac{1}{\lambda_\alpha - z} = \frac{1}{n} \operatorname{Tr}(K - z \operatorname{Id})^{-1}.$$

The pointwise convergence $m_K(z) \to m_\mu(z)$ a.s. over $z \in \mathbb{C}^+$ implies $\lim \operatorname{spec} K = \mu$. For $z = x + i\eta \in \mathbb{C}^+$, the value $\pi^{-1} \operatorname{Im} m_\mu(z)$ is the density function of the convolution of $\mu$ with the distribution $\operatorname{Cauchy}(0, \eta)$ at $x \in \mathbb{R}$. Hence, the function $m_\mu(z)$ uniquely defines $\mu$, and evaluating $\pi^{-1} \operatorname{Im} m_\mu(x + i\eta)$ for small $\eta > 0$ yields an approximation for the density function of $\mu$ (provided this density exists at $x$).

An example of this type of characterization is given by the *Marcenko-Pastur map*, which describes the spectra of sample covariance matrices [40]: Let $X \in \mathbb{R}^{d \times n}$ have i.i.d. $\mathcal{N}(0, 1/d)$ entries, let $\Phi \in \mathbb{R}^{n \times n}$ be deterministic and positive semi-definite, and let $n \to \infty$ such that $\lim \operatorname{spec} \Phi = \mu$ and $n/d \to \gamma \in (0, \infty)$. Then the sample covariance matrix $\Phi^{1/2} X^\top X \Phi^{1/2}$ has an almost sure spectral limit,

$$\lim \operatorname{spec} \Phi^{1/2} X^\top X \Phi^{1/2} = \rho_\gamma^{\mathrm{MP}} \boxtimes \mu. \tag{5}$$

We will call this limit $\rho_\gamma^{\mathrm{MP}} \boxtimes \mu$ the Marcenko-Pastur map of $\mu$ with aspect ratio $\gamma$. This distribution $\rho_\gamma^{\mathrm{MP}} \boxtimes \mu$ may be defined by its Stieltjes transform $m(z)$, which solves the Marcenko-Pastur fixed point equation [40]

$$m(z) = \int \frac{1}{x(1 - \gamma - \gamma z m(z)) - z} d\mu(x). \tag{6}$$

## 3 Main results

### 3.1 Assumptions

We use Greek indices $\alpha$, $\beta$, etc. for samples in $\{1, \ldots, n\}$, and Roman indices $i$, $j$, etc. for neurons in $\{1, \ldots, d\}$. For a matrix $X \in \mathbb{R}^{d \times n}$, we denote by $\mathbf{x}_\alpha$ its $\alpha^{\text{th}}$ column and by $\mathbf{x}_i^\top$ its $i^{\text{th}}$ row. $\|\cdot\|$ is the $\ell_2$-norm for vectors and $\ell_2 \to \ell_2$ operator norm for matrices. Id is the identity matrix.

**Definition 3.1.** Let $\varepsilon, B > 0$. A matrix $X \in \mathbb{R}^{d \times n}$ is $(\varepsilon, B)$-**orthonormal** if its columns satisfy, for every $\alpha \neq \beta \in \{1, \ldots, n\}$,

$$\left| \|\mathbf{x}_\alpha\|^2 - 1 \right| \leq \varepsilon, \qquad \left| \mathbf{x}_\alpha^\top \mathbf{x}_\beta \right| \leq \varepsilon, \qquad \|X\| \leq B, \qquad \sum_{\alpha=1}^n (\|\mathbf{x}_\alpha\|^2 - 1)^2 \leq B^2.$$

**Assumption 3.2.** The number of layers $L \geq 1$ is fixed, and $n, d_0, d_1, \ldots, d_L \to \infty$, such that

(a) The weights $\theta = (\mathrm{vec}(W_1), \ldots, \mathrm{vec}(W_L), \mathbf{w})$ are i.i.d. and distributed as $\mathcal{N}(0,1)$.

(b) The activation $\sigma(x)$ is twice differentiable, with $\sup_{x \in \mathbb{R}} |\sigma'(x)|, |\sigma''(x)| \leq \lambda_\sigma$ for some $\lambda_\sigma < \infty$. For $\xi \sim \mathcal{N}(0,1)$, we have $\mathbb{E}[\sigma(\xi)] = 0$ and $\mathbb{E}[\sigma^2(\xi)] = 1$.

(c) The input $X \in \mathbb{R}^{d_0 \times n}$ is $(\varepsilon_n, B)$-orthonormal in the sense of Definition 3.1, where $B$ is a constant, and $\varepsilon_n n^{1/4} \to 0$ as $n \to \infty$.

(d) As $n \to \infty$, $\lim \mathrm{spec}\, X^\top X = \mu_0$ for a probability distribution $\mu_0$ on $[0, \infty)$, and $\lim n/d_\ell = \gamma_\ell$ for constants $\gamma_\ell \in (0, \infty)$ and each $\ell = 1, 2, \ldots, L$.

Part (c) quantifies our assumption of approximate pairwise orthogonality of the training samples. Although not completely general, it encompasses many settings of independent samples with input dimension $d_0 \asymp n$, including:

- Non-white Gaussian inputs $\mathbf{x}_\alpha \sim \mathcal{N}(0, \Sigma)$, for any $\Sigma$ satisfying $\mathrm{Tr}\, \Sigma = 1$ and $\|\Sigma\| \lesssim 1/n$.
- Inputs $\mathbf{x}_\alpha$ drawn from certain multi-class Gaussian mixture models, in the high-dimensional asymptotic regimes that were studied in [13, 39, 37, 36, 38].
- Inputs that may be expressed as $\sqrt{d_0} \cdot \mathbf{x}_\alpha = f(\mathbf{z}_\alpha)$, where $\mathbf{z}_\alpha \in \mathbb{R}^m$ has independent entries satisfying a log-Sobolev inequality, and $f : \mathbb{R}^m \to \mathbb{R}^{d_0}$ is any Lipschitz function.

In particular, the limit spectral law $\mu_0$ in Assumption 3.2(d) can be very different from the Marcenko-Pastur spectrum that would correspond to $X$ having i.i.d. entries. This approximate orthogonality is implied by the following more technical convex concentration property, which is discussed further in [55, 1]. We prove this result in Appendix B.

**Proposition 3.3.** *Let* $X = (\mathbf{x}_1, \ldots, \mathbf{x}_n) \in \mathbb{R}^{d_0 \times n}$, *where* $\mathbf{x}_1, \ldots, \mathbf{x}_n$ *are independent training samples satisfying* $\mathbb{E}[\mathbf{x}_\alpha] = 0$ *and* $\mathbb{E}[\|\mathbf{x}_\alpha\|^2] = 1$. *Suppose, for some constant* $c_0 > 0$, *that* $d_0 \geq c_0 n$, *and each vector* $\sqrt{d_0} \cdot \mathbf{x}_\alpha$ *satifies the convex concentration property*

$$\mathbb{P}\left[ \left| \varphi(\sqrt{d_0} \cdot \mathbf{x}_\alpha) - \mathbb{E}\varphi(\sqrt{d_0} \cdot \mathbf{x}_\alpha) \right| \geq t \right] \leq 2e^{-c_0 t^2}$$

*for every* $t > 0$ *and every 1-Lipschitz convex function* $\varphi : \mathbb{R}^{d_0} \to \mathbb{R}$. *Then for any* $k > 0$, *with probability* $1 - n^{-k}$, $X$ *is* $\left( \sqrt{\frac{C \log n}{d_0}}, B \right)$-*orthonormal for some* $C, B > 0$ *depending only on* $c_0, k$.

In Assumptions 3.2(a) and (b), the scaling of $\theta$ and the conditions $\mathbb{E}[\sigma(\xi)] = 0$ and $\mathbb{E}[\sigma^2(\xi)] = 1$, together with the parametrization (1), ensure that all pre-activations have approximate mean 0 and variance 1. This may be achieved in practice by batch normalization [26]. For $\xi \sim \mathcal{N}(0,1)$, we define the following constants associated to $\sigma(x)$. We verify in Proposition C.1 that under Assumption 3.2(b), we have $b_\sigma^2 \leq 1 \leq a_\sigma$.

$$b_\sigma = \mathbb{E}[\sigma'(\xi)], \quad a_\sigma = \mathbb{E}[\sigma'(\xi)^2], \quad q_\ell = (b_\sigma^2)^{L-\ell}, \quad r_\ell = a_\sigma^{L-\ell}, \quad r_+ = \sum_{\ell=0}^{L-1} r_\ell - q_\ell. \quad (7)$$

## 3.2  Spectrum of the Conjugate Kernel

Recall the Marcenko-Pastur map (5). Let $\mu_1, \mu_2, \mu_3, \ldots$ be the sequence of probability distributions on $[0, \infty)$ defined recursively by

$$\mu_\ell = \rho_{\gamma_\ell}^{\mathrm{MP}} \boxtimes \left( (1 - b_\sigma^2) + b_\sigma^2 \cdot \mu_{\ell-1} \right). \quad (8)$$

Here, $\mu_0$ is the input limit spectrum in Assumption 3.2(d), $b_\sigma$ is defined in (7), and $(1 - b_\sigma^2) + b_\sigma^2 \cdot \mu$ denotes the translation and rescaling of $\mu$ that is the distribution of $(1 - b_\sigma^2) + b_\sigma^2 \lambda$ when $\lambda \sim \mu$.

The following theorem shows that these distributions $\mu_1, \mu_2, \mu_3, \ldots$ are the asymptotic limits of the empirical eigenvalue distributions of the CK across the layers. Thus, the limit distribution for each layer $\ell$ is a Marcenko-Pastur map of a translation and rescaling of that of the preceding layer $\ell - 1$.

**Theorem 3.4.** *Suppose Assumption 3.2 holds, and define* $\mu_1, \ldots, \mu_L$ *by (8). Then (marginally) for each* $\ell = 1, \ldots, L$, *we have* $\lim \mathrm{spec}\, X_\ell^\top X_\ell = \mu_\ell$. *In particular,*

$$\lim \mathrm{spec}\, K^{CK} = \mu_L.$$

*Furthermore,* $\|K^{CK}\| \leq C$ *a.s. for a constant* $C > 0$ *and all large* $n$.

If $\sigma(x)$ is such that $b_\sigma = 0$, then each distribution $\mu_\ell$ is simply the Marcenko-Pastur law $\rho_{\gamma_\ell}^{\text{MP}}$. This special case was previously conjectured in [46] and proven in [9], for input data $X$ with i.i.d. entries. Note that for such non-linearities, the limiting CK spectrum does not depend on the spectrum $\mu_0$ of the input data, and furthermore $\mu_1 = \ldots = \mu_L$ if the layers have the same width $d_1 = \ldots = d_L$. Implications of this for the network discrimination ability in classification tasks and for learning performance have been discussed previously in [13, 46, 39, 38, 3].

To connect Theorem 3.4 to our next result on the NTK, let us describe the iteration (8) more explicitly using a recursive sequence of fixed-point equations derived from the Marcenko-Pastur equation (6): Let $m_\ell(z)$ be the Stieltjes transform of $\mu_\ell$, and define

$$\tilde{t}_\ell(z_{-1}, z_\ell) = \lim_{n \to \infty} \frac{1}{n} \operatorname{Tr}(z_{-1} \operatorname{Id} + z_\ell X_\ell^\top X_\ell)^{-1} = \frac{1}{z_\ell} m_\ell \left( -\frac{z_{-1}}{z_\ell} \right).$$

Applying the Marcenko-Pastur equation (6) to $m_\ell(-z_{-1}/z_\ell)$, and introducing $\tilde{s}_\ell(z_{-1}, z_\ell) = [z_\ell(1 - \gamma_\ell + \gamma_\ell z_{-1} \tilde{t}_\ell(z_{-1}, z_\ell))]^{-1}$, one may check that (8) may be written as the pair of equations

$$\tilde{t}_\ell(z_{-1}, z_\ell) = \tilde{t}_{\ell-1}\left( z_{-1} + \frac{1 - b_\sigma^2}{\tilde{s}_\ell(z_{-1}, z_\ell)}, \frac{b_\sigma^2}{\tilde{s}_\ell(z_{-1}, z_\ell)} \right), \tag{9}$$

$$\tilde{s}_\ell(z_{-1}, z_\ell) = (1/z_\ell) + \gamma_\ell \left( \tilde{s}_\ell(z_{-1}, z_\ell) - z_{-1} \tilde{s}_\ell(z_{-1}, z_\ell) \tilde{t}_\ell(z_{-1}, z_\ell) \right), \tag{10}$$

where (10) is a rearrangement of the definition of $\tilde{s}_\ell$. Applying (9) to substitute $\tilde{t}_\ell(z_{-1}, z_\ell)$ in (10), the equation (10) is a fixed-point equation that defines $\tilde{s}_\ell$ in terms of $\tilde{t}_{\ell-1}$. Then (9) defines $\tilde{t}_\ell$ in terms of $\tilde{s}_\ell$ and $\tilde{t}_{\ell-1}$. The limit Stieltjes transform for $K^{\text{CK}}$ is the specialization $m_{\text{CK}}(z) = \tilde{t}_L(-z, 1)$.

### 3.3 Spectrum of the Neural Tangent Kernel

In the neural network model (1), an application of the chain rule yields an explicit form

$$K^{\text{NTK}} = X_L^\top X_L + \sum_{\ell=1}^{L} (S_\ell^\top S_\ell) \odot (X_{\ell-1}^\top X_{\ell-1})$$

for certain matrices $S_\ell \in \mathbb{R}^{d_\ell \times n}$, where $\odot$ is the Hadamard (entrywise) product. We refer to Appendix G.1 for the exact expression; see also [25, Eq. (1.7)]. Our spectral analysis of $K^{\text{NTK}}$ relies on the following approximation, which shows that the limit spectrum of $K^{\text{NTK}}$ is equivalent to a linear combination of the CK matrices $X_0^\top X_0, \ldots, X_L^\top X_L$ and Id. We prove this in Appendix G.1.

**Lemma 3.5.** *Under Assumption 3.2, letting $r_+$ and $q_\ell$ be as defined in (7),*

$$\lim \operatorname{spec} K^{NTK} = \lim \operatorname{spec} \left( r_+ \operatorname{Id} + X_L^\top X_L + \sum_{\ell=0}^{L-1} q_\ell X_\ell^\top X_\ell \right).$$

By this lemma, if $b_\sigma = 0$, then $q_0 = \ldots = q_{L-1} = 0$ and the limit spectrum of $K^{\text{NTK}}$ reduces to the limit spectrum of $r_+ \operatorname{Id} + X_L^\top X_L$ which is a translation of $\rho_{\gamma_L}^{\text{MP}}$ described in Theorem 3.4. Thus we assume in the following that $b_\sigma \neq 0$. Our next result provides an analytic description of the limit spectrum of $K^{\text{NTK}}$, by extending (9,10) to characterize the trace of rational functions of $X_0^\top X_0, \ldots, X_L^\top X_L$ and Id.

Denote the closed lower-half complex plane with 0 removed as $\mathbb{C}^* = \overline{\mathbb{C}^-} \setminus \{0\}$. For $\ell = 0, 1, 2, \ldots$, we define recursively two sequences of functions

$$t_\ell : (\mathbb{C}^- \times \mathbb{R}^\ell \times \mathbb{C}^*) \times \mathbb{C}^{\ell+2} \to \mathbb{C}, \qquad (\mathbf{z}, \mathbf{w}) \mapsto t_\ell(\mathbf{z}, \mathbf{w})$$

$$s_\ell : \mathbb{C}^- \times \mathbb{R}^\ell \times \mathbb{C}^* \to \mathbb{C}^+, \qquad \mathbf{z} \mapsto s_\ell(\mathbf{z}).$$

where $\mathbf{z} = (z_{-1}, z_0, \ldots, z_\ell) \in \mathbb{C}^- \times \mathbb{R}^\ell \times \mathbb{C}^*$ and $\mathbf{w} = (w_{-1}, w_0, \ldots, w_\ell) \in \mathbb{C}^{\ell+2}$. We will define these functions such that $t_\ell(\mathbf{z}, \mathbf{w})$ will be the value of

$$\lim_{n \to \infty} n^{-1} \operatorname{Tr}(z_{-1} \operatorname{Id} + z_0 X_0^\top X_0 + \ldots + z_\ell X_\ell^\top X_\ell)^{-1}(w_{-1} \operatorname{Id} + w_0 X_0^\top X_0 + \ldots + w_\ell X_\ell^\top X_\ell).$$

For $\ell = 0$, we define the first function $t_0$ by

$$t_0\left( (z_{-1}, z_0), (w_{-1}, w_0) \right) = \int \frac{w_{-1} + w_0 x}{z_{-1} + z_0 x} d\mu_0(x) \tag{11}$$

For $\ell \geq 1$, we then define the functions $s_\ell$ and $t_\ell$ recursively by

$$s_\ell(\mathbf{z}) = (1/z_\ell) + \gamma_\ell t_{\ell-1}\big(\mathbf{z}_{\text{prev}}(s_\ell(\mathbf{z}), \mathbf{z}),\, (1 - b_\sigma^2, 0, \ldots, 0, b_\sigma^2)\big), \tag{12}$$

$$t_\ell(\mathbf{z}, \mathbf{w}) = (w_\ell/z_\ell) + t_{\ell-1}\big(\mathbf{z}_{\text{prev}}(s_\ell(\mathbf{z}), \mathbf{z}),\, \mathbf{w}_{\text{prev}}\big) \tag{13}$$

where we write as shorthand

$$\mathbf{z}_{\text{prev}}(s_\ell(\mathbf{z}), \mathbf{z}) \equiv \left(z_{-1} + \frac{1 - b_\sigma^2}{s_\ell(\mathbf{z})}, z_0, \ldots, z_{\ell-2}, z_{\ell-1} + \frac{b_\sigma^2}{s_\ell(\mathbf{z})}\right) \in \mathbb{C}^- \times \mathbb{R}^{\ell-1} \times \mathbb{C}^*, \tag{14}$$

$$\mathbf{w}_{\text{prev}} \equiv (w_{-1}, \ldots, w_{\ell-1}) - (w_\ell/z_\ell) \cdot (z_{-1}, \ldots, z_{\ell-1}) \in \mathbb{C}^{\ell+1}. \tag{15}$$

**Proposition 3.6.** *Suppose $b_\sigma \neq 0$. For each $\ell \geq 1$ and any $\mathbf{z} \in \mathbb{C}^- \times \mathbb{R}^\ell \times \mathbb{C}^*$, there is a unique solution $s_\ell(\mathbf{z}) \in \mathbb{C}^+$ to the fixed-point equation (12).*

Hence, (12) defines the function $s_\ell$ in terms of the function $t_{\ell-1}$, and this is then used in (13) to define $t_\ell$. This is illustrated diagrammatically as

$$
\begin{array}{ccccccc}
t_0 & \to & t_1 & \to & t_2 & \to & \cdots \\
\downarrow & \nearrow & \downarrow & \nearrow & \downarrow & \nearrow & \\
s_1 & & s_2 & & s_3 & &
\end{array}
$$

Specializing the function $t_L$ for the last layer $L$ to the values $(z_{-1}, z_0, \ldots, z_{L-1}, z_L) = (r_+, q_0, \ldots, q_{L-1}, 1)$ and $(w_{-1}, w_0, \ldots, w_L) = (1, 0, \ldots, 0)$, we obtain an analytic description for the limit spectrum of $K^{\text{NTK}}$ via its Stieltjes transform.

**Theorem 3.7.** *Suppose $b_\sigma \neq 0$. Under Assumption 3.2, for any fixed values $z_{-1}, z_0, \ldots, z_L \in \mathbb{R}$ where $z_L \neq 0$, we have $\lim \operatorname{spec}(z_{-1} \operatorname{Id} + z_0 X_0^\top X_0 + \ldots + z_L X_L^\top X_L) = \nu$ where $\nu$ is the probability distribution with Stieltjes transform $m_\nu(z) = t_L((-z + z_{-1}, z_0, \ldots, z_L), (1, 0, \ldots, 0))$.*

*In particular, $\lim \operatorname{spec} K^{\text{NTK}}$ is the probability distribution with Stieltjes transform*

$$m_{\text{NTK}}(z) = t_L\Big((-z + r_+, q_0, \ldots, q_{L-1}, 1), (1, 0, \ldots, 0)\Big).$$

*Furthermore, $\|K^{\text{NTK}}\| \leq C$ a.s. for a constant $C > 0$ and all large $n$.*

We remark that Theorem 3.7 encompasses the previous result in Theorem 3.4 for $K^{\text{CK}} = X_L^\top X_L$, by specializing to $(z_0, \ldots, z_{L-1}, z_L) = (0, \ldots, 0, 1)$. Under this specialization, $s_\ell(z_{-1}, 0, \ldots, 0, z_\ell) = \tilde{s}_\ell(z_{-1}, z_\ell)$, $t_\ell((z_{-1}, 0, \ldots, 0, z_\ell), (1, 0, \ldots, 0)) = \tilde{t}_\ell(z_{-1}, z_\ell)$, and (12,13) reduce to (9,10).

### 3.4 Extension to multi-dimensional outputs and rescaled parametrizations

Theorem 3.7 pertains to a network with scalar outputs, under the "NTK-parametrization" of network weights in (1). As neural network models used in practice often have multi-dimensional outputs and may be parametrized differently for backpropagation, we state here the extension of the preceding result to a network with $k$-dimensional output and a general scaling of the weights.

Consider the model

$$f_\theta(\mathbf{x}) = W_{L+1}^\top \frac{1}{\sqrt{d_L}} \sigma\left(W_L \frac{1}{\sqrt{d_{L-1}}} \sigma\left(\ldots \frac{1}{\sqrt{d_2}} \sigma\left(W_2 \frac{1}{\sqrt{d_1}} \sigma(W_1 \mathbf{x})\right)\right)\right) \in \mathbb{R}^k \tag{16}$$

where $W_{L+1}^\top \in \mathbb{R}^{k \times d_L}$. We write the coordinates of $f_\theta$ as $(f_\theta^1, \ldots, f_\theta^k)$, and the vectorized output for all training samples $X \in \mathbb{R}^{d_0 \times n}$ as $f_\theta(X) = (f_\theta^1(X), \ldots, f_\theta^k(X)) \in \mathbb{R}^{nk}$. We consider the NTK

$$K^{\text{NTK}} = \sum_{\ell=1}^{L+1} \tau_\ell \Big(\nabla_{W_\ell} f_\theta(X)\Big)^\top \Big(\nabla_{W_\ell} f_\theta(X)\Big) \in \mathbb{R}^{nk \times nk}. \tag{17}$$

For $\tau_1 = \ldots = \tau_{L+1} = 1$, this is a flattening of the NTK defined in [27], and we recall briefly its derivation from gradient-flow training in Appendix H.1. We consider general constants $\tau_1, \ldots, \tau_{L+1} > 0$ to allow for a different learning rate for each weight matrix $W_\ell$, which may arise from backpropagation in the model (16) using a parametrization with different scalings of the weights.

**Theorem 3.8.** *Fix any $k \geq 1$. Suppose Assumption 3.2 holds, and $b_\sigma \neq 0$. Then $\|K^{\text{NTK}}\| \leq C$ a.s. for a constant $C > 0$ and all large $n$, and $\lim \operatorname{spec} K^{\text{NTK}}$ is the probability distribution with Stieltjes transform*

$$m_{\text{NTK}}(z) = t_L\Big((-z + \tau \cdot r_+,\ \tau_1 q_0, \ldots, \tau_L q_{L-1}, \tau_{L+1}), (1, 0, \ldots, 0)\Big), \quad \tau \cdot r_+ \equiv \sum_{\ell=0}^{L-1} \tau_{\ell+1}(r_\ell - q_\ell).$$

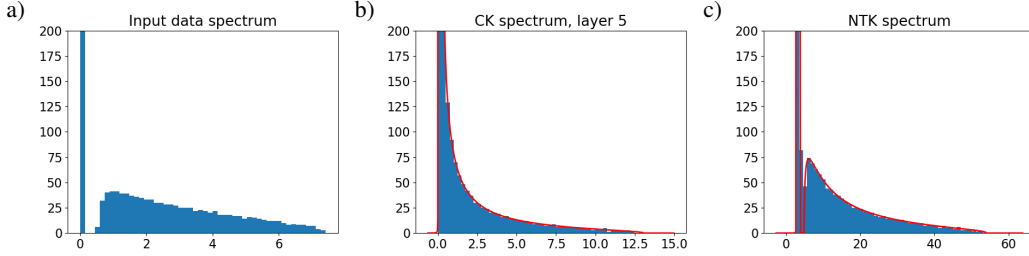

Figure 1: Simulated spectra at initialization for i.i.d. Gaussian training samples in a 5-layer network, for (a) the input gram matrix $X_0^\top X_0$, (b) $K^{\mathrm{CK}} = X_5^\top X_5$, and (c) $K^{\mathrm{NTK}}$. Numerical computations of the limit spectra in Theorems 3.4 and 3.7 are superimposed in red.

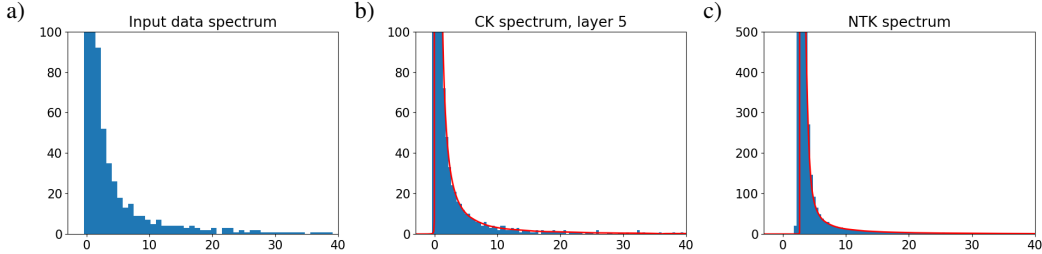

Figure 2: Same plots as Figure 1, for 5000 training samples from CIFAR-10 with 10 leading PCs removed.

## 4 Experiments

We describe in Appendix A an algorithm to numerically compute the limit spectral densities of Theorem 3.7. The computational cost is independent of the dimensions $(n, d_0, \ldots, d_L)$, and each limit density below was computed within a few seconds on our laptop computer. Using this procedure, we investigate the accuracy of the theoretical predictions of Theorems 3.4 and 3.7. Finally, we conclude by examining the spectra of $K^{\mathrm{CK}}$ and $K^{\mathrm{NTK}}$ after network training.

### 4.1 Simulated Gaussian training data

We consider $n = 3000$ training samples with i.i.d. $\mathcal{N}(0, 1/d_0)$ entries, input dimension $d_0 = 1000$, and $L = 5$ hidden layers of dimensions $d_1 = \ldots = d_5 = 6000$. We take $\sigma(x) \propto \tan^{-1}(x)$, normalized so that $\mathbb{E}[\sigma(\xi)^2] = 1$. A close agreement between the observed and limit spectra is displayed in Figure 1, for both $K^{\mathrm{CK}}$ and $K^{\mathrm{NTK}}$ at initialization. The CK spectra for intermediate layers are depicted in Appendix J.4.

We highlight two qualitative phenomena: The spectral distribution of the NTK (at initialization) is separated from 0, as explained by the $\mathrm{Id}$ component in Lemma 3.5. Across layers $\ell = 1, \ldots, L$, there is a merging of the spectral bulk components of the CK, and an extension of its spectral support.

### 4.2 CIFAR-10 training data

We consider $n = 5000$ samples randomly selected from the CIFAR-10 training set [32], with input dimension $d_0 = 3072$, and $L = 5$ hidden layers of dimensions $d_1 = \ldots = d_5 = 10000$. Strong principal component structure may cause the training samples to have large pairwise inner-products, which is shown in Appendix J.1. Thus, we pre-process the training samples by removing the leading 10 PCs—a few example images before and after this removal are depicted in Appendix J.3. A close agreement between the observed and limit spectra is displayed in Figure 2, for both $K^{\mathrm{CK}}$ and $K^{\mathrm{NTK}}$. Results without removing these leading 10 PCs are presented in Appendix J.2, where there is close agreement for $K^{\mathrm{CK}}$ but a deviation from the theoretical prediction for $K^{\mathrm{NTK}}$. This suggests that the approximation in Lemma 3.5 is sensitive to large but low-rank perturbations of $X$.

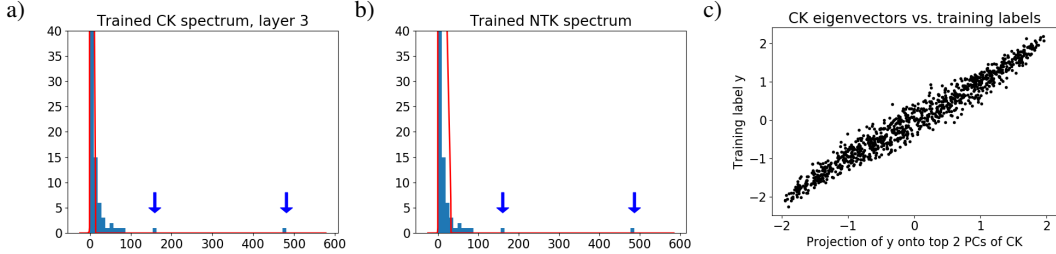

Figure 3: Eigenvalues of (a) $K^{\text{CK}}$ and (b) $K^{\text{NTK}}$ in a *trained* network, for training labels $y_\alpha = \sigma(\mathbf{x}_\alpha^\top \mathbf{v})$. The limit spectra at random initialization of weights are shown in red. Large outlier eigenvalues, indicated by blue arrows, emerge over training. (c) The projection of training labels onto the first 2 eigenvectors of the trained matrix $K^{\text{CK}}$ accounts for 96% of the training label variance.

### 4.3 CK and NTK spectra after training

We consider $n = 1000$ training samples $(\mathbf{x}_\alpha, y_\alpha)$, with $\mathbf{x}_\alpha$ uniformly distributed on the unit sphere of dimension $d_0 = 800$, and $y_\alpha = \sigma(\mathbf{x}_\alpha^\top \mathbf{v})$ for a fixed $\mathbf{v} \in \mathbb{R}^{d_0}$ on the sphere of radius $\sqrt{d_0}$. We train a 3-layer network with widths $d_1 = d_2 = d_3 = 800$, without biases, using the Adam optimizer in Keras with learning rate 0.01, batch size 32, and 300 training epochs. The final mean-squared training error is $10^{-4}$, and the test-sample prediction-$R^2$ is 0.81.

Figure 3 depicts the spectra of $K^{\text{CK}}$ and $K^{\text{NTK}}$ for the trained weights $\theta$. Intermediate layers are shown in Appendix J.4. We observe that the bulk spectra of $K^{\text{CK}}$ and $K^{\text{NTK}}$ are elongated from their random initializations. Furthermore, large outlier eigenvalues emerge in both $K^{\text{CK}}$ and $K^{\text{NTK}}$ over training. The corresponding eigenvectors are highly predictive of the training labels $\mathbf{y}$, suggesting the emergence of these eigenvectors as the primary mechanism of training in this example.

We describe in Appendix J.5 a second training example for a binary classification task on CIFAR-10, where similar qualitative phenomena are observed for the trained $K^{\text{CK}}$. This may suggest a path to understanding the learning process of deep neural networks, for future study.

## 5 Conclusion

We have provided analytic descriptions of the empirical eigenvalue distributions of the Conjugate Kernel (CK) and Neural Tangent Kernel (NTK) of large feedforward neural networks at random initialization, under a general condition for the input samples. Our work uses techniques of random matrix theory to provide an asymptotic analysis in a limiting regime where network width grows linearly with sample size. The resulting limit spectra exhibit "high-dimensional noise" that is not present in analyses of the infinite-width limit alone. This type of high-dimensional limit has been previously studied for networks with a single hidden layer, and our work develops new proof techniques to extend these characterizations to multi-layer networks, in a systematic and recursive form.

Our results contribute to the theoretical understanding of neural networks in two ways: First, an increasingly large body of literature studies the training and generalization errors of linear regression models using random features derived from the neural network CK and NTK. In the linear-width setting of our current paper, such results are typically based on asymptotic approximations for the Stieltjes transforms and resolvents of the associated kernel and covariance matrices. Our work develops theoretical tools that may enable the extension of these studies to random features regression models that are derived from deep networks with possibly many layers.

Second, the linear-width asymptotic regime may provide a simple setting for studying feature learning and neural network training outside of the "lazy" regime, and which is arguably closer to the operating regimes of neural network models in some practical applications. Our experimental results suggest interesting phenomena in the spectral evolutions of the CK and NTK that may potentially arise during training in this regime, and our theoretical characterizations of their spectra for random weights may provide a first step towards the analysis of these phenomena.

## Broader Impact

This work performs theoretical analysis that aims to extend our understanding of training and generalization in multi-layer neural networks. A better theoretical understanding of training and generalization in these models may ultimately help us to (1) understand the mechanisms by which social biases may be propagated by artificial systems, and prevent this from occurring, and (2) increase the robustness and fault-tolerance of artificial systems built on such models.

## Acknowledgments and Disclosure of Funding

This research is supported in part by NSF Grant DMS-1916198. We would like to thank John Lafferty and Ganlin Song for helpful discussions regarding the Neural Tangent Kernel.

## Footnotes

[1]In this paper, we use "conjugate kernel" and "neural tangent kernel" to refer to these matrices for a finite-width network, rather than their infinite-width limits.

[2]We caution that this does not imply convergence of the largest and smallest eigenvalues of $K$ to the support of $\mu$, which is a stronger notion of convergence than what we study in this work.

[3]Note that some authors use a negative sign convention and define $m_\mu(z)$ as $\int 1/(z - x) d\mu(x)$.

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
