[Supplementary Material · supplementary.pdf]

# A  Numerical solution of the fixed-point equations

Theorem 3.7 characterizes the limit Stieltjes transform $m(z)$ of matrices such as $K^{\mathrm{CK}}$ and $K^{\mathrm{NTK}}$. By the discussion in Section 2.2, a numerical approximation to the density functions of the corresponding spectral distributions may be obtained by computing $m(z)$ for $z = x + i\eta$, across a fine grid of values $x \in \mathbb{R}$ and for a fixed small imaginary part $\eta > 0$. We describe here one possible approach for this computation.

To compute the limit spectrum for $z_{-1}\operatorname{Id} + z_0 X_0^\top X_0 + \ldots + z_L X_L^\top X_L$ and general values $z_{-1}, \ldots, z_L \in \mathbb{R}$, fix the spectral argument $z = x + i\eta$ and denote

$$\mathbf{z}_L = (-z + z_{-1}, z_0, \ldots, z_L), \quad \mathbf{z}_{L-1} = \mathbf{z}_{\mathrm{prev}}(s_L(\mathbf{z}_L), \mathbf{z}_L), \quad \mathbf{z}_{L-2} = \mathbf{z}_{\mathrm{prev}}(s_{L-1}(\mathbf{z}_{L-1}), \mathbf{z}_{L-1}), \text{ etc.}$$

Here, for $s \in \mathbb{C}^+$ and $\mathbf{z} \in \mathbb{C}^- \times \mathbb{R}^\ell \times \mathbb{C}^*$, the quantity

$$\mathbf{z}_{\mathrm{prev}}(s, \mathbf{z}) = \left(z_{-1} + \frac{1 - b_\sigma^2}{s}, z_0, \ldots, z_{\ell-2}, z_{\ell-1} + \frac{b_\sigma^2}{s}\right) \in \mathbb{C}^- \times \mathbb{R}^{\ell-1} \times \mathbb{C}^*$$

is as defined in (14). Denote $s_\ell \equiv s_\ell(\mathbf{z}_\ell)$ for each $\ell = 1, \ldots, L$. Observe that, if we are given $s_1, \ldots, s_L$, then the value $t_\ell(\mathbf{z}_\ell, \mathbf{w})$ may be directly computed from (13), for any $\ell \in \{0, \ldots, L\}$ and any vector $\mathbf{w} \in \mathbb{C}^{\ell+2}$. This is because the fixed points needed to compute the arguments $\mathbf{z}_{\mathrm{prev}}(s_\ell(\mathbf{z}_\ell), \mathbf{z}_\ell)$, $\mathbf{z}_{\mathrm{prev}}(s_{\ell-1}(\mathbf{z}_{\ell-1}), \mathbf{z}_{\ell-1})$, etc. for the successive evaluations of $t_\ell$, $t_{\ell-1}$, etc. are provided by this given sequence $s_1, \ldots, s_L$.

Thus, we apply an iterative procedure of initializing $s_1^{(0)}, \ldots, s_L^{(0)} \in \mathbb{C}^+$, and computing the *simultaneous* updates $s_1^{(t+1)}, \ldots, s_L^{(t+1)}$ using the previous values $s_1^{(t)}, \ldots, s_L^{(t)}$. That is, we iterate the following two steps:

1. Set $\mathbf{z}_L^{(t)} = \mathbf{z}_L$, and compute $\mathbf{z}_{L-1}^{(t)} = \mathbf{z}_{\mathrm{prev}}(s_L^{(t)}, \mathbf{z}_L^{(t)})$, $\mathbf{z}_{L-2}^{(t)} = \mathbf{z}_{\mathrm{prev}}(s_{L-1}^{(t)}, \mathbf{z}_{L-1}^{(t)})$, etc.

2. Compute an update $s_\ell^{(t+1)}$ for the value of $s_\ell(\mathbf{z}_\ell)$ and each $\ell = 1, \ldots, L$, using the right side of (12) with $\mathbf{z}_\ell^{(t)}$ and $\mathbf{z}_{\ell-1}^{(t)} \equiv \mathbf{z}_{\mathrm{prev}}(s_\ell^{(t)}, \mathbf{z}_\ell^{(t)})$ in place of $\mathbf{z}_\ell$ and $\mathbf{z}_{\mathrm{prev}}(s_\ell(\mathbf{z}_\ell), \mathbf{z}_\ell)$.

After this iteration converges to fixed points $s_1^*, \ldots, s_L^*$, we then compute $m(z) = t_L(\mathbf{z}_L, (1, 0, \ldots, 0))$ using (13) and these fixed points. For each successive value $z = x + i\eta$ along the grid of values $x \in \mathbb{R}$, we initialize $s_1^{(0)}, \ldots, s_L^{(0)}$ by linear interpolation from the computed fixed points at the preceding two values of $x$ along this grid, for faster computation.

Note that for each value $z = x + i\eta$, if the above iteration converges to fixed points $s_1^*, \ldots, s_L^* \in \mathbb{C}^+$, then this procedure computes the correct value for $m(z)$: This is because, denoting

$$\mathbf{z}_{L-1}^* = \mathbf{z}_{\mathrm{prev}}(s_L^*, \mathbf{z}_L), \quad \mathbf{z}_{L-2}^* = \mathbf{z}_{\mathrm{prev}}(s_{L-1}^*, \mathbf{z}_{L-1}^*), \quad \ldots, \quad \mathbf{z}_1^* = \mathbf{z}_{\mathrm{prev}}(s_2^*, \mathbf{z}_2^*),$$

it may be checked iteratively from (12,13) and the uniqueness guarantee of Proposition 3.6 that $s_1^* = s_1(\mathbf{z}_1^*)$, then $s_2^* = s_2(\mathbf{z}_2^*)$, etc., and finally that $s_L^* = s_L(\mathbf{z}_L)$. This then means that $\mathbf{z}_{L-1}^* = \mathbf{z}_{\mathrm{prev}}(s_L(\mathbf{z}_L), \mathbf{z}_L) = \mathbf{z}_{L-1}$, then $\mathbf{z}_{L-2}^* = \mathbf{z}_{\mathrm{prev}}(s_{L-1}(\mathbf{z}_{L-1}), \mathbf{z}_{L-1}) = \mathbf{z}_{L-2}$, etc., and so $s_\ell^* = s_\ell(\mathbf{z}_\ell)$ for each $\ell$. Then this method computes the correct value for $m(z) = t_L(\mathbf{z}_L, (1, 0, \ldots, 0))$.

We have found in practice that the above iteration occasionally converges to fixed points $s_1, \ldots, s_L$ not belonging to $\mathbb{C}^+$ (i.e. this is not a mapping from $(\mathbb{C}^+)^L$ to $(\mathbb{C}^+)^L$). If this occurs, we randomly re-initialize $s_1^{(0)}, \ldots, s_L^{(0)} \in \mathbb{C}^+$, and we have found that the method reaches the correct fixed point within a small number of random initializations.

To clarify this approach, let us illustrate this computation in a simple example: Consider $L = 2$. Fix any grid value $x \in \mathbb{R}$ and $\eta > 0$. An approximate density function for the limit spectrum of $X_2^\top X_2$ at $x$ is given by $\frac{1}{\pi} \operatorname{Im} t_2((-z, 0, 0, 1), (1, 0, 0, 0))$, where $z = x + i\eta$. Based on equations (11,12,13),

$$
\begin{aligned}
t_2((-z, 0, 0, 1), (1, 0, 0, 0)) &= t_1\left(\left(-z + \frac{1 - b_\sigma^2}{s_2}, 0, \frac{b_\sigma^2}{s_2}\right), (1, 0, 0)\right) \\
&= t_0\left(\left(-z + \frac{1 - b_\sigma^2}{s_2} + \frac{1 - b_\sigma^2}{s_1}, \frac{b_\sigma^2}{s_1}\right), (1, 0)\right) \\
&= \int \left(-z + \frac{1 - b_\sigma^2}{s_2} + \frac{1 - b_\sigma^2}{s_1} + \frac{b_\sigma^2}{s_1} x\right)^{-1} d\mu_0(x),
\end{aligned}
$$

where $s_1, s_2 \in \mathbb{C}^+$ satisfy the fixed point equations

$$s_2 = 1 + \gamma_2 s_2 + \gamma_2 t_0 \left( \left( -z + \frac{1 - b_\sigma^2}{s_2} + \frac{1 - b_\sigma^2}{s_1}, \frac{b_\sigma^2}{s_1} \right), (s_2 z, 0) \right) \tag{18}$$

$$s_1 = \frac{s_2}{b_\sigma^2} + \gamma_1 t_0 \left( \left( -z + \frac{1 - b_\sigma^2}{s_2} + \frac{1 - b_\sigma^2}{s_1}, \frac{b_\sigma^2}{s_1} \right), (1 - b_\sigma^2, b_\sigma^2) \right). \tag{19}$$

We randomly initialize $s_1^{(0)}, s_2^{(0)} \in \mathbb{C}^+$, and update $s_1^{(t+1)}, s_2^{(t+1)}$ simultaneously by substituting $s_1 = s_1^{(t)}$ and $s_2 = s_2^{(t)}$ into the right side of (18) and (19). We iterate this until convergence, and then substitute into the above expression for $t_2((-z, 0, 0, 1), (1, 0, 0, 0))$ to approximate the limit spectral density of $X_2^\top X_2$ at $x$.

## B Proof of $(\varepsilon, B)$-orthonormality for independent input training samples

We prove Proposition 3.3. For convenience, in this section, we denote the input dimension $d_0$ simply as $d$, and we denote the rescaled input by $\widetilde{X} = \sqrt{d}\, X$, with columns $\tilde{\mathbf{x}}_\alpha = \sqrt{d} \cdot \mathbf{x}_\alpha$.

**Bound for $\|\tilde{\mathbf{x}}_\alpha\|^2$:** Note that $\mathbb{E}[\|\tilde{\mathbf{x}}_\alpha\|^2] = d$. Applying the convex concentration property and [1, Theorem 2.5] with $A = \mathrm{Id}$, we have for any $t > 0$ that

$$\mathbb{P}\Big[ \big| \|\tilde{\mathbf{x}}_\alpha\|^2 - d \big| > t \Big] \le 2\exp\left( -c\min\left( \frac{t^2}{d}, t \right) \right) \tag{20}$$

for a constant $c$ depending only on $c_0$. Applying this for $t = \sqrt{Kd\log n}$ and a union bound, with probability $1 - 2ne^{-cK\log n}$,

$$\Big| \|\tilde{\mathbf{x}}_\alpha\|^2 - d \Big| \le \sqrt{Kd\log n} \quad \text{for all } \alpha \in [n]. \tag{21}$$

Rescaling, this shows $\big| \|\mathbf{x}_\alpha\|^2 - 1 \big| \le \sqrt{(K\log n)/d}$.

**Bound for $\tilde{\mathbf{x}}_\alpha^\top \tilde{\mathbf{x}}_\beta$:** Since $\tilde{\mathbf{x}}_\alpha$ and $\tilde{\mathbf{x}}_\beta$ are independent, conditional on $\tilde{\mathbf{x}}_\beta$, we have $\mathbb{E}[\tilde{\mathbf{x}}_\alpha^\top \tilde{\mathbf{x}}_\beta \mid \tilde{\mathbf{x}}_\beta] = 0$, and the map $\tilde{\mathbf{x}}_\alpha \mapsto \tilde{\mathbf{x}}_\alpha^\top \tilde{\mathbf{x}}_\beta$ is convex and $\|\tilde{\mathbf{x}}_\beta\|$-Lipschitz. Then the convex concentration property implies, for any $t > 0$,

$$\mathbb{P}\Big[ |\tilde{\mathbf{x}}_\alpha^\top \tilde{\mathbf{x}}_\beta| > t \Big| \tilde{\mathbf{x}}_\beta \Big] \le 2e^{-c_0 t^2/\|\tilde{\mathbf{x}}_\beta\|^2}.$$

On the event (21), applying this for $t = \sqrt{Kd\log n}$, this probability is at most $2e^{-cK\log n}$. Taking a union bound, with probability $1 - 2n^2 e^{-cK\log n}$,

$$\Big| \tilde{\mathbf{x}}_\alpha^\top \tilde{\mathbf{x}}_\beta \Big| \le \sqrt{Kd\log n} \quad \text{for all } \alpha \ne \beta \in [n].$$

Rescaling, this shows $|\mathbf{x}_\alpha^\top \mathbf{x}_\beta| \le \sqrt{(K\log n)/d}$.

**Bound for $\|\widetilde{X}\|$:** Fix any unit vector $\mathbf{v} = (v_1, \dots, v_n) \in \mathbb{R}^n$. By [31, Lemma C.11], the random vector $\widetilde{X}\mathbf{v}$ also satisfies the convex concentration property, with a modified constant $c_0'$. Note that $\mathbb{E}[\|\widetilde{X}\mathbf{v}\|^2] = d\|\mathbf{v}\|^2 = d$. Then, as in (20), we have

$$\mathbb{P}\Big[ \big| \|\widetilde{X}\mathbf{v}\|^2 - d \big| > t \Big] \le 2\exp\left( -c\min\left( \frac{t^2}{d}, t \right) \right).$$

Applying this with $t = (B^2/4 - 1)d$, and taking a union bound over a $1/2$-net $\mathcal{N}$ of the unit ball $\{\mathbf{v} \in \mathbb{R}^n : \|\mathbf{v}\| = 1\}$ with cardinality $5^n$, we have with probability at least $1 - 5^n \cdot 2e^{-cB^2 d}$ that

$$\|\widetilde{X}\mathbf{v}\| \le (B/2)\sqrt{d} \quad \text{for all } \mathbf{v} \in \mathcal{N}.$$

Since

$$\|\widetilde{X}\| = \sup_{\mathbf{v}:\|\mathbf{v}\|=1} \|\widetilde{X}\mathbf{v}\| \le \sup_{\mathbf{v} \in \mathcal{N}} \|\widetilde{X}\mathbf{v}\| + \|\widetilde{X}\|/2,$$

we have $\|\widetilde{X}\| \le B\sqrt{d}$ on this event. Rescaling, this shows $\|X\| \le B$.

**Bound for $\sum_{\alpha=1}^{n}(\|\tilde{\mathbf{x}}_\alpha\|^2 - d)^2$:** Define $\mathbf{z} = (z_1, \ldots, z_n)$ where $\mathbf{z}_\alpha = \|\tilde{\mathbf{x}}_\alpha\|^2 - d$. Fixing any unit vector $\mathbf{v} = (v_1, \ldots, v_n) \in \mathbb{R}^n$, let us first bound $\mathbf{v}^\top \mathbf{z}$: We have

$$\mathbf{v}^\top \mathbf{z} = \sum_{\alpha=1}^{n} v_\alpha(\|\tilde{\mathbf{x}}_\alpha\|^2 - d),$$

which has mean 0. Note that integrating the tail bound (20) yields the sub-exponential condition

$$\mathbb{E}\left[\exp\left(\lambda(\|\tilde{\mathbf{x}}_\alpha\|^2 - d)\right)\right] \leq \exp(Cd\lambda^2) \quad \text{for all } |\lambda| \leq c'$$

and some constants $C, c' > 0$. (See e.g. [10, Theorem 2.3], applied with $(v, c) = (C'd, C')$ and a large enough constant $C' > 0$.) Then, as $\tilde{\mathbf{x}}_1, \ldots, \tilde{\mathbf{x}}_n$ are independent and $\|\mathbf{v}\|^2 = 1$, also

$$\mathbb{E}[e^{\lambda \mathbf{v}^\top \mathbf{z}}] = \mathbb{E}\left[\exp\left(\lambda \sum_{\alpha=1}^{n} v_\alpha(\|\tilde{\mathbf{x}}_\alpha\|^2 - d)\right)\right] \leq \exp(Cd\lambda^2) \quad \text{for all } |\lambda| \leq c'.$$

For any $t > 0$, applying this with $\lambda = \min(t/(2Cd), c')$ yields the sub-exponential tail bound

$$\mathbb{P}[\mathbf{v}^\top \mathbf{z} \geq t] \leq e^{-\lambda t} \mathbb{E}[e^{\lambda \mathbf{v}^\top \mathbf{z}}] \leq \exp\left(-c \min\left(\frac{t^2}{d}, t\right)\right).$$

Now applying this for $t = (B/2)d$, and again taking a union bound over a $1/2$-net $\mathcal{N}$ of the unit ball, we have with probability $1 - 5^n \cdot e^{-cBd}$ that

$$\mathbf{v}^\top \mathbf{z} \leq (B/2)d \quad \text{for all } \mathbf{v} \in \mathcal{N}.$$

On this event, we have as above that $\|\mathbf{z}\| \leq Bd$, so $\|\mathbf{z}\|^2 \leq B^2 d^2$. Rescaling, this shows $\sum_{\alpha=1}^{n}(\|\tilde{\mathbf{x}}_\alpha\|^2 - 1)^2 \leq B^2$.

Applying all of the above bounds for sufficiently large constants $K, B > 0$, we obtain that these bounds hold with probability at least $1 - n^{-k}$, which yields Proposition 3.3.

## C  Overview of proofs and preliminary lemmas

The proofs of Theorems 3.4, 3.7, and 3.8 are contained in the subsequent Appendices D–H. We provide here an outline of the argument.

We will apply induction across the layers $\ell = 1, \ldots, L$, analyzing the post-activation matrix $X_\ell$ of each layer conditional on the previous post-activations $X_0, \ldots, X_{\ell-1}$ (i.e. with respect to only the randomness of $W_\ell$). For the Conjugate Kernel, this will entail analyzing the Stieltjes transform

$$\frac{1}{n} \operatorname{Tr}(X_L^\top X_L - z \operatorname{Id})^{-1}$$

conditional on the previous layers. For the Neural Tangent Kernel, given the approximation in Lemma 3.5, this will entail analyzing the Stieltjes transform

$$\frac{1}{n} \operatorname{Tr}(A + X_L^\top X_L - z \operatorname{Id})^{-1}$$

conditional on the previous layers, where $A$ is a linear combination of $X_0^\top X_0, \ldots, X_{L-1}^\top X_{L-1}$, and $\operatorname{Id}$. Note that this matrix $A$ is deterministic conditional on the previous layers.

In Appendix D, we carry out a non-asymptotic analysis of $(\varepsilon, B)$-orthonormality. In particular, we show that if the deterministic input $X \equiv X_0$ is $(\varepsilon, B)$-orthonormal, then $X_1$ is $(C\varepsilon, CB)$-orthonormal with high probability, for a constant $C > 0$ depending only on $\lambda_\sigma$. Note that we require the fourth technical condition

$$\sum_{\alpha=1}^{n}(\|\mathbf{x}_\alpha\|^2 - 1)^2 \leq B^2$$

in Definition 3.1 to ensure that the operator norm $\|X_1\|$ remains of constant order, as otherwise $X_1$ may have a rank-one component whose norm grows slowly with $n$. Applying this result conditionally for every layer, Assumption 3.2 then implies that $X_0, \ldots, X_L$ are all $(\tilde{\varepsilon}_n, \tilde{B})$-orthonormal for modified parameters $(\tilde{\varepsilon}_n, \tilde{B})$ with high probability.

In Appendix E, we carry out the analysis of the trace

$$\frac{1}{n} \operatorname{Tr}(A + \alpha X_1^\top X_1 - z \operatorname{Id})^{-1}$$

in a single layer, for a deterministic $(\varepsilon_n, B)$-orthonormal input $X_0$, symmetric matrix $A \in \mathbb{R}^{n \times n}$, and spectral parameters $\alpha \in \mathbb{C}^* \equiv \overline{\mathbb{C}^-} \setminus 0$ and $z \in \mathbb{C}^+$. We allow $\alpha \in \mathbb{C}^*$ (rather than fixing $\alpha = 1$), as the subsequent induction argument for the NTK will require this extension. When $A = 0$ and $\alpha = 1$, this reduces to the analysis in [39], and also mirrors the proof of the Marcenko-Pastur equation (6). For $A \neq 0$, this trace will depend jointly on $A$ and the second-moment matrix $\Phi_1 \in \mathbb{R}^{n \times n}$ for the rows of $X_1$. We derive a fixed-point equation in terms of $A$ and $\Phi_1$, which approximates this trace in the $n \to \infty$ limit.

In Appendix F, we prove Theorem 3.4 on the CK, by specializing this analysis to the setting $A = 0$ and $\alpha = 1$. The inductive loop is closed via an entrywise approximation of the second-moment matrix $\Phi_\ell$ in each layer by a linear combination of $X_{\ell-1}^\top X_{\ell-1}$ and Id in the previous layer. The main argument for this approximation has been carried out in Appendix D.

In Appendix G, we prove Theorem 3.7 on the NTK. Our analysis reduces the trace of any linear combination of $X_0^\top X_0, \ldots, X_L^\top X_L$ and Id to the trace of a more general rational function of $X_0^\top X_0, \ldots, X_{L-1}^\top X_{L-1}$ and Id in the previous layer. In order to close the inductive loop, we analyze the trace of such a rational function across layers, and show that it may be characterized by the recursive fixed-point equations (12) and (13). In Appendix G, we also establish the approximation in Lemma 3.5 and the existence and uniqueness of the fixed point to (12).

Finally, in Appendix H, we prove Theorem 3.8, which is a minor extension of Theorem 3.7.

**Notation.** In the proof, $\mathbf{v}^*$ and $M^*$ denote the conjugate transpose. For a complex matrix $M \in \mathbb{C}^{n \times n}$, we denote by

$$\operatorname{tr} M = n^{-1} \operatorname{Tr} M$$

the normalized matrix trace, by $\|M\| = \sup_{\mathbf{v} \in \mathbb{C}^n : \|\mathbf{v}\|=1} \|M\mathbf{v}\|$ the operator norm, and by $\|M\|_F = (\operatorname{Tr} M^*M)^{1/2} = (\sum_{\alpha,\beta} |M_{\alpha\beta}|^2)^{1/2}$ the Frobenius norm. Note that we have

$$|\operatorname{tr} M| \leq \|M\| \leq \|M\|_F, \qquad \|M\|_F \leq \sqrt{n}\|M\|, \qquad |\operatorname{tr} AB| \leq n^{-1}\|A\|_F\|B\|_F.$$

Let us collect here a few basic results, which we will use in the subsequent sections.

**Proposition C.1.** *Under Assumption 3.2(b), the constants $a_\sigma$ and $b_\sigma$ in (7) satisfy*

$$|b_\sigma| \leq 1 \leq \sqrt{a_\sigma} \leq \lambda_\sigma.$$

*For a universal constant $C > 0$, the activation function $\sigma$ satisfies*

$$|\sigma(x)| \leq \lambda_\sigma(|x| + C) \qquad \text{for all } x \in \mathbb{R}. \tag{22}$$

*Proof.* It is clear from definition that $a_\sigma \leq \lambda_\sigma^2$. By the Gaussian Poincaré inequality,

$$1 = \mathbb{E}[\sigma(\xi)^2] = \operatorname{Var}[\sigma(\xi)] \leq \mathbb{E}[\sigma'(\xi)^2] = a_\sigma.$$

By Gaussian integration-by-parts and Cauchy-Schwarz,

$$|b_\sigma| = |\mathbb{E}[\sigma'(\xi)]| = |\mathbb{E}[\xi \cdot \sigma(\xi)]| \leq \mathbb{E}[\xi^2]^{1/2}\mathbb{E}[\sigma(\xi)^2]^{1/2} = 1.$$

We have

$$|\sigma(0)| \leq \mathbb{E}[|\sigma(0) - \sigma(\xi)|] + \mathbb{E}[|\sigma(\xi)|] \leq \lambda_\sigma\mathbb{E}[|\xi|] + \mathbb{E}[\sigma(\xi)^2]^{1/2} \leq C\lambda_\sigma \tag{23}$$

(the last inequality applying $\lambda_\sigma \geq 1$). Then $|\sigma(x)| \leq |\sigma(0)| + \lambda_\sigma|x| \leq \lambda_\sigma(|x| + C)$. $\square$

**Proposition C.2.** *Suppose $M = U + iV \in \mathbb{C}^{n \times n}$, where the real and imaginary parts $U, V \in \mathbb{R}^{n \times n}$ are symmetric, and $V$ is invertible with either $V \succeq c_0 \operatorname{Id}$ or $V \preceq -c_0 \operatorname{Id}$ for a value $c_0 > 0$. Then $M$ is invertible, and $\|M^{-1}\| \leq 1/c_0$.*

*Proof.* For any unit vector $\mathbf{v} \in \mathbb{C}^n$,

$$\|M\mathbf{v}\| = \|M\mathbf{v}\| \cdot \|\mathbf{v}\| \geq |\mathbf{v}^*M\mathbf{v}| = |\mathbf{v}^*U\mathbf{v} + i \cdot \mathbf{v}^*V\mathbf{v}| \geq |\mathbf{v}^*V\mathbf{v}|,$$

the last step holding because $U, V$ are real-symmetric so that $\mathbf{v}^*U\mathbf{v}$ and $\mathbf{v}^*V\mathbf{v}$ are both real. By the given assumption on $V$, we have $|\mathbf{v}^*V\mathbf{v}| \geq c_0$, so $\|M\mathbf{v}\| \geq c_0$ for every unit vector $\mathbf{v} \in \mathbb{C}^n$. Then $M$ is invertible, and $\|M^{-1}\| \leq 1/c_0$. $\square$

**Proposition C.3.** *Let $M, \widetilde{M} \in \mathbb{R}^{n \times n}$ be any two symmetric matrices satisfying*

$$\frac{1}{n}\|M - \widetilde{M}\|_F^2 \to 0$$

*a.s. as $n \to \infty$. If $\lim \operatorname{spec} M = \nu$ for a probability distribution $\nu$ on $\mathbb{R}$, then also $\lim \operatorname{spec} \widetilde{M} = \nu$.*

*Proof.* For fixed $z \in \mathbb{C}^+$, let $m(z) = \operatorname{tr}(M - z \operatorname{Id})^{-1}$ and $\tilde{m}(z) = \operatorname{tr}(\widetilde{M} - z \operatorname{Id})^{-1}$ be the Stieltjes transforms. Then applying $A^{-1} - B^{-1} = A^{-1}(B - A)B^{-1}$, we may bound their difference by

$$
\begin{aligned}
|m(z) - \tilde{m}(z)|^2 &= \frac{1}{n^2}\left| \operatorname{Tr}[(M - z \operatorname{Id})^{-1} - (\widetilde{M} - z \operatorname{Id})^{-1}] \right|^2 \\
&= \frac{1}{n^2}\left| \operatorname{Tr}(M - z \operatorname{Id})^{-1}(\widetilde{M} - M)(\widetilde{M} - z \operatorname{Id})^{-1} \right|^2 \\
&\leq \frac{1}{n^2}\|\widetilde{M} - M\|_F^2 \|(M - z \operatorname{Id})^{-1}(\widetilde{M} - z \operatorname{Id})^{-1}\|_F^2 \\
&\leq \frac{1}{n}\|\widetilde{M} - M\|_F^2 \|(M - z \operatorname{Id})^{-1}\|^2 \|(\widetilde{M} - z \operatorname{Id})^{-1}\|^2
\end{aligned}
$$

Applying $\|(M - z \operatorname{Id})^{-1}\| \leq 1/\operatorname{Im} z$ by Proposition C.2, and similarly for $\widetilde{M}$, the given condition shows that $m(z) - \tilde{m}(z) \to 0$ a.s., pointwise over $z \in \mathbb{C}^+$. If $\lim \operatorname{spec} M = \nu$, then $m(z) \to m_\nu(z) \equiv \int (x - z)^{-1} d\nu(x)$ a.s., and hence also $\tilde{m}(z) \to m_\nu(z)$ a.s. and $\lim \operatorname{spec} \widetilde{M} = \nu$. $\qquad \square$

## D  Propagation of approximate pairwise orthogonality

In this section, we work in the following (non-asymptotic) setting of a single layer: Consider any deterministic matrix $X \in \mathbb{R}^{d \times n}$, let $W \in \mathbb{R}^{\check{d} \times d}$ have i.i.d. $\mathcal{N}(0, 1)$ entries, and set

$$\check{X} = \frac{1}{\sqrt{\check{d}}}\sigma(WX) \in \mathbb{R}^{\check{d} \times n}. \tag{24}$$

Note that $\check{X}$ has i.i.d. rows with distribution $\sigma(\mathbf{w}^\top X)/\sqrt{\check{d}}$, where $\mathbf{w} \sim \mathcal{N}(0, \operatorname{Id})$. Define the second-moment matrix of $\check{X}$ by

$$\Phi = \mathbb{E}[\check{X}^\top \check{X}] = \mathbb{E}[\sigma(\mathbf{w}^\top X)^\top \sigma(\mathbf{w}^\top X)] \in \mathbb{R}^{n \times n} \tag{25}$$

where the expectations are over the standard Gaussian matrix $W$ and standard Gaussian vector $\mathbf{w}$. Let $\Phi_{\alpha\beta}$ denote the $(\alpha, \beta)$ entry of $\Phi$ for any $\alpha, \beta \in [n]$. We show in this section the following result.

**Lemma D.1.** *Suppose $X$ is $(\varepsilon, B)$-orthonormal where $\varepsilon < 1/\lambda_\sigma$. Then for universal constants $C, c > 0$, with probability at least $1 - 2n^2 e^{-c\check{d}\varepsilon^2} - 3e^{-cn}$, the matrix $\check{X}$ remains $(\check{\varepsilon}, \check{B})$-orthonormal with*

$$\check{\varepsilon} = C\lambda_\sigma^2 \varepsilon, \qquad \check{B} = C\left(1 + n/\check{d}\right)\lambda_\sigma^2 B.$$

**Corollary D.2.** *Under Assumption 3.2, there exist parameters $(\tilde{\varepsilon}_n, \tilde{B})$ still satisfying $\tilde{\varepsilon}_n n^{1/4} \to 0$, such that a.s. for all large $n$, every matrix $X_0, \ldots, X_L$ is $(\tilde{\varepsilon}_n, \tilde{B})$-orthonormal.*

*Proof.* Note that increasing $\varepsilon_n$ represents a weaker assumption, so we may assume without loss of generality that $\varepsilon_n \geq n^{-0.49}$. Then by Lemma D.1, there is a constant $C_0 \geq 1$ depending on $\lambda_\sigma, \gamma_1, \ldots, \gamma_L$, such that if $X_{\ell-1}$ is $(C_0^{\ell-1}\varepsilon_n, C_0^{\ell-1}B)$-orthonormal, then conditional on this event, $X_\ell$ is $(C_0^\ell \varepsilon_n, C_0^\ell B)$-orthonormal with probability at least $1 - e^{-n^{0.01}}$ for all large $n$. Thus, setting $\tilde{\varepsilon}_n = C_0^L \varepsilon_n$ and $\tilde{B} = C_0^L B$, with probability at least $1 - Le^{-n^{0.01}}$, every matrix $X_0, \ldots, X_L$ is $(\tilde{\varepsilon}_n, \tilde{B})$-orthonormal. The almost sure statement then follows from the Borel-Cantelli Lemma. $\quad \square$

In the remainder of this section, we prove Lemma D.1. We divide the proof into Lemmas D.3, D.4, and D.5 below, which check the individual requirements for $(\check{\varepsilon}, \check{B})$-orthonormality of $\check{X}$. We denote by $C, C', c, c' > 0$ universal constants that may change from instance to instance.

**Lemma D.3.** *If $X$ is $(\varepsilon, B)$-orthonormal where $\varepsilon < 1/\lambda_\sigma$, then for universal constants $C, c > 0$:*

*(a) For all $\alpha \neq \beta \in [n]$,*

$$|\Phi_{\alpha\beta} - b_\sigma^2 \mathbf{x}_\alpha^\top \mathbf{x}_\beta| \leq C\lambda_\sigma^2 \varepsilon^2 \tag{26}$$

$$\left|\mathbb{E}_{\mathbf{w}\sim\mathcal{N}(0,\mathrm{Id})}[\sigma(\mathbf{w}^\top \mathbf{x}_\alpha)]\right| \leq C\lambda_\sigma \left|\|\mathbf{x}_\alpha\|^2 - 1\right| \leq C\lambda_\sigma \varepsilon \tag{27}$$

$$|\Phi_{\alpha\alpha} - 1| \leq C\lambda_\sigma \left|\|\mathbf{x}_\alpha\|^2 - 1\right| \leq C\lambda_\sigma \varepsilon \tag{28}$$

*(b) With probability at least $1 - 2n^2 e^{-c\check{d}\varepsilon^2}$, simultaneously for all $\alpha \neq \beta \in [n]$, the columns of $\check{X}$ satisfy*

$$\left|\|\check{\mathbf{x}}_\alpha\|^2 - 1\right| \leq C\lambda_\sigma^2 \varepsilon, \qquad \left|\check{\mathbf{x}}_\alpha^\top \check{\mathbf{x}}_\beta\right| \leq C\lambda_\sigma^2 \varepsilon.$$

Note that (26) establishes an approximation which is second-order in $\varepsilon$—this will be important in our later arguments which approximate $\Phi$ in Frobenius norm.

*Proof.* For part (a), observe that $(\zeta_\alpha, \zeta_\beta) \equiv (\mathbf{w}^\top \mathbf{x}_\alpha, \mathbf{w}^\top \mathbf{x}_\beta)$ is bivariate Gaussian, with mean 0 and covariance

$$\Sigma = \begin{pmatrix} \|\mathbf{x}_\alpha\|^2 & \mathbf{x}_\alpha^\top \mathbf{x}_\beta \\ \mathbf{x}_\alpha^\top \mathbf{x}_\beta & \|\mathbf{x}_\beta\|^2 \end{pmatrix} = \mathrm{Id} + \Delta$$

where $\Delta$ is entrywise bounded by $\varepsilon$. Then performing a Gram-Schmidt orthogonalization procedure, for some independent standard Gaussian variables $\xi_\alpha, \xi_\beta \sim \mathcal{N}(0,1)$, we have

$$\zeta_\alpha = u_\alpha \xi_\alpha, \qquad \zeta_\beta = u_\beta \xi_\beta + v_\beta \xi_\alpha \tag{29}$$

where $u_\alpha, u_\beta > 0$ and $v_\beta \in \mathbb{R}$ satisfy $|u_\alpha - 1|, |u_\beta - 1|, |v_\beta| \leq C\varepsilon$ for a universal constant $C > 0$.

By a Taylor expansion of $\sigma(\zeta)$ around $\zeta = \xi$, there exists a random variable $\eta$ between $\zeta$ and $\xi$ such that

$$\sigma(\zeta) = \sigma(\xi) + \sigma'(\xi)(\zeta - \xi) + \frac{1}{2}\sigma''(\eta)(\zeta - \xi)^2. \tag{30}$$

For $\alpha \neq \beta$, applying this for both $\zeta_\alpha$ and $\zeta_\beta$, noting that the product of leading terms satisfies $\mathbb{E}[\sigma(\xi_\alpha)\sigma(\xi_\beta)] = 0$, and applying also the bounds $|\sigma'(x)|, |\sigma''(x)| \leq \lambda_\sigma$ where $\lambda_\sigma \geq 1$, it is easy to check that

$$\Phi_{\alpha\beta} = \mathbb{E}[\sigma(\zeta_\alpha)\sigma(\zeta_\beta)] = \mathbb{E}\left[\sigma(\xi_\alpha) \cdot \sigma'(\xi_\beta)(\zeta_\beta - \xi_\beta) + \sigma(\xi_\beta) \cdot \sigma'(\xi_\alpha)(\zeta_\alpha - \xi_\alpha)\right] + \text{remainder}$$

where this remainder has magnitude at most $C\lambda_\sigma^2 \varepsilon^2$. For the first term, substituting (29) and applying independence of $\xi_\alpha$ and $\xi_\beta$, we have

$$\mathbb{E}\left[\sigma(\xi_\alpha) \cdot \sigma'(\xi_\beta)(\zeta_\beta - \xi_\beta) + \sigma(\xi_\beta) \cdot \sigma'(\xi_\alpha)(\zeta_\alpha - \xi_\alpha)\right]$$
$$= (u_\beta - 1)\mathbb{E}[\sigma(\xi_\alpha)] \cdot \mathbb{E}[\sigma'(\xi_\beta)\xi_\beta] + v_\beta \mathbb{E}[\sigma(\xi_\alpha)\xi_\alpha] \cdot \mathbb{E}[\sigma'(\xi_\beta)] + (u_\alpha - 1)\mathbb{E}[\sigma(\xi_\beta)] \cdot \mathbb{E}[\sigma'(\xi_\alpha)\xi_\alpha].$$

Applying $\mathbb{E}[\sigma(\xi)] = 0$ and the integration-by-parts identity $\mathbb{E}[\sigma(\xi)\xi] = \mathbb{E}[\sigma'(\xi)] = b_\sigma$, this term equals $v_\beta b_\sigma^2$. From (29), we have $u_\alpha v_\beta = \mathbb{E}[\zeta_\alpha \zeta_\beta] = \mathbf{x}_\alpha^\top \mathbf{x}_\beta$. Since $|u_\alpha - 1| \leq C\varepsilon$ and $|\mathbf{x}_\alpha^\top \mathbf{x}_\beta| \leq \varepsilon$, this implies $|v_\beta b_\sigma^2 - b_\sigma^2 \mathbf{x}_\alpha^\top \mathbf{x}_\beta| \leq Cb_\sigma^2 \varepsilon^2 \leq C\lambda_\sigma^2 \varepsilon^2$. Combining these yields (26). Similarly, from a first-order Taylor expansion analogous to (30),

$$\left|\mathbb{E}[\sigma(\mathbf{w}^\top \mathbf{x}_\alpha)]\right| = \left|\mathbb{E}[\sigma(\zeta_\alpha)] - \mathbb{E}[\sigma(\xi_\alpha)]\right| \leq C\lambda_\sigma \cdot |u_\alpha - 1|,$$

$$|\Phi_{\alpha\alpha} - 1| = \left|\mathbb{E}[\sigma(\zeta_\alpha)^2] - \mathbb{E}[\sigma(\xi_\alpha)^2]\right| \leq C\max\left(\lambda_\sigma \cdot |u_\alpha - 1|, \; \lambda_\sigma^2 \cdot |u_\alpha - 1|^2\right).$$

The bounds (27) and (28) follow from the observations $u_\alpha^2 = \mathbb{E}[\zeta_\alpha^2] = \|\mathbf{x}_\alpha\|^2$ and $|u_\alpha - 1| \leq |u_\alpha - 1| \cdot |u_\alpha + 1| = |u_\alpha^2 - 1| \leq \varepsilon$.

For part (b), let $\mathbf{w}_k^\top$ be the $k^{\text{th}}$ row of $W$. Then by definition of $\check{X}$, for any $\alpha, \beta \in [n]$ (including $\alpha = \beta$),

$$\check{\mathbf{x}}_\alpha^\top \check{\mathbf{x}}_\beta = \frac{1}{\check{d}} \sum_{k=1}^{\check{d}} \sigma\left(\mathbf{w}_k^\top \mathbf{x}_\alpha\right) \sigma\left(\mathbf{w}_k^\top \mathbf{x}_\beta\right).$$

We apply Bernstein's inequality: Denote by $\|\cdot\|_{\psi_2}$ and $\|\cdot\|_{\psi_1}$ the sub-Gaussian and sub-exponential norms of a random variable. For any deterministic vector $\mathbf{x} \in \mathbb{R}^d$, the function $\mathbf{w} \mapsto \sigma(\mathbf{w}^\top \mathbf{x})$ is $\lambda_\sigma \|\mathbf{x}\|$-Lipschitz. Then for $\mathbf{w} \sim \mathcal{N}(0, \mathrm{Id})$ and a universal constant $C > 0$, we have by Gaussian concentration-of-measure

$$\|\sigma(\mathbf{w}^\top \mathbf{x}_\alpha) - \mathbb{E}[\sigma(\mathbf{w}^\top \mathbf{x}_\alpha)]\|_{\psi_2} \leq C\lambda_\sigma \|\mathbf{x}_\alpha\|.$$

From (27), $|\mathbb{E}[\sigma(\mathbf{w}^\top \mathbf{x}_\alpha)]| \leq C\lambda_\sigma \varepsilon$. Thus (recalling that $|\|\mathbf{x}_\alpha\| - 1| \leq \varepsilon$), we have $\|\sigma(\mathbf{w}^\top \mathbf{x}_\alpha)\|_{\psi_2} \leq C\lambda_\sigma$ for a constant $C > 0$, and similarly for $\mathbf{x}_\beta$. So

$$\|\sigma(\mathbf{w}^\top \mathbf{x}_\alpha)\sigma(\mathbf{w}^\top \mathbf{x}_\beta)\|_{\psi_1} \leq \|\sigma(\mathbf{w}^\top \mathbf{x}_\alpha)\|_{\psi_2}\|\sigma(\mathbf{w}^\top \mathbf{x}_\beta)\|_{\psi_2} \leq C\lambda_\sigma^2. \tag{31}$$

Applying Bernstein's inequality (see [53, Theorem 2.8.1]), for a universal constant $c > 0$ and any $t > 0$,

$$\mathbb{P}\left[\left|\check{\mathbf{x}}_\alpha^\top \check{\mathbf{x}}_\beta - \mathbb{E}[\check{\mathbf{x}}_\alpha^\top \check{\mathbf{x}}_\beta]\right| > t\right] \leq 2\exp\left(-c\check{d}\min\left(\frac{t^2}{\lambda_\sigma^4}, \frac{t}{\lambda_\sigma^2}\right)\right).$$

Applying this for $t = \lambda_\sigma^2 \varepsilon$ and taking a union bound over all $\alpha, \beta \in [n]$, we get

$$\mathbb{P}\left[\left|\check{\mathbf{x}}_\alpha^\top \check{\mathbf{x}}_\beta - \mathbb{E}[\check{\mathbf{x}}_\alpha^\top \check{\mathbf{x}}_\beta]\right| \leq \lambda_\sigma^2 \varepsilon \text{ for all } \alpha, \beta \in [n]\right] \geq 1 - 2n^2 \exp\left(-c\check{d} \cdot \varepsilon^2\right). \tag{32}$$

Since $\mathbb{E}[\check{\mathbf{x}}_\alpha^\top \check{\mathbf{x}}_\beta] = \Phi_{\alpha\beta}$, part (b) now follows from part (a). $\qquad\square$

**Lemma D.4.** *If $X$ is $(\varepsilon, B)$-orthonormal where $\varepsilon < 1/\lambda_\sigma$, then for universal constants $C, c > 0$:*

*(a)* $\|\Phi\| \leq C\lambda_\sigma^2 B^2$.

*(b) With probability at least $1 - 2e^{-cn}$, $\|\check{X}\| \leq C\left(1 + \sqrt{n/\check{d}}\right)\lambda_\sigma B$.*

*Proof.* For part (a), define

$$\Sigma = \mathbb{E}\left[\sigma(\mathbf{w}^\top X)^\top \sigma(\mathbf{w}^\top X)\right] - \mathbb{E}[\sigma(\mathbf{w}^\top X)]^\top \mathbb{E}[\sigma(\mathbf{w}^\top X)] \tag{33}$$

where the first term on the right is $\Phi$. Then

$$\|\Sigma\| = \sup_{\mathbf{v}:\|\mathbf{v}\|=1} \mathbf{v}^\top \Sigma \mathbf{v} = \sup_{\mathbf{v}:\|\mathbf{v}\|=1}\left|\mathbb{E}\left[\left(\sigma(\mathbf{w}^\top X)\mathbf{v}\right)^2\right] - \mathbb{E}\left[\sigma(\mathbf{w}^\top X)\mathbf{v}\right]^2\right| = \sup_{\mathbf{v}:\|\mathbf{v}\|=1} \mathrm{Var}\left[\sigma(\mathbf{w}^\top X)\mathbf{v}\right].$$

We bound this variance using the Gaussian Poincaré inequality: Let us fix $\mathbf{v} \in \mathbb{R}^n$ with $\|\mathbf{v}\| = 1$ and define

$$F(\mathbf{w}) = \sigma(\mathbf{w}^\top X)\mathbf{v} = \sum_{\alpha=1}^n v_\alpha \sigma(\mathbf{w}^\top \mathbf{x}_\alpha).$$

Then, letting $\mathbf{u} \in \mathbb{R}^n$ be the vector with entries $u_\alpha = v_\alpha \sigma'(\mathbf{w}^\top \mathbf{x}_\alpha)$,

$$\nabla F(\mathbf{w}) = \sum_{\alpha=1}^n v_\alpha \sigma'(\mathbf{w}^\top \mathbf{x}_\alpha) \cdot \mathbf{x}_\alpha = X\mathbf{u}, \qquad \|\nabla F(\mathbf{w})\| \leq \|X\| \cdot \|\mathbf{u}\| \leq \lambda_\sigma B. \tag{34}$$

Then by the Gaussian Poincaré inequality, $\mathrm{Var}[F(\mathbf{w})] \leq \mathbb{E}[\|\nabla F(\mathbf{w})\|^2] \leq \lambda_\sigma^2 B^2$, so $\|\Sigma\| \leq \lambda_\sigma^2 B^2$. In addition, by (27), the difference between $\Phi$ and $\Sigma$ is a rank-one perturbation controlled by

$$\|\Phi - \Sigma\| = \|\mathbb{E}[\sigma(\mathbf{w}^\top X)]\|^2 = \sum_{\alpha=1}^n \mathbb{E}[\sigma(\mathbf{w}^\top \mathbf{x}_\alpha)]^2 \leq C\lambda_\sigma^2 \sum_{\alpha=1}^n (\|\mathbf{x}_\alpha\|^2 - 1)^2 \leq C\lambda_\sigma^2 B^2, \tag{35}$$

the last inequality using the final condition of $(\varepsilon, B)$-orthonormality in Definition 3.1. This establishes part (a).

For part (b), we apply the concentration result of [54, Eq. (5.26)] for matrices with independent sub-Gaussian rows. For any fixed unit vector $\mathbf{v} \in \mathbb{R}^n$, recall from (34) that $F(\mathbf{w}) = \sigma(\mathbf{w}^\top X)\mathbf{v}$ is $\lambda_\sigma B$-Lipschitz. Then by Gaussian concentration-of-measure,

$$\|F(\mathbf{w}) - \mathbb{E}[F(\mathbf{w})]\|_{\psi_2} \leq C\lambda_\sigma B.$$

We have $|\mathbb{E}[F(\mathbf{w})]| \leq \|\mathbb{E}[\sigma(\mathbf{w}^\top X)]\| \leq C\lambda_\sigma B$ by (35), so also $\|F(\mathbf{w})\|_{\psi_2} \leq C\lambda_\sigma B$. This holds for any unit vector $\mathbf{v} \in \mathbb{R}^n$, hence $\|\sigma(\mathbf{w}^\top X)\|_{\psi_2} \leq C\lambda_\sigma B$ for the vector sub-Gaussian norm. Thus, $\sqrt{d}\breve{X}/(\lambda_\sigma B)$ has i.i.d. rows whose sub-Gaussian norm is at most a universal constant. Recalling $\Phi = \mathbb{E}[\breve{X}^\top \breve{X}]$ and applying [54, Eq. (5.26)] with $A = \sqrt{d}\breve{X}/(\lambda_\sigma B)$, we obtain for some universal constants $C, c > 0$ that

$$\mathbb{P}\left[\|\breve{X}^\top \breve{X} - \Phi\| > \max(\delta, \delta^2)\|\Phi\|\right] \leq 2e^{-ct^2}, \qquad \delta = C\sqrt{n/\breve{d}} + t/\sqrt{\breve{d}}.$$

Note that the complementary event $\|\breve{X}^\top \breve{X} - \Phi\| \leq \max(\delta, \delta^2)\|\Phi\|$ implies

$$\|\breve{X}\| \leq \sqrt{(1 + \max(\delta, \delta^2))\|\Phi\|} \leq (1 + C'\delta)\sqrt{\|\Phi\|}$$

for a constant $C' > 0$. Then choosing $t = \sqrt{n}$ and applying part (a) yields part (b). $\qquad\square$

**Lemma D.5.** *If $X$ is $(\varepsilon, B)$-orthonormal where $\varepsilon < 1/\lambda_\sigma$, then for universal constants $C, c > 0$, with probability at least $1 - e^{-cn}$, the columns of $\breve{X}$ satisfy*

$$\sum_{\alpha=1}^n (\|\breve{\mathbf{x}}_\alpha\|^2 - 1)^2 \leq C\left(1 + n^2/\breve{d}^2\right)\lambda_\sigma^4 B^2.$$

Let us remark that in settings where $\varepsilon \gg 1/\sqrt{n}$, applying Lemma D.3(b) to bound each term $(\|\breve{\mathbf{x}}_\alpha\|^2 - 1)^2$ separately would not yield a constant-order bound for this sum. The proof below performs a more careful analysis of the combined fluctuations of $(\|\breve{\mathbf{x}}_\alpha\|^2 - 1)^2$.

*Proof.* Let $\mathbf{z} = (z_1, \ldots, z_n) \in \mathbb{R}^n$ and $\mathbf{r} = (r_1, \ldots, r_n) \in \mathbb{R}^n$ be defined as

$$z_\alpha = \|\breve{\mathbf{x}}_\alpha\|^2 - \mathbb{E}[\|\breve{\mathbf{x}}_\alpha\|^2], \qquad r_\alpha = \mathbb{E}[\|\breve{\mathbf{x}}_\alpha\|^2] - 1.$$

The quantity to be bounded is $\|\mathbf{z} + \mathbf{r}\|^2$. Note that $\|\mathbf{z} + \mathbf{r}\|^2 \leq 2\|\mathbf{z}\|^2 + 2\|\mathbf{r}\|^2$. We have

$$\mathbb{E}[\|\breve{\mathbf{x}}_\alpha\|^2] = \mathbb{E}\left[\frac{1}{\breve{d}}\sum_{i=1}^{\breve{d}} \sigma(\mathbf{w}_i^\top \mathbf{x}_\alpha)^2\right] = \Phi_{\alpha\alpha},$$

so applying (28) from Lemma D.3,

$$\|\mathbf{r}\|^2 = \sum_{\alpha=1}^n (\Phi_{\alpha\alpha} - 1)^2 \leq C\lambda_\sigma^2 \sum_{\alpha=1}^n (\|\mathbf{x}_\alpha\|^2 - 1)^2 \leq C\lambda_\sigma^2 B^2. \tag{36}$$

Thus it remains to bound $\|\mathbf{z}\|^2$.

Let $\mathcal{N}$ be a $1/2$-net of the unit ball $\{\mathbf{w} \in \mathbb{R}^n : \|\mathbf{w}\| = 1\}$, of cardinality $|\mathcal{N}| \leq 5^n$. Then

$$\|\mathbf{z}\| = \sup_{\mathbf{w}:\|\mathbf{w}\|\leq 1} \mathbf{w}^\top \mathbf{z} \leq \sup_{\mathbf{v}\in\mathcal{N}} \mathbf{v}^\top \mathbf{z} + \|\mathbf{z}\|/2,$$

so $\|\mathbf{z}\| \leq 2\sup_{\mathbf{v}\in\mathcal{N}} \mathbf{v}^\top \mathbf{z}$. For each fixed vector $\mathbf{v} = (v_1, \ldots, v_n) \in \mathcal{N}$, we have

$$\mathbf{v}^\top \mathbf{z} = \sum_{\alpha=1}^n v_\alpha \cdot \frac{1}{\breve{d}}\sum_{i=1}^{\breve{d}} \left(\sigma(\mathbf{w}_i^\top \mathbf{x}_\alpha)^2 - \mathbb{E}[\sigma(\mathbf{w}_i^\top \mathbf{x}_\alpha)^2]\right)$$

$$= \frac{1}{\breve{d}}\sum_{i=1}^{\breve{d}} \left(\sum_{\alpha=1}^n \left(\sigma(\mathbf{w}_i^\top \mathbf{x}_\alpha)^2 - \mathbb{E}[\sigma(\mathbf{w}_i^\top \mathbf{x}_\alpha)^2]\right)v_\alpha\right). \tag{37}$$

We will bound the sub-exponential norm of each summand $i = 1, \ldots, \breve{d}$ and apply Bernstein's inequality.

For $\mathbf{w} \sim \mathcal{N}(0, \mathrm{Id})$, denote

$$\mathbf{q} \equiv \mathbf{q}(\mathbf{w}) = (q_1, \ldots, q_n) = (\mathbf{w}^\top \mathbf{x}_1, \ldots, \mathbf{w}^\top \mathbf{x}_n), \qquad F(\mathbf{q}) = \sum_{\alpha=1}^n \left(\sigma(q_\alpha)^2 - \mathbb{E}[\sigma(q_\alpha)^2]\right)v_\alpha.$$

Observe that $\mathbf{q}(\mathbf{w}) = X^\top \mathbf{w}$. Thus we wish to bound the sub-exponential norm of $F(\mathbf{q}(\mathbf{w}))$ when $\mathbf{w} \sim \mathcal{N}(0, \mathrm{Id})$. By the Gaussian Sobolev inequality (see [2, Eq. (3)]), for any $p \geq 2$,

$$\|F(\mathbf{q}(\mathbf{w}))\|_{L^p} \leq \sqrt{p} \cdot \left\| \|\nabla_{\mathbf{w}} F(\mathbf{q}(\mathbf{w}))\| \right\|_{L^p} \tag{38}$$

where $\|Y\|_{L^p} = \mathbb{E}[|Y|^p]^{1/p}$ denotes the $L^p$-norm of a random variable (and $\|\nabla_{\mathbf{w}} F(\mathbf{q}(\mathbf{w}))\|$ is the usual $\ell_2$ vector norm of the gradient of $F(\mathbf{q}(\mathbf{w}))$ in $\mathbf{w}$). By the chain rule,

$$\nabla_{\mathbf{w}} F(\mathbf{q}(\mathbf{w})) = X \cdot \nabla_{\mathbf{q}} F(\mathbf{q}),$$

so

$$\|\nabla_{\mathbf{w}} F(\mathbf{q}(\mathbf{w}))\|^2 \leq \|X\|^2 \|\nabla_{\mathbf{q}} F(\mathbf{q})\|^2 \leq B^2 \|\nabla_{\mathbf{q}} F(\mathbf{q})\|^2.$$

We have $(\partial/\partial q_\alpha) F(\mathbf{q}) = 2\sigma(q_\alpha)\sigma'(q_\alpha)v_\alpha$, so

$$\|\nabla_{\mathbf{q}} F(\mathbf{q})\|^2 = \sum_{\alpha=1}^{n} 4\sigma(q_\alpha)^2 \sigma'(q_\alpha)^2 v_\alpha^2 \leq 4\lambda_\sigma^2 \sum_{\alpha=1}^{n} \sigma(q_\alpha)^2 v_\alpha^2.$$

Recalling (31), we have $\|\sigma(q_\alpha)^2\|_{\psi_1} = \|\sigma(\mathbf{w}^\top \mathbf{x}_\alpha)^2\|_{\psi_1} \leq C\lambda_\sigma^2$. Then

$$\left\| \sum_{\alpha=1}^{n} \sigma(q_\alpha)^2 v_\alpha^2 \right\|_{\psi_1} \leq C\lambda_\sigma^2 \sum_{\alpha=1}^{n} v_\alpha^2 = C\lambda_\sigma^2,$$

so

$$\left\| \|\nabla_{\mathbf{w}} F(\mathbf{q}(\mathbf{w}))\|^2 \right\|_{\psi_1} \leq C\lambda_\sigma^4 B^2.$$

This implies the bound (see [53, Proposition 2.7.1]), for any $p \geq 1$,

$$\left\| \|\nabla_{\mathbf{w}} F(\mathbf{q}(\mathbf{w}))\| \right\|_{L^{2p}}^{2p} = \mathbb{E}\left[ \|\nabla_{\mathbf{w}} F(\mathbf{q}(\mathbf{w}))\|^{2p} \right] = \left\| \|\nabla_{\mathbf{w}} F(\mathbf{q}(\mathbf{w}))\|^2 \right\|_{L^p}^p \leq (C'\lambda_\sigma^4 B^2 \cdot p)^p$$

for a universal constant $C' > 0$. Thus, applying this to (38), we obtain for any $p \geq 2$

$$\|F(\mathbf{q}(\mathbf{w}))\|_{L^p} \leq \sqrt{p} \cdot C\lambda_\sigma^2 B\sqrt{p} = C\lambda_\sigma^2 B \cdot p.$$

Finally, this implies (see again [53, Proposition 2.7.1]) $\|F(\mathbf{q}(\mathbf{w}))\|_{\psi_1} \leq C'\lambda_\sigma^2 B$ for a universal constant $C' > 0$, which is our desired bound on the sub-exponential norm of $F(\mathbf{q}(\mathbf{w}))$.

Applying this and Bernstein's inequality to (37), for any $t > 0$,

$$\mathbb{P}[\mathbf{v}^\top \mathbf{z} > t] \leq \exp\left( -c\check{d} \min\left( \frac{t^2}{\lambda_\sigma^4 B^2}, \frac{t}{\lambda_\sigma^2 B} \right) \right).$$

Setting

$$t = C_0 \lambda_\sigma^2 B \cdot \max(\delta, \delta^2), \qquad \delta = \sqrt{n/\check{d}}$$

for a large enough constant $C_0 > 0$, and taking the union bound over all $5^n$ vectors $\mathbf{v} \in \mathcal{N}$, we get

$$\mathbb{P}[\|\mathbf{z}\| > 2t] \leq \mathbb{P}\left[ \sup_{\mathbf{v} \in \mathcal{N}} \mathbf{v}^\top \mathbf{z} > t \right] \leq e^{-cn}$$

for a constant $c > 0$. Combining with the bound on $\|\mathbf{r}\|^2$ in (36), we obtain the lemma. $\qquad\square$

## E  Resolvent analysis for a single layer

We consider the same setting of a single layer as in the preceding section. Let $\check{X}$ and $\Phi$ be defined by the deterministic input $X \in \mathbb{R}^{d \times n}$ and Gaussian matrix $W \in \mathbb{R}^{\check{d} \times d}$ as in (24) and (25), and define the ($n$-dependent) aspect ratio

$$\gamma = n/\check{d}.$$

Consider a deterministic real-symmetric matrix $A \in \mathbb{R}^{n \times n}$, and two (possibly $n$-dependent) spectral arguments $\alpha \in \mathbb{C}^*$ and $z \in \mathbb{C}^+$, where $\mathbb{C}^* = \overline{\mathbb{C}^-} \setminus \{0\}$. We study the matrix

$$A + \alpha \check{X}^\top \check{X} - z\,\mathrm{Id}\,.$$

We collect here the set of assumptions that we will use in this section.

**Assumption E.1.** There are constants $B, C_0, c_0 > 0$ such that

(a) $\alpha \in \mathbb{C}^*$ and $z \in \mathbb{C}^+$, and $\gamma, |\alpha|, |z|, \operatorname{Im} z \in [c_0, C_0]$.

(b) $X$ is $(\varepsilon_n, B)$-orthonormal, where $\varepsilon_n < n^{-0.01}$.

(c) $A \in \mathbb{R}^{n \times n}$ is deterministic and symmetric, satisfying $\|A\| \leq C_0$.

(d) $W$ has i.i.d. $\mathcal{N}(0, 1)$ entries, and $\sigma(x)$ satisfies Assumption 3.2(b).

Throughout this section, $C, C', c, c', n_0 > 0$ denote constants changing from instance to instance that may depend on $\lambda_\sigma$ and the above values $B, C_0, c_0$.

Proposition C.2 ensures that $A + \alpha \breve{X}^\top \breve{X} - z \operatorname{Id}$ is invertible. Define the resolvent

$$R = (A + \alpha \breve{X}^\top \breve{X} - z \operatorname{Id})^{-1} \in \mathbb{C}^{n \times n} \tag{39}$$

and the deterministic ($n$-dependent) parameter

$$\bar{s} = \alpha^{-1} + \gamma \cdot \mathbb{E}[\operatorname{tr} R\Phi]. \tag{40}$$

The goal of this section is to prove the following result, which approximates this resolvent $R$ by replacing the random matrix $\alpha \breve{X}^\top \breve{X}$ with a deterministic matrix $\bar{s}^{-1}\Phi$, and provides an approximate fixed-point equation that defines this parameter $\bar{s}$.

For $A = 0$ and $\alpha = 1$, we will verify in Appendix F that this result reduces to the Marcenko-Pastur equation (6).

**Lemma E.2.** *Under Assumption E.1, there are constants $C, c, c', n_0 > 0$ such that for all $n \geq n_0$, any deterministic matrix $M \in \mathbb{C}^{n \times n}$, and any $t \in (n^{-1}, c')$,*

*(a)* $\mathbb{P}\left[\left|\operatorname{tr} RM - \operatorname{tr}\left(A + \bar{s}^{-1}\Phi - z\operatorname{Id}\right)^{-1}M\right| > \|M\|t\right] \leq Cne^{-cnt^2}$

*(b)* $\mathbb{P}\left[\left|\bar{s} - \left(\alpha^{-1} + \gamma \operatorname{tr}\left(A + \bar{s}^{-1}\Phi - z\operatorname{Id}\right)^{-1}\Phi\right)\right| > t\right] \leq Cne^{-cnt^2}$

## E.1 Basic bounds

**Proposition E.3.** *Under Assumption E.1, deterministically for some constants $C, c, n_0 > 0$ and all $n \geq n_0$,*

$$\|R\| \leq C, \qquad \|\Phi\| \leq C, \qquad |\bar{s}| \leq C, \qquad \operatorname{Im}\bar{s} \geq c.$$

*Furthermore, with probability at least $1 - 2e^{-c'n}$ for a constant $c' > 0$,*

$$\operatorname{Im}\operatorname{tr} R\Phi \geq c.$$

*Proof.* We may write $A + \alpha \breve{X}^\top \breve{X} - z\operatorname{Id} = U + iV$ where $U = A + (\operatorname{Re}\alpha)\breve{X}^\top \breve{X} - (\operatorname{Re}z)\operatorname{Id}$ and $V = (\operatorname{Im}\alpha)\breve{X}^\top \breve{X}^\top - (\operatorname{Im}z)\operatorname{Id}$. Both $U$ and $V$ are symmetric, and $V \preceq (-\operatorname{Im}z)\operatorname{Id}$ because $\operatorname{Im}\alpha \leq 0$ and $\operatorname{Im}z > 0$. Then $\|R\| \leq 1/\operatorname{Im}z \leq C$ by Proposition C.2.

The bound $\|\Phi\| \leq C$ comes from Lemma D.4(a) and the $(\varepsilon_n, B)$-orthonormality assumption for $X$. Then from the definition of $\bar{s}$ in (40) and the bounds $\|R\|, \|\Phi\| \leq C$, we have also $|\bar{s}| \leq C$. For the lower bound for $\operatorname{Im}\bar{s}$ and $\operatorname{Im}\operatorname{tr} R\Phi$, let us write

$$\operatorname{tr} R\Phi = \operatorname{tr}\left(\frac{R + R^*}{2}\right)\Phi + \operatorname{tr}\left(\frac{R - R^*}{2}\right)\Phi.$$

The first trace is real because $R + R^*$ is Hermitian, so

$$\operatorname{Im}\operatorname{tr} R\Phi = \operatorname{Im}\operatorname{tr}\left(\frac{R - R^*}{2}\right)\Phi.$$

Denoting $Y = A + \alpha \breve{X}^\top \breve{X} - z\operatorname{Id}$ and applying the identity $A^{-1} - B^{-1} = A^{-1}(B - A)B^{-1}$, we have

$$R - R^* = Y^{-1} - (Y^*)^{-1} = Y^{-1}(Y^* - Y)(Y^*)^{-1} = R(Y^* - Y)R^*.$$

Then, writing $Y = U + iV$ as above and applying $Y^* - Y = -2iV$, we get

$$\operatorname{Im} \operatorname{tr} R\Phi = \operatorname{Im}(-i \cdot \operatorname{tr} RVR^*\Phi)$$
$$= \operatorname{Re}\left(-(\operatorname{Im}\alpha) \cdot \operatorname{tr} R\check{X}^\top \check{X} R^*\Phi + (\operatorname{Im} z) \cdot \operatorname{tr} RR^*\Phi\right).$$

Since $\operatorname{tr} R\check{X}^\top \check{X} R^*\Phi = \operatorname{tr} \Phi^{1/2} R\check{X}^\top \check{X} R^*\Phi^{1/2}$, where this matrix is positive semi-definite, this trace is real and non-negative. Similarly, $\operatorname{tr} RR^*\Phi$ is real and non-negative. Then the above yields the lower bound

$$\operatorname{Im} \operatorname{tr} R\Phi \geq \operatorname{Im} z \cdot \operatorname{tr} RR^*\Phi \geq \operatorname{Im} z \cdot \lambda_{\min}(RR^*) \cdot \operatorname{tr} \Phi,$$

where $\lambda_{\min}(RR^*)$ is the smallest eigenvalue of $RR^*$. By (28) and the condition $\varepsilon_n < n^{-0.01}$, we have $\operatorname{tr} \Phi \geq c$ for a constant $c > 0$ and large enough $n_0$. Observe that $\lambda_{\min}(RR^*) = 1/\|Y\|^2$, and $\|Y\| \leq \|A\| + |\alpha| \cdot \|\check{X}\|^2 + |z|$. By Lemma D.4(b), with probability $1 - 2e^{-c'n}$, we have $\|\check{X}\| \leq C$, so putting this together yields $\operatorname{Im} \operatorname{tr} R\Phi \geq c$ with this probability. Finally, for the deterministic bound $\operatorname{Im} \bar{s} \geq c$, we may apply $\operatorname{Im} \operatorname{tr} R\Phi \geq c$ on the event where $\|\check{X}\| \leq C$ holds, and $\operatorname{Im} \operatorname{tr} R\Phi \geq 0$ on the complementary event. Taking an expectation and applying the definition (40) yields $\operatorname{Im} \bar{s} \geq c$.  □

### E.2 Resolvent approximation

We recall the result of [39, Lemma 1], which establishes concentration of quadratic forms in the rows of $\check{X}$. The following is its specialization to standard Gaussian matrices $W$, and stated in our notation.

**Lemma E.4** ([39]). *Suppose $\sigma(x)$ is $\lambda_\sigma$-Lipschitz, and let $\check{\mathbf{x}}_i^\top$ be a row of $\check{X}$. Then for any deterministic matrix $Y \in \mathbb{R}^{n \times n}$ with $\|Y\| \leq 1$, for some constants $C, c > 0$ (depending on $\lambda_\sigma$), and for any $t > 0$,*

$$\mathbb{P}\left(\left|\frac{1}{\gamma}\check{\mathbf{x}}_i^\top Y \check{\mathbf{x}}_i - \operatorname{tr} Y\Phi\right| > t\right) \leq C\exp\left(-\frac{cn}{\|X\|^2}\min\left(\frac{t^2}{t_0^2}, t\right)\right) \tag{41}$$

*where $t_0 = |\sigma(0)| + \lambda_\sigma \|X\|\sqrt{1/\gamma}$.*

Using this result, we establish the following approximation for the resolvent $R$ in (39).

**Lemma E.5.** *Consider any deterministic matrix $M \in \mathbb{C}^{n \times n}$, and set*

$$\delta_n = \operatorname{tr} M - \operatorname{tr} R\left(A + \frac{1}{\alpha^{-1} + \gamma \operatorname{tr} R\Phi}\Phi - z\operatorname{Id}\right)M.$$

*Under Assumption E.1, there exist constants $C, c, c', n_0 > 0$ such that for all $n \geq n_0$ and $t \in (n^{-1}, c')$,*

$$\mathbb{P}[|\delta_n| > \|M\|t] \leq Cne^{-cnt^2}.$$

*Proof.* By rescaling $M$, we may assume that $\|M\| \leq 1$. We have $\operatorname{Id} = R(A + \alpha\check{X}^\top \check{X} - z\operatorname{Id}) = RA + \alpha R\check{X}^\top \check{X} - zR$. Writing $\check{X}^\top \check{X} = \sum_i \check{\mathbf{x}}_i\check{\mathbf{x}}_i^\top$ (where $\check{\mathbf{x}}_i^\top$ is the $i^{\text{th}}$ row of $\check{X}$), multiplying by $M$, and taking the normalized trace $\operatorname{tr} = n^{-1}\operatorname{Tr}$,

$$\operatorname{tr} M = \operatorname{tr} RAM + \alpha \operatorname{tr} R\check{X}^\top \check{X}M - z\operatorname{tr} RM$$

$$= \operatorname{tr} RAM + \frac{\alpha}{n}\sum_{i=1}^{\check{d}} \check{\mathbf{x}}_i^\top MR\check{\mathbf{x}}_i - z\operatorname{tr} RM.$$

Hence

$$\delta_n = \frac{\alpha}{n}\sum_{i=1}^{\check{d}} \check{\mathbf{x}}_i^\top MR\check{\mathbf{x}}_i - \frac{\operatorname{tr} R\Phi M}{\alpha^{-1} + \gamma\operatorname{tr} R\Phi}.$$

Let us define the leave-one-out resolvent, for each $1 \leq i \leq \check{d}$,

$$R^{(i)} = \left(A + \alpha\sum_{j:j\neq i}\check{\mathbf{x}}_j\check{\mathbf{x}}_j^\top - z\operatorname{Id}\right)^{-1}.$$

We may then decompose $\delta_n$ as $\delta_n = J_1 + \gamma J_2$ where (recalling $\gamma = n/\check{d}$)

$$J_1 = \frac{1}{n} \sum_{i=1}^{\check{d}} \left( \alpha \check{\mathbf{x}}_i^\top M R \check{\mathbf{x}}_i - \frac{\gamma \operatorname{tr} R^{(i)} \Phi M}{\alpha^{-1} + \gamma \operatorname{tr} R^{(i)} \Phi} \right),$$

$$J_2 = \frac{1}{n} \sum_{i=1}^{\check{d}} \left( \frac{\operatorname{tr} R^{(i)} \Phi M}{\alpha^{-1} + \gamma \operatorname{tr} R^{(i)} \Phi} - \frac{\operatorname{tr} R \Phi M}{\alpha^{-1} + \gamma \operatorname{tr} R \Phi} \right).$$

Let us denote these summands as

$$J_1^{(i)} = \alpha \check{\mathbf{x}}_i^\top M R \check{\mathbf{x}}_i - \frac{\gamma \operatorname{tr} R^{(i)} \Phi M}{\alpha^{-1} + \gamma \operatorname{tr} R^{(i)} \Phi} \quad \text{and} \quad J_2^{(i)} = \frac{\operatorname{tr} R^{(i)} \Phi M}{\alpha^{-1} + \gamma \operatorname{tr} R^{(i)} \Phi} - \frac{\operatorname{tr} R \Phi M}{\alpha^{-1} + \gamma \operatorname{tr} R \Phi}.$$

**Bound for $J_1$.** Momentarily fix the index $i \in \{1, \ldots, \check{d}\}$. Applying the Sherman-Morrison identity, we have

$$R = R^{(i)} - \frac{\alpha R^{(i)} \check{\mathbf{x}}_i \check{\mathbf{x}}_i^\top R^{(i)}}{1 + \alpha \check{\mathbf{x}}_i^\top R^{(i)} \check{\mathbf{x}}_i}. \tag{42}$$

Then, introducing $A_1 = \check{\mathbf{x}}_i^\top M R^{(i)} \check{\mathbf{x}}_i$ and $A_2 = \check{\mathbf{x}}_i^\top R^{(i)} \check{\mathbf{x}}_i$,

$$\alpha \check{\mathbf{x}}_i^\top M R \check{\mathbf{x}}_i = \alpha A_1 - \frac{\alpha^2 A_1 A_2}{1 + \alpha A_2} = \frac{A_1}{\alpha^{-1} + A_2}.$$

Recall that the rows of $\check{X}$ are i.i.d. Let $\check{X}^{(i)}$ be the matrix $\check{X}$ with the $i^{\text{th}}$ row $\check{\mathbf{x}}_i$ removed, and let $\mathbb{E}_{\check{\mathbf{x}}_i}[\cdot]$ be the expectation over only $\check{\mathbf{x}}_i$ (i.e. conditional on $\check{X}^{(i)}$). Observe that $R^{(i)}$ is a function of $\check{X}^{(i)}$. Applying Proposition E.3 with $\check{X}^{(i)}$ in place of $\check{X}$, we see that $\|R^{(i)}\|$ and $\|M R^{(i)}\|$ are both bounded by a constant. Then applying Lemma E.4 conditional on $\check{X}^{(i)}$, and recalling the bound (22) for $\sigma(0)$, there are constants $C, c > 0$ for which

$$\mathbb{P}[|A_k - \mathbb{E}_{\check{\mathbf{x}}_i}[A_k]| > t] \leq C e^{-cn \min(t^2, t)} \qquad \text{for } k = 1, 2.$$

Note that

$$\mathbb{E}_{\check{\mathbf{x}}_i}[A_1] = \operatorname{Tr} M R^{(i)} \mathbb{E}[\check{\mathbf{x}}_i \check{\mathbf{x}}_i^\top] = \frac{1}{\check{d}} \operatorname{Tr} M R^{(i)} \Phi = \gamma \operatorname{tr} R^{(i)} \Phi M.$$

Similarly, $\mathbb{E}_{\check{\mathbf{x}}_i}[A_2] = \gamma \operatorname{tr} R^{(i)} \Phi$, so

$$J_1^{(i)} = \frac{A_1}{\alpha^{-1} + A_2} - \frac{\mathbb{E}_{\check{\mathbf{x}}_i}[A_1]}{\alpha^{-1} + \mathbb{E}_{\check{\mathbf{x}}_i}[A_2]}.$$

Applying Proposition E.3, we have for some constants $C, c, c' > 0$, on an event $\mathcal{E}(\check{X}^{(i)})$ of probability $1 - 2e^{-c'n}$, that

$$|\mathbb{E}_{\check{\mathbf{x}}_i}[A_1]| \leq C, \qquad |\alpha^{-1} + \mathbb{E}_{\check{\mathbf{x}}_i}[A_2]| \geq \operatorname{Im}(\alpha^{-1} + \mathbb{E}_{\check{\mathbf{x}}_i}[A_2]) \geq c.$$

Then, for any $t$ such that $t < c/2$, on the event where $|A_1 - \mathbb{E}_{\check{\mathbf{x}}_i}[A_1]| \leq t$, $|A_2 - \mathbb{E}_{\check{\mathbf{x}}_i}[A_2]| \leq t$, and $\mathcal{E}(\check{X}^{(i)})$ all hold,

$$\left| J_1^{(i)} \right| \leq \frac{|A_1 - \mathbb{E}_{\check{\mathbf{x}}_i}[A_1]|}{|\alpha^{-1} + A_2|} + |\mathbb{E}_{\check{\mathbf{x}}_i}[A_1]| \cdot \frac{|A_2 - \mathbb{E}_{\check{\mathbf{x}}_i}[A_2]|}{|\alpha^{-1} + A_2| \cdot |\alpha^{-1} + \mathbb{E}_{\check{\mathbf{x}}_i}[A_2]|} \leq Ct. \tag{43}$$

Thus, for $t < c'$ and a sufficiently small constant $c' > 0$, we have $\mathbb{P}[|J_1^{(i)}| \geq t] \leq C e^{-cnt^2}$. Applying a union bound over $i \in \{1, \ldots, \check{d}\}$, this yields $\mathbb{P}[|J_1| \geq t] \leq Cn e^{-cnt^2}$.

**Bound for $J_2$.** Applying the identity $A^{-1} - B^{-1} = A^{-1}(B - A)B^{-1}$,

$$R^{(i)} - R = R^{(i)}(R^{-1} - (R^{(i)})^{-1})R = \alpha R^{(i)} \check{\mathbf{x}}_i \check{\mathbf{x}}_i^\top R.$$

Then, applying also the bounds $\|R\|, \|R^{(i)}\| \leq C$ from Proposition E.3,

$$|\operatorname{tr}(R^{(i)} - R)\Phi M| = \frac{1}{n} |\alpha \check{\mathbf{x}}_i^\top R \Phi M R^{(i)} \check{\mathbf{x}}_i| \leq \frac{C \|\check{X}\|^2}{n}.$$

Applying Lemma D.4(b), with probability $1 - 2e^{-cn}$, this is at most $C/n$ for every $i \in \{1, \ldots, \check{d}\}$. Similarly, $|\operatorname{tr}(R^{(i)} - R)\Phi| \leq C/n$ with this probability. Applying again $|\operatorname{tr} R \Phi M| \leq C$, $|\alpha^{-1} + \gamma \operatorname{tr} R \Phi| \geq c$, and an argument similar to (43), we obtain $|J_2^{(i)}| \leq C'/n$ for a constant $C' > 0$. Taking a union bound over $i \in \{1, \ldots, \check{d}\}$, this yields $\mathbb{P}[|J_2| > C/n] \leq C'n e^{-cn}$. Combining these bounds for $J_1$ and $J_2$, choosing $t > cn^{-1}$, and re-adjusting the constants yields the lemma. $\qquad \square$

### E.3 Proof of Lemma E.2

We now prove Lemma E.2 using Lemma E.5. Define the random $n$-dependent parameter

$$s = \alpha^{-1} + \gamma \operatorname{tr} R\Phi,$$

so that $\bar{s} = \mathbb{E}[s]$. The following establishes concentration of $s$ around $\bar{s}$.

**Lemma E.6.** *Under Assumption E.1, for some constants $c, n_0 > 0$, all $n \geq n_0$, and any $t > 0$,*

$$\mathbb{P}\left[|s - \bar{s}| > t\right] \leq 2e^{-cnt^2}.$$

*Proof.* Define $F(W) = \gamma \operatorname{tr} R\Phi$, where $R$ and $\check{X}$ are considered as a function of $W$. Fix any matrices $W, \Delta \in \mathbb{R}^{\check{d} \times n}$ where $\|\Delta\|_F = 1$, and define $W_t = W + t\Delta$. Then, applying $\partial R = -R(\partial(R^{-1}))R$ and $R = R^\top$,

$$
\begin{aligned}
\operatorname{vec}(\Delta)^\top (\nabla F(W)) = \frac{d}{dt}\Big|_{t=0} F(W_t) &= -\gamma \operatorname{tr} R\left(\frac{d}{dt}\Big|_{t=0} R^{-1}\right) R\Phi \\
&= -2\gamma\alpha \operatorname{tr} R\left(\check{X}^\top \cdot \frac{d}{dt}\Big|_{t=0} \check{X}\right) R\Phi \\
&= -\frac{2\gamma\alpha}{\sqrt{\check{d}}} \operatorname{tr} R\left(\check{X}^\top \cdot (\sigma'(WX) \odot (\Delta X))\right) R\Phi,
\end{aligned}
$$

where $\odot$ is the Hadamard product, and $\sigma'$ is applied entrywise. Applying Proposition E.3,

$$\left|\operatorname{vec}(\Delta)^\top(\nabla F(W))\right| \leq \frac{C}{\sqrt{\check{d}}} \cdot \left\|R\check{X}^\top \cdot (\sigma'(WX) \odot (\Delta X)) \cdot R\right\| \leq \frac{C'}{\sqrt{\check{d}}} \cdot \|R\check{X}^\top\| \cdot \|\sigma'(WX) \odot (\Delta X)\|.$$

For the first term,

$$\|R\check{X}^\top\|^2 = \frac{1}{|\alpha|}\|R(\alpha\check{X}^\top\check{X})R^*\| \leq \frac{1}{|\alpha|}\left(\|R(A + \alpha\check{X}^\top\check{X} - z\operatorname{Id})R^*\| + \|R(A - z\operatorname{Id})R^*\|\right)$$

$$\leq \frac{1}{|\alpha|}(\|R\| + \|R\|^2(\|A\| + |z|)) \leq C.$$

For the second term,

$$\|\sigma'(WX) \odot (\Delta X)\| \leq \|\sigma'(WX) \odot (\Delta X)\|_F \leq \lambda_\sigma\|\Delta X\|_F \leq \lambda_\sigma\|\Delta\|_F \cdot \|X\| \leq C.$$

Thus $|\operatorname{vec}(\Delta)^\top(\nabla F(W))| \leq C/\sqrt{n}$. This holds for every $\Delta$ such that $\|\Delta\|_F = 1$, so $F(W)$ is $C/\sqrt{n}$-Lipschitz in $W$ with respect to the Frobenius norm. Then the result follows from Gaussian concentration of measure. $\qquad\square$

To conclude the proof of Lemma E.2, we may again assume $\|M\| \leq 1$ by rescaling $M$. Set

$$\widetilde{M} = \left(A + \bar{s}^{-1}\Phi - z\operatorname{Id}\right)^{-1} M.$$

Note that $\bar{s}^{-1} \in \mathbb{C}^-$, so $\|\widetilde{M}\| \leq \|(A + \bar{s}^{-1}\Phi - z\operatorname{Id})^{-1}\| \leq C$ by Proposition C.2. Applying Lemma E.5 with $\widetilde{M}$,

$$\mathbb{P}\left[\left|\operatorname{tr}\widetilde{M} - \operatorname{tr} R\left(A + s^{-1}\Phi - z\operatorname{Id}\right)\widetilde{M}\right| > t\right] \leq Cne^{-cnt^2} \tag{44}$$

for all $t \in (n^{-1}, c')$. Furthermore, applying the definition of $\widetilde{M}$,

$$
\begin{aligned}
\left|\operatorname{tr} R\left(A + s^{-1}\Phi - z\operatorname{Id}\right)\widetilde{M} - \operatorname{tr} RM\right| &= \left|\operatorname{tr} R\left((A + s^{-1}\Phi - z\operatorname{Id}) - (A + \bar{s}^{-1}\Phi - z\operatorname{Id})\right)\widetilde{M}\right| \\
&= |s^{-1} - \bar{s}^{-1}| \cdot |\operatorname{tr} R\Phi\widetilde{M}| \leq C|s^{-1} - \bar{s}^{-1}|.
\end{aligned}
$$

Recall that $|\bar{s}| \geq \operatorname{Im}\bar{s} \geq c$. Then, on the event where $|s - \bar{s}| \leq t$ and $t < c/2$, we have $|s^{-1} - \bar{s}^{-1}| \leq Ct$. Then applying Lemma E.6, for some constants $c, c' > 0$ and all $t \in (0, c')$,

$$\mathbb{P}\left[|\operatorname{tr} R\left(A + s^{-1}\Phi - z\operatorname{Id}\right)\widetilde{M} - \operatorname{tr} RM| > t\right] \leq 2e^{-cnt^2}.$$

Combining this with (44) yields Lemma E.2(a). Specializing Lemma E.2(a) to $M = \Phi$, we obtain

$$\mathbb{P}\left[|s - \left(\alpha^{-1} + \gamma\operatorname{tr}(A + \bar{s}^{-1}\Phi - z\operatorname{Id})^{-1}\Phi\right)| > t\right] \leq Cne^{-cnt^2}.$$

Applying again Lemma E.6 to bound $|s - \bar{s}|$, we obtain Lemma E.2(b).

# F    Analysis for the Conjugate Kernel

Theorem 3.4 is a special case of Theorem 3.7, but let us provide here a simpler argument. Define, for each layer, the $n \times n$ matrices

$$\Phi_\ell = \mathbb{E}_{\mathbf{w}}\left[\sigma(\mathbf{w}^\top X_{\ell-1})^\top \sigma(\mathbf{w}^\top X_{\ell-1})\right] \tag{45}$$

$$\tilde{\Phi}_\ell = b_\sigma^2 X_{\ell-1}^\top X_{\ell-1} + (1 - b_\sigma^2)\,\mathrm{Id} \tag{46}$$

where $\mathbb{E}_{\mathbf{w}}$ denotes the expectation over only the random vector $\mathbf{w} \sim \mathcal{N}(0, \mathrm{Id})$. Here, $\Phi_\ell$ and $\tilde{\Phi}_\ell$ are deterministic conditional on $X_{\ell-1}$, but are random unconditionally for $\ell \geq 2$. For each fixed $\ell = 1, \ldots, L$, we will show

$$\lim \mathrm{spec}\, \Phi_\ell = \lim \mathrm{spec}\, \tilde{\Phi}_\ell. \tag{47}$$

Conditional on $X_{\ell-1}$, the spectral limit of $X_\ell^\top X_\ell$ was shown in [39] to be a Marcenko-Pastur map of the spectral limit of $\Phi_\ell$—we reproduce a short proof below under our assumptions, by specializing Lemma E.2 to $\alpha = 1$ and $A = 0$. Combining with (47) and iterating from $\ell = 1, \ldots, L$ yields Theorem 3.4.

**Lemma F.1.** *Under Assumption 3.2, for each $\ell = 1, \ldots, L$, almost surely as $n \to \infty$,*

$$\frac{1}{n}\|\Phi_\ell - \tilde{\Phi}_\ell\|_F^2 \to 0.$$

*Proof.* By Corollary D.2, increasing $(\varepsilon_n, B)$ as needed, we may assume that each matrix $X_0, \ldots, X_L$ is $(\varepsilon_n, B)$-orthonormal. Denote by $\Phi_\ell[\alpha, \beta]$ and $\tilde{\Phi}_\ell[\alpha, \beta]$ the $(\alpha, \beta)$ entries of these matrices. Then Lemma D.3(a) shows for $\alpha \neq \beta$ that

$$|\Phi_\ell[\alpha, \beta] - \tilde{\Phi}_\ell[\alpha, \beta]| \leq C\varepsilon_n^2.$$

For $\alpha = \beta$, applying $\tilde{\Phi}_\ell[\alpha, \alpha] = 1 - b_\sigma^2 + b_\sigma^2\|\mathbf{x}_\alpha^{\ell-1}\|^2$, we have

$$|\Phi_\ell[\alpha, \alpha] - \tilde{\Phi}_\ell[\alpha, \alpha]| \leq |\Phi_\ell[\alpha, \alpha] - 1| + b_\sigma^2|\|\mathbf{x}_\alpha^{\ell-1}\|^2 - 1| \leq C\varepsilon_n.$$

Then

$$\|\Phi_\ell - \tilde{\Phi}_\ell\|_F^2 \leq Cn(n-1)\varepsilon_n^4 + Cn\varepsilon_n^2,$$

and the result follows from the condition $\varepsilon_n n^{1/4} \to 0$. $\qquad\qquad\square$

*Proof of Theorem 3.4.* By Corollary D.2, we may assume that each matrix $X_0, \ldots, X_L$ is $(\varepsilon_n, B)$-orthonormal. This implies the bounds $\|X_\ell\| \leq C$ and $\|K^{\mathrm{CK}}\| \leq C$ for all large $n$.

For the spectral convergence, suppose by induction that $\lim \mathrm{spec}\, X_{\ell-1}^\top X_{\ell-1} = \mu_{\ell-1}$, where the base case $\lim \mathrm{spec}\, X_0^\top X_0 = \mu_0$ holds by assumption. Defining

$$\nu_\ell = (1 - b_\sigma^2) + b_\sigma^2 \cdot \mu_{\ell-1},$$

Proposition C.3 and Lemma F.1 together show that

$$\lim \mathrm{spec}\, \Phi_\ell = \lim \mathrm{spec}\, \tilde{\Phi}_\ell = \nu_\ell.$$

Specializing Lemma E.2(b) to the setting $A = 0$, $\alpha = 1$, $X = X_{\ell-1}$, and $\check{X} = X_\ell$, and choosing $t \equiv t_n$ such that $t_n \to 0$ and $nt_n^2 \gg \log n$, we obtain

$$\left|\bar{s} - 1 - (n/d_\ell)\,\mathrm{tr}(\bar{s}^{-1}\Phi_\ell - z\,\mathrm{Id})^{-1}\Phi_\ell\right| \to 0 \tag{48}$$

a.s. as $n \to \infty$, where

$$\bar{s} = 1 + \frac{n}{d_\ell}\mathbb{E}_{W_\ell}[\mathrm{tr}(X_\ell^\top X_\ell - z\,\mathrm{Id})^{-1}\Phi_\ell].$$

Here, this expectation is taken over only $W_\ell$ (i.e. conditional on $X_0, \ldots, X_{\ell-1}$).

Proposition E.3 verifies that $\bar{s}$ is bounded as $n \to \infty$, so for any subsequence in $n$, there is a further sub-subsequence along which $\bar{s} \to s_0$ for a limit $s_0 \equiv s_0(z) \in \mathbb{C}^+$. Applying $A^{-1} - B^{-1} = A^{-1}(B - A)B^{-1}$ and Propositions C.2 and E.3,

$$
\left| \operatorname{tr}(\bar{s}^{-1}\Phi_\ell - z \operatorname{Id})^{-1}\Phi_\ell - \operatorname{tr}(s_0^{-1}\Phi_\ell - z \operatorname{Id})^{-1}\Phi_\ell \right|
$$

$$
= |s_0^{-1} - s^{-1}| \cdot \operatorname{tr} \left| (s_0^{-1}\Phi_\ell - z \operatorname{Id})^{-1}\Phi_\ell(\bar{s}^{-1}\Phi_\ell - z \operatorname{Id})^{-1}\Phi_\ell \right|
$$

$$
\leq |s_0^{-1} - s^{-1}| \cdot \|(s_0^{-1}\Phi_\ell - z \operatorname{Id})^{-1}\| \cdot \|(\bar{s}^{-1}\Phi_\ell - z \operatorname{Id})^{-1}\| \cdot \|\Phi_\ell\|^2
$$

$$
\leq C|s_0^{-1} - s^{-1}|.
$$

Thus, along the sub-subsequence where $\bar{s} \to s_0$, we get

$$
\operatorname{tr}(\bar{s}^{-1}\Phi_\ell - z \operatorname{Id})^{-1}\Phi_\ell - \operatorname{tr}(s_0^{-1}\Phi_\ell - z \operatorname{Id})^{-1}\Phi_\ell \to 0. \tag{49}
$$

We have also

$$
\operatorname{tr}(s_0^{-1}\Phi_\ell - z \operatorname{Id})^{-1}\Phi_\ell \to \int \frac{x}{s_0^{-1}x - z} d\nu_\ell(x), \tag{50}
$$

since the function $x \mapsto x/(s_0^{-1}x - z)$ is continuous and bounded over $\mathbb{R}$, and $\lim \operatorname{spec} \Phi_\ell = \nu_\ell$. Thus, taking the limit of (48) along this sub-subsequence, the value $s_0$ must satisfy

$$
s_0 - 1 - \gamma_\ell \int \frac{x}{s_0^{-1}x - z} d\nu_\ell(x) = 0. \tag{51}
$$

Now applying Lemma E.2(a) with $M = \operatorname{Id}$, and taking the limit along this sub-subsequence, by a similar argument we obtain that

$$
\operatorname{tr}(X_\ell^\top X_\ell - z \operatorname{Id})^{-1} \to \int \frac{1}{s_0^{-1}x - z} d\nu_\ell(x). \tag{52}
$$

Denoting this limit by $m_\ell(z)$, and rewriting (51) by applying

$$
\int \frac{x}{s_0^{-1}x - z} d\nu_\ell(x) = s_0 \int \left( 1 + \frac{z}{s_0^{-1}x - z} \right) d\nu_\ell(x) = s_0(1 + zm_\ell(z)),
$$

we get $s_0^{-1} = 1 - \gamma_\ell - \gamma_\ell z m_\ell(z)$. Applying this back to the definition of $m_\ell(z)$ in (52), this shows that $m_\ell(z)$ satisfies the Marcenko-Pastur equation

$$
m(z) = \int \frac{1}{x(1 - \gamma_\ell - \gamma_\ell z m(z)) - z} d\nu_\ell(x),
$$

so $m_\ell(z)$ is the Stieltjes transform of $\mu_\ell = \rho_{\gamma\ell}^{\mathrm{MP}} \boxtimes \nu_\ell = \rho_{\gamma\ell}^{\mathrm{MP}} \boxtimes ((1 - b_\sigma^2) + b_\sigma^2 \cdot \mu_{\ell-1})$.

We have shown that $\operatorname{tr}(X_\ell^\top X_\ell - z \operatorname{Id})^{-1} \to m_\ell(z)$ almost surely along this sub-subsequence in $n$. Since, for every subsequence in $n$, there exists such a sub-subsequence, this implies $\lim_{n \to \infty} \operatorname{tr}(X_\ell^\top X_\ell - z \operatorname{Id})^{-1} = m_\ell(z)$ almost surely. Thus $\lim \operatorname{spec} X_\ell^\top X_\ell = \mu_\ell$, which completes the induction. $\qquad\square$

# G  Analysis for the Neural Tangent Kernel

## G.1  Spectral approximation and operator norm bound

We first prove the spectral approximation stated in Lemma 3.5, as well as the operator norm bound $\|K^{\mathrm{NTK}}\| \leq C$. The following form of $K^{\mathrm{NTK}}$ is derived also in [25, Eq. (1.7)]: Denote by $\mathbf{x}_\alpha^\ell$ the $\alpha^{\mathrm{th}}$ column of $X_\ell$. For each $\ell = 1, \ldots, L$, define the matrix $S_\ell \in \mathbb{R}^{d_\ell \times n}$ whose $\alpha^{\mathrm{th}}$ column is given by

$$
\mathbf{s}_\alpha^\ell = D_\alpha^\ell \frac{W_{\ell+1}^\top}{\sqrt{d_\ell}} D_\alpha^{\ell+1} \frac{W_{\ell+2}^\top}{\sqrt{d_{\ell+1}}} D_\alpha^{\ell+2} \cdots \frac{W_L^\top}{\sqrt{d_{L-1}}} D_\alpha^L \frac{\mathbf{w}}{\sqrt{d_L}}, \tag{53}
$$

where we define diagonal matrices indexed by $\alpha \in [n]$ and $k \in [L]$ as

$$
D_\alpha^k \equiv \operatorname{diag}\left( \sigma'(W_k \mathbf{x}_\alpha^{k-1}) \right) \in \mathbb{R}^{d_k \times d_k}.
$$

Applying the chain rule, we may verify for each input sample $\mathbf{x}_\alpha$ that

$$\nabla_\mathbf{w} f_\theta(\mathbf{x}_\alpha) = \mathbf{x}_\alpha^L \in \mathbb{R}^{d_L}, \quad \nabla_{W_\ell} f_\theta(\mathbf{x}_\alpha) = \mathbf{s}_\alpha^\ell \otimes \mathbf{x}_\alpha^{\ell-1} \in \mathbb{R}^{d_\ell d_{\ell-1}}.$$

Then

$$\left(\nabla_\mathbf{w} f_\theta(X)\right)^\top \left(\nabla_\mathbf{w} f_\theta(X)\right) = X_L^\top X_L,$$
$$\left(\nabla_{W_\ell} f_\theta(X)\right)^\top \left(\nabla_{W_\ell} f_\theta(X)\right) = (S_\ell^\top S_\ell) \odot (X_{\ell-1}^\top X_{\ell-1}),$$

where $\odot$ is the Hadamard product. Thus, the NTK is given by

$$K^{\text{NTK}} = \left(\nabla_\theta f_\theta(X)\right)^\top \left(\nabla_\theta f_\theta(X)\right) = X_L^\top X_L + \sum_{\ell=1}^{L}(S_\ell^\top S_\ell) \odot (X_{\ell-1}^\top X_{\ell-1}). \qquad (54)$$

**Lemma G.1.** *Let $X \in \mathbb{R}^{d \times n}$ be $(\varepsilon, B)$-orthonormal, let $W \in \mathbb{R}^{\check{d} \times d}$ have i.i.d. $\mathcal{N}(0,1)$ entries, and let $\mathbf{x}_\alpha, \mathbf{x}_\beta$ be two columns of $X$ where $\alpha \neq \beta$. Then for universal constants $C, c > 0$ and any $t > 0$:*

*(a) With probability at least $1 - 2e^{-c\check{d}t^2}$,*

$$\left| \frac{1}{\check{d}} \text{Tr} \left( \text{diag} \left(\sigma'(W\mathbf{x}_\alpha)\right) \text{diag} \left(\sigma'(W\mathbf{x}_\beta)\right) \right) - b_\sigma^2 \right| \leq C\lambda_\sigma^2 (\varepsilon + t).$$

*(b) Let $M \in \mathbb{R}^{d \times d}$ be any deterministic symmetric matrix, and denote*

$$T(\mathbf{x}_\alpha, \mathbf{x}_\beta) = \frac{1}{\check{d}} \text{Tr} \left( \text{diag} \left(\sigma'(W\mathbf{x}_\alpha)\right) W M W^\top \text{diag} \left(\sigma'(W\mathbf{x}_\beta)\right) \right).$$

*With probability at least $1 - (2\check{d} + 2)e^{-c\min(t^2\check{d}, t\sqrt{\check{d}})}$,*

$$\left| T(\mathbf{x}_\alpha, \mathbf{x}_\beta) - b_\sigma^2 \text{Tr} M \right| \leq C\lambda_\sigma^2 \left( \varepsilon\sqrt{d} + t\sqrt{d} + t\sqrt{\check{d}} \right) \|M\|_F.$$

*Furthermore, both (a) and (b) hold with $(\mathbf{x}_\alpha, \mathbf{x}_\alpha)$ in place of $(\mathbf{x}_\alpha, \mathbf{x}_\beta)$, upon replacing $b_\sigma^2$ by $a_\sigma$.*

*Proof.* Write $\mathbf{w}_k^\top \in \mathbb{R}^d$ for the $k^{\text{th}}$ row of $W$. Then

$$\frac{1}{\check{d}} \text{Tr} \left( \text{diag} \left(\sigma'(W\mathbf{x}_\alpha)\right) \text{diag} \left(\sigma'(W\mathbf{x}_\beta)\right) \right) = \frac{1}{\check{d}} \sum_{k=1}^{\check{d}} \sigma'(\mathbf{w}_k^\top \mathbf{x}_\alpha) \sigma'(\mathbf{w}_k^\top \mathbf{x}_\beta).$$

Applying $\sigma'(\mathbf{w}_k^\top \mathbf{x}_\alpha)\sigma'(\mathbf{w}_k^\top \mathbf{x}_\beta) \in [-\lambda_\sigma^2, \lambda_\sigma^2]$ and Hoeffding's inequality,

$$\mathbb{P}\left[ \left| \frac{1}{\check{d}} \sum_{k=1}^{\check{d}} \left( \sigma'(\mathbf{w}_k^\top \mathbf{x}_\alpha)\sigma'(\mathbf{w}_k^\top \mathbf{x}_\beta) - \mathbb{E}[\sigma'(\mathbf{w}_k^\top \mathbf{x}_\alpha)\sigma'(\mathbf{w}_k^\top \mathbf{x}_\beta)] \right) \right| > \lambda_\sigma^2 t \right] \leq 2e^{-c\check{d}t^2}.$$

To bound the mean, recall that $(\zeta_\alpha, \zeta_\beta) \equiv (\mathbf{w}_k^\top \mathbf{x}_\alpha, \mathbf{w}_k^\top \mathbf{x}_\beta)$ is bivariate Gaussian, which we may write as

$$\zeta_\alpha = u_\alpha \xi_\alpha, \qquad \zeta_\beta = u_\beta \xi_\beta + v_\beta \xi_\alpha$$

as in (29). Here, $\xi_\alpha, \xi_\beta \sim \mathcal{N}(0,1)$ are independent, $u_\alpha, u_\beta > 0$ and $v_\beta \in \mathbb{R}$, and these satisfy $|u_\alpha - 1|, |u_\beta - 1|, |v_\beta| \leq C\varepsilon$. Applying the Taylor expansion

$$\sigma'(\zeta) = \sigma'(\xi) + \sigma''(\eta)(\zeta - \xi)$$

for some $\eta$ between $\zeta$ and $\xi$, and the conditions $\mathbb{E}[\sigma'(\xi)] = b_\sigma$ and $|\sigma''(x)| \leq \lambda_\sigma$, it is easy to check that $|\mathbb{E}[\sigma'(\zeta_\alpha)\sigma'(\zeta_\beta)] - b_\sigma^2| \leq C\lambda_\sigma^2 \varepsilon$. Then part (a) follows. The statement with $(\mathbf{x}_\alpha, \mathbf{x}_\alpha)$ and $a_\sigma$ follows similarly from this Taylor expansion and the bound $|\mathbb{E}[\sigma'(\zeta_\alpha)^2] - a_\sigma| \leq C\lambda_\sigma^2 \varepsilon$.

For part (b), we write

$$T(\mathbf{x}_\alpha, \mathbf{x}_\beta) = \frac{1}{\check{d}} \sum_{k=1}^{\check{d}} \sigma'(\mathbf{w}_k^\top \mathbf{x}_\alpha)\sigma'(\mathbf{w}_k^\top \mathbf{x}_\beta) \cdot \mathbf{w}_k^\top M \mathbf{w}_k.$$

By the Hanson-Wright inequality (see [50, Theorem 1.1]),

$$\mathbb{P}\Big[\big|\mathbf{w}_k^\top M \mathbf{w}_k - \operatorname{Tr} M\big| > \|M\|_F \cdot t\sqrt{\check{d}}\Big] \le 2e^{-c\min(t^2\check{d},\,t\sqrt{\check{d}})}$$

for a constant $c > 0$. Then, applying $|\sigma'(x)| \le \lambda_\sigma$ and a union bound over $k = 1,\ldots,\check{d}$, with probability at least $1 - 2\check{d}e^{-c\min(t^2\check{d},\,t\sqrt{\check{d}})}$,

$$\left| T(\mathbf{x}_\alpha,\mathbf{x}_\beta) - \operatorname{Tr} M \cdot \frac{1}{\check{d}}\sum_{k=1}^{\check{d}} \sigma'(\mathbf{w}_k^\top \mathbf{x}_\alpha)\sigma'(\mathbf{w}_k^\top \mathbf{x}_\beta) \right| \le \|M\|_F \cdot \lambda_\sigma^2 t\sqrt{\check{d}}.$$

Then part (b) follows from combining with part (a), and applying $\operatorname{Tr} M \le \sqrt{d}\|M\|_F$. $\qquad\square$

**Corollary G.2.** *Let $\mathbf{s}_\alpha^\ell$ be as defined in (53), and let $q_\ell, r_\ell$ be the constants in (7). Under Assumption 3.2, for a constant $C > 0$, almost surely for all large $n$ and for all $\ell \in [L]$ and $\alpha \ne \beta \in [n]$,*

$$\left| \mathbf{s}_\alpha^{\ell\top} \mathbf{s}_\beta^\ell - q_{\ell-1} \right| \le C\max(\varepsilon_n, n^{-0.48}), \qquad \left| \|\mathbf{s}_\alpha^\ell\|^2 - r_{\ell-1} \right| \le C\max(\varepsilon_n, n^{-0.48}). \tag{55}$$

*Proof.* By Corollary D.2, we may assume that each matrix $X_0, \ldots, X_L$ is $(\varepsilon_n, B)$-orthonormal. Since a larger value of $\varepsilon_n$ corresponds to a weaker assumption, we may assume without loss of generality that $\varepsilon_n \ge n^{-0.48}$.

Fix $\ell \in [L]$ and $\alpha, \beta \in [n]$, and define

$$M_\ell = D_\alpha^\ell D_\beta^\ell$$

$$M_k = D_\alpha^k \frac{W_k}{\sqrt{d_{k-1}}} \cdots D_\alpha^{\ell+1} \frac{W_{\ell+1}}{\sqrt{d_\ell}} D_\alpha^\ell D_\beta^\ell \frac{W_{\ell+1}^\top}{\sqrt{d_\ell}} D_\beta^{\ell+1} \cdots \frac{W_k^\top}{\sqrt{d_{k-1}}} D_\beta^k \quad \text{for} \quad \ell+1 \le k \le L. \tag{56}$$

Recalling the definition (53) and applying the Hanson-Wright inequality conditional on $W_1, \ldots, W_L$,

$$\left| \mathbf{s}_\alpha^{\ell\top} \mathbf{s}_\beta^\ell - \frac{1}{d_L}\operatorname{Tr} M_L \right| \le C\varepsilon_n\sqrt{n} \cdot \frac{1}{d_L}\|M_L\|_F \tag{57}$$

with probability $1 - e^{-c\min(\varepsilon_n^2 n,\,\varepsilon_n\sqrt{n})} \ge 1 - e^{-n^{0.01}}$. Next, for each $k = L, L-1, \ldots, \ell+1$, we apply Lemma G.1(b) conditional on $W_1, \ldots, W_{k-1}$, with $t = \varepsilon_n$, $M = M_{k-1}/d_{k-1}$, $d = d_{k-1}$, and $\check{d} = d_k$. Note that $k - 1 \ge \ell \ge 1$, so that both $d_{k-1}$ and $d_k$ are proportional to $n$. Then

$$\left| \frac{1}{d_k}\operatorname{Tr} M_k - b_\sigma^2 \cdot \frac{1}{d_{k-1}}\operatorname{Tr} M_{k-1} \right| \le C\varepsilon_n\sqrt{n} \cdot \frac{1}{d_{k-1}}\|M_{k-1}\|_F$$

with probability $1 - e^{-n^{0.01}}$. Finally, for $k = \ell$, applying Lemma G.1(a) conditional on $W_1, \ldots, W_{\ell-1}$ and with $t = \varepsilon_n$,

$$\left| \frac{1}{d_\ell}\operatorname{Tr} M_\ell - b_\sigma^2 \right| \le C\varepsilon_n$$

with probability $1 - e^{-n^{0.01}}$. Combining these bounds, with probability $1 - C'e^{-n^{0.01}}$,

$$\left| \mathbf{s}_\alpha^{\ell\top} \mathbf{s}_\beta^\ell - (b_\sigma^2)^{L-\ell+1} \right| \le \frac{C\varepsilon_n}{\sqrt{n}}\left( \|M_L\|_F + \ldots + \|M_\ell\|_F + \sqrt{n} \right).$$

We also have $\|W_k/\sqrt{d_k}\| \le C$ for each $k = 2, \ldots, L$ with probability $1 - C'e^{-cn}$, see e.g. [53, Theorem 4.4.5]. Then, applying $\|D_k\| \le \lambda_\sigma$, we have $\|M_k\|_F \le C\sqrt{n}\|M_k\| \le C'\sqrt{n}$ for every $k = 1, \ldots, L$. Then the first bound of (55) follows. The second bound of (55) is the same, applying Lemma G.1 for $(\mathbf{x}_\alpha, \mathbf{x}_\alpha)$ instead of $(\mathbf{x}_\alpha, \mathbf{x}_\beta)$. The almost sure statement follows from the Borel-Cantelli Lemma. $\qquad\square$

**Lemma G.3.** *Under Assumption 3.2, almost surely as $n \to \infty$,*

$$\frac{1}{n}\left\| K^{NTK} - \left( r_+ \operatorname{Id} + X_L^\top X_L + \sum_{\ell=0}^{L-1} q_\ell X_\ell^\top X_\ell \right) \right\|_F^2 \to 0.$$

*Furthermore, for a constant $C > 0$, almost surely for all large $n$, $\|K^{NTK}\| \le C$.*

*Proof.* By Corollary D.2, we may assume that each matrix $X_0, \ldots, X_L$ is $(\varepsilon_n, B)$-orthonormal. Then

$$\left| \mathbf{x}_\alpha^{\ell-1 \top} \mathbf{x}_\beta^{\ell-1} \right| \leq \varepsilon_n, \qquad \left| \|\mathbf{x}_\alpha^{\ell-1}\|^2 - 1 \right| \leq \varepsilon_n.$$

Increasing $\varepsilon_n$ if necessary, we may assume $\varepsilon_n \geq n^{-0.48}$. Combining with (55), we have for the off-diagonal entries of the Hadamard product that

$$\left| ((S_\ell^\top S_\ell) \odot (X_{\ell-1}^\top X_{\ell-1}))[\alpha, \beta] - q_{\ell-1} X_{\ell-1}^\top X_{\ell-1}[\alpha, \beta] \right| \leq C\varepsilon_n^2,$$

and for the diagonal entries that

$$\left| ((S_\ell^\top S_\ell) \odot (X_{\ell-1}^\top X_{\ell-1})[\alpha, \alpha] - q_{\ell-1}(X_{\ell-1}^\top X_{\ell-1})[\alpha, \alpha] - (r_{\ell-1} - q_{\ell-1}) \right|$$

$$\leq \left| ((S_\ell^\top S_\ell) \odot (X_{\ell-1}^\top X_{\ell-1})[\alpha, \alpha] - r_{\ell-1} \right| + q_{\ell-1} \left| X_{\ell-1}^\top X_{\ell-1}[\alpha, \alpha] - 1 \right| \leq C\varepsilon_n.$$

Then applying this to (54),

$$\left\| K^{\mathrm{NTK}} - \left( r_+ \operatorname{Id} + X_L^\top X_L + \sum_{\ell=0}^{L-1} q_\ell X_\ell^\top X_\ell \right) \right\|_F^2 \leq Cn(n-1)\varepsilon_n^4 + Cn\varepsilon_n^2.$$

The first statement of the lemma then follows from the assumption $\varepsilon_n n^{1/4} \to 0$.

For the second statement on the operator norm, we have

$$\|(S_\ell^\top S_\ell) \odot (X_{\ell-1}^\top X_{\ell-1})\| \leq \max_{\alpha=1}^n \left| \mathbf{s}_\alpha^{\ell \top} \mathbf{s}_\alpha^\ell \right| \cdot \|X_{\ell-1}^\top X_{\ell-1}\|.$$

See [29, Eq. (3.7.9)], applied with $X = Y = S_\ell$. Then $\|K^{\mathrm{NTK}}\| \leq C$ follows from (54), the $(\varepsilon_n, B)$-orthonormality of each matrix $X_{\ell-1}$, and the bound for $\|\mathbf{s}_\alpha^\ell\|^2$ in (55). □

Combining Lemma G.3 and Proposition C.3, this proves Lemma 3.5.

As a remark, Lemmas G.3 and 3.5 imply $\lim \operatorname{spec} K^{\mathrm{NTK}} = \lim \operatorname{spec}(r_+ \operatorname{Id} + X_L^\top X_L)$ when $b_\sigma = 0$, since every $q_\ell = 0$ in this case. Thus, the Stieltjes transform of $\lim \operatorname{spec} K^{\mathrm{NTK}}$ is actually $m_{\mathrm{NTK}}(z) = m(-r_+ + z)$ defined by the Stieltjes transform of $\rho_\gamma^{\mathrm{MP}}$ in (6) with $\gamma = \gamma_L$. Thus in the following arguments for the limit spectrum of $K^{\mathrm{NTK}}$, we restrict to the case $b_\sigma \neq 0$.

## G.2  Unique solution of the fixed-point equation

Let $A, \Phi \in \mathbb{R}^{n \times n}$ be symmetric matrices, where $\Phi$ is positive semi-definite. Let $z \in \mathbb{C}^+$, $\alpha \in \mathbb{C}^*$, and $\gamma > 0$. For $s \in \mathbb{C}^+$, define

$$S(s) = (A + s^{-1}\Phi - z\operatorname{Id})^{-1}, \quad f_n(s) = \alpha^{-1} + \gamma \operatorname{tr} S(s)\Phi.$$

**Lemma G.4.** *(a) For any $s \in \mathbb{C}^+$, setting $S \equiv S(s)$,*

$$\operatorname{Im} f_n(s) \geq \operatorname{Im} z \cdot \gamma \operatorname{tr} S\Phi S^* \geq 0.$$

*(b) For any $s_1, s_2 \in \mathbb{C}^+$, setting $S_1 \equiv S(s_1)$ and $S_2 \equiv S(s_2)$,*

$$|f_n(s_1) - f_n(s_2)|$$
$$\leq |s_1 - s_2| \cdot \left( \frac{\operatorname{Im} f_n(s_1) - \operatorname{Im} z \cdot \gamma \operatorname{tr} S_1 \Phi S_1^*}{\operatorname{Im} s_1} \right)^{1/2} \left( \frac{\operatorname{Im} f_n(s_2) - \operatorname{Im} z \cdot \gamma \operatorname{tr} S_2 \Phi S_2^*}{\operatorname{Im} s_2} \right)^{1/2}$$

*Proof.* For part (a), let us write

$$S\Phi = S\Phi S^*(A + s^{-1}\Phi - z\operatorname{Id})^* = S\Phi S^* A + (1/s^*)S\Phi S^*\Phi - z^* S\Phi S^*.$$

Since $S\Phi S^*$ is Hermitian and positive semi-definite, the quantities $\operatorname{tr} S\Phi S^* A$, $\operatorname{tr} S\Phi S^*\Phi$, and $\operatorname{tr} S\Phi S^*$ are all real, and the latter two are nonnegative. Then

$$\operatorname{Im} f_n(s) = \operatorname{Im} \alpha^{-1} + \gamma \operatorname{Im} \operatorname{tr} S\Phi = \operatorname{Im} \alpha^{-1} + \frac{\operatorname{Im} s}{|s|^2} \cdot \gamma \operatorname{tr} S\Phi S^*\Phi + \operatorname{Im} z \cdot \gamma \operatorname{tr} S\Phi S^*. \quad (58)$$

Each term on the right side of (58) is nonnegative, and dropping the first two of these terms yields (a).

For part (b), applying the identity $A^{-1} - B^{-1} = A^{-1}(B - A)B^{-1}$, we have

$$S_1 - S_2 = S_1(s_2^{-1}\Phi - s_1^{-1}\Phi)S_2 = \frac{s_1 - s_2}{s_1 s_2} S_1 \Phi S_2,$$

so

$$f_n(s_1) - f_n(s_2) = \gamma \operatorname{tr} S_1 \Phi - \gamma \operatorname{tr} S_2 \Phi = \frac{\gamma(s_1 - s_2)}{s_1 s_2} \operatorname{tr} S_1 \Phi S_2 \Phi.$$

Applying Cauchy-Schwarz to the inner-product $\langle S_1, S_2 \rangle_\Phi = \operatorname{tr} S_1 \Phi S_2^* \Phi$,

$$|\operatorname{tr} S_1 \Phi S_2 \Phi|^2 = |\langle S_1, S_2^* \rangle_\Phi|^2 \leq \langle S_1, S_1 \rangle_\Phi \cdot \langle S_2^*, S_2^* \rangle_\Phi = \operatorname{tr} S_1 \Phi S_1^* \Phi \cdot \operatorname{tr} S_2 \Phi S_2^* \Phi.$$

Then

$$|f_n(s_1) - f_n(s_2)| \leq |s_1 - s_2| \cdot \left( \frac{\gamma \operatorname{tr} S_1 \Phi S_1^* \Phi}{|s_1|^2} \right)^{1/2} \left( \frac{\gamma \operatorname{tr} S_2 \Phi S_2^* \Phi}{|s_2|^2} \right)^{1/2}.$$

Dropping $\operatorname{Im} \alpha^{-1}$ in (58) and applying this to upper-bound $\gamma \operatorname{tr} S\Phi S^* \Phi / |s|^2$, part (b) follows. □

**Corollary G.5.** *As $n \to \infty$, suppose that $f_n(s) \to f(s)$ pointwise for each $s \in \mathbb{C}^+$, the empirical spectral distributions of $\Phi$ and $A$ converge weakly to deterministic limits, and the limit for $\Phi$ is not the point distribution at 0. Then the fixed-point equation $s = f(s)$ has at most one solution $s \in \mathbb{C}^+$.*

*Proof.* Let us first show that for each $s \in \mathbb{C}^+$ and a value $c_0(s) > 0$ independent of $n$,

$$\liminf_{n \to \infty} \operatorname{tr} S(s)\Phi S(s)^* \geq c_0(s) > 0. \tag{59}$$

Denoting $S \equiv S(s)$ and applying the von Neumann trace inequality,

$$\operatorname{tr} S\Phi S^* = \frac{1}{n} \operatorname{Tr} \Phi S^* S \geq \frac{1}{n} \sum_{\alpha=1}^{n} \lambda_\alpha(\Phi)\lambda_{n+1-\alpha}(S^* S),$$

where $\lambda_1(\cdot) \geq \ldots \geq \lambda_n(\cdot)$ denote the sorted eigenvalues. Since $\Phi$ has a non-degenerate limit spectrum, there is a constant $\varepsilon > 0$ for which $\lambda_{\varepsilon n}(\Phi) > \varepsilon$ for all large $n$. (Throughout the proof, $\varepsilon n$, $\varepsilon n/2$, etc. should be understood as their roundings to the nearest integer.) Then

$$\operatorname{tr} S\Phi S^* \geq \varepsilon \cdot \frac{1}{n} \sum_{\alpha=1}^{\varepsilon n} \lambda_{n+1-\alpha}(S^* S).$$

Denoting by $\sigma_\alpha(\cdot)$ the $\alpha^{\text{th}}$ largest singular value, observe that

$$\lambda_{n+1-\alpha}(S^* S) = \sigma_{n+1-\alpha}(S)^2 = \sigma_\alpha(A + s^{-1}\Phi - z\operatorname{Id})^{-2}.$$

Applying $\sigma_{\alpha+\beta-1}(A + B) \leq \sigma_\alpha(A) + \sigma_\beta(B)$, we have

$$\sigma_\alpha(A + s^{-1}\Phi - z\operatorname{Id}) \leq \sigma_{\alpha/2}(A) + |s|^{-1}\sigma_{\alpha/2+1}(\Phi) + |z|.$$

Since the spectra of $A$ and $\Phi$ converge to deterministic limits, this implies that there is a constant $C(s) > 0$ (also depending on $z$ and $\varepsilon$) such that $\sigma_\alpha(A + s^{-1}\Phi - z\operatorname{Id}) \leq C(s)$ for every $\alpha \in [\varepsilon n/2, \varepsilon n]$ and all large $n$. Thus

$$\operatorname{tr} S\Phi S^* \geq \varepsilon \cdot \frac{\varepsilon n - \varepsilon n/2}{n} \cdot C(s)^{-2}$$

for all large $n$, and this shows the claim (59).

Then, taking the limit $n \to \infty$ in Lemma G.4(b), we get

$$|f(s_1) - f(s_2)| \leq |s_1 - s_2| \cdot \left( \frac{\operatorname{Im} f(s_1) - \operatorname{Im} z \cdot \gamma c_0(s_1)}{\operatorname{Im} s_1} \right)^{1/2} \left( \frac{\operatorname{Im} f(s_2) - \operatorname{Im} z \cdot \gamma c_0(s_2)}{\operatorname{Im} s_2} \right)^{1/2}.$$

If $s_1 = f(s_1)$ and $s_2 = f(s_2)$, then this yields $|s_1 - s_2| \leq |s_1 - s_2| \cdot h(s_1, s_2)$ for some quantity $h(s_1, s_2) \in [0, 1)$, where $h(s_1, s_2) < 1$ strictly because $c_0(s_1), c_0(s_2) > 0$. This contradiction implies $s_1 = s_2$, so the equation $s = f(s)$ has at most one solution $s \in \mathbb{C}^+$. □

## G.3 Proof of Proposition 3.6 and Theorem 3.7

The operator norm bound in Theorem 3.7 was shown in Lemma G.3. For the spectral convergence, note that by Lemma 3.5, the limit Stieltjes transform of $K^{\text{NTK}}$ at any $z \in \mathbb{C}^+$ is given by

$$m_{\text{NTK}}(z) = \lim_{n \to \infty} \text{tr}\left( (-z + r_+)\,\text{Id} + X_L^\top X_L + \sum_{\ell=0}^{L-1} q_\ell X_\ell^\top X_\ell \right)^{-1},$$

provided that this limit exists and defines the Stieltjes transform of a probability measure. For

$$\mathbf{z} = (z_{-1}, \dots, z_\ell) \in \mathbb{C}^- \times \mathbb{R}^\ell \times \mathbb{C}^*, \qquad \mathbf{w} = (w_{-1}, \dots, w_\ell) \in \mathbb{C}^{\ell+2},$$

recall the functions

$$\mathbf{z} \mapsto s_\ell(\mathbf{z}), \quad (\mathbf{z}, \mathbf{w}) \mapsto t_\ell(\mathbf{z}, \mathbf{w})$$

defined recursively by (12) and (13). Proposition 3.6 and Theorem 3.7 are immediate consequences of the following extended result.

**Lemma G.6.** *Suppose $b_\sigma \neq 0$. Under Assumption 3.2, for each $\ell = 1, \dots, L$:*

*(a) For every $\mathbf{z} \in \mathbb{C}^- \times \mathbb{R}^\ell \times \mathbb{C}^*$, the equation (12) has a unique fixed point $s_\ell(\mathbf{z}) \in \mathbb{C}^+$.*

*(b) For every $(\mathbf{z}, \mathbf{w}) \in (\mathbb{C}^- \times \mathbb{R}^\ell \times \mathbb{C}^*) \times \mathbb{C}^{\ell+2}$, almost surely*

$$t_\ell(\mathbf{z}, \mathbf{w})$$
$$= \lim_{n \to \infty} \text{tr}\left( z_{-1}\,\text{Id} + z_0 X_0^\top X_0 + \dots + z_\ell X_\ell^\top X_\ell \right)^{-1} \left( w_{-1}\,\text{Id} + w_0 X_0^\top X_0 + \dots + w_\ell X_\ell^\top X_\ell \right).$$
(60)

*In particular, for any $z_{-1}, \dots, z_\ell \in \mathbb{R}$ where $z_\ell \neq 0$,*

$$\lim \text{spec}\ z_{-1}\,\text{Id} + z_0 X_0^\top X_0 + \dots + z_\ell X_\ell^\top X_\ell = \nu$$

*where $\nu$ is a probability measure on $\mathbb{R}$ with Stieltjes transform*

$$m(z) = t_\ell\Big( (-z + z_{-1}, z_0, \dots, z_\ell), (1, 0, \dots, 0) \Big).$$

*Proof.* By Corollary D.2, we may assume that each matrix $X_0, \dots, X_L$ is $(\varepsilon_n, B)$-orthonormal.

Define $\Phi_\ell, \tilde{\Phi}_\ell$ by (45) and (46). For $\mathbf{z} = (z_{-1}, \dots, z_\ell)$, let us write as shorthand

$$\mathbf{z} \cdot \mathbf{X}^\top \mathbf{X}(\ell) = z_{-1}\,\text{Id} + z_0 X_0^\top X_0 + \dots + z_\ell X_\ell^\top X_\ell,$$

where the parenthetical $(\ell)$ signifies the index of the last term in this sum. Let us define similarly $\mathbf{w} \cdot \mathbf{X}^\top \mathbf{X}(\ell)$.

Note that part (b) holds for $\ell = 0$, by the assumption $\lim \text{spec}\ X_0^\top X_0 = \mu_0$, the definition of $t_0((z_{-1}, z_0), (w_{-1}, w_0))$ in (11), and the fact that the function $x \mapsto (w_{-1} + w_0 x)/(z_{-1} + z_0 x)$ is continuous and bounded over the non-negative real line when $z_{-1} \in \mathbb{C}^-$ and $z_0 \in \mathbb{C}^*$.

We induct on $\ell$. Suppose that part (b) holds for $\ell - 1$. To show part (a) for $\ell$, fix any $\mathbf{z} = (z_{-1}, \dots, z_\ell) \in \mathbb{C}^- \times \mathbb{R}^\ell \times \mathbb{C}^*$ (not depending on $n$) and consider the matrix

$$R = \left( \mathbf{z} \cdot \mathbf{X}^\top \mathbf{X}(\ell) \right)^{-1}.$$
(61)

We apply the analysis of Appendix E, conditional on $X_0, \dots, X_{\ell-1}$, and with the identifications

$$\check{X} = X_\ell, \qquad X = X_{\ell-1}, \qquad \check{d} = d_\ell, \qquad d = d_{\ell-1},$$

$$A = z_0 X_0^\top X_0 + \dots + z_{\ell-1} X_{\ell-1}^\top X_{\ell-1}, \qquad \alpha = z_\ell, \qquad z = -z_{-1}.$$

Observe that $\alpha \in \mathbb{C}^*$ and $z \in \mathbb{C}^-$. The matrix $R$ in (61) is exactly

$$R = (A + \alpha \check{X}^\top \check{X} - z\,\text{Id})^{-1}.$$

Since each $X_0, \ldots, X_{\ell-1}$ is $(\varepsilon_n, B)$-orthonormal, we have $\|A\| \leq C$ for some constant $C > 0$ (depending on $z_{-1}, \ldots, z_\ell, \lambda_\sigma$). Thus Assumption E.1 holds, conditional on $X_0, \ldots, X_{\ell-1}$. Let us define the $n$-dependent parameter

$$\bar{s} = \frac{1}{\alpha} + \frac{n}{d_\ell} \operatorname{tr} \mathbb{E}_{W_\ell}[R\Phi_\ell]$$

where this expectation is over only the weights $W_\ell$. Then, applying Lemma E.2(b) with a value $t \equiv t_n$ such that $t \to 0$ and $nt^2 \gg \log n$, we obtain

$$\left| \bar{s} - \frac{1}{\alpha} - \frac{n}{d_\ell} \operatorname{tr}(A + \bar{s}^{-1}\Phi_\ell - z\operatorname{Id})^{-1}\Phi_\ell \right| \to 0 \tag{62}$$

almost surely as $n \to \infty$.

Proposition E.3 shows that $|\bar{s}|$ is bounded, so for any subsequence in $n$, there is a further sub-subsequence where $\bar{s} \to s_0$ for a limit $s_0 \equiv s_0(\mathbf{z}) \in \mathbb{C}^+$. Let us now replace $\bar{s}$ and $\Phi_\ell$ above by $s_0$ and $\tilde{\Phi}_\ell$: First we have

$$\operatorname{tr}\left(A + \bar{s}^{-1}\Phi_\ell - z\operatorname{Id}\right)^{-1}\Phi_\ell - \operatorname{tr}\left(A + s_0^{-1}\tilde{\Phi}_\ell - z\operatorname{Id}\right)^{-1}\Phi_\ell \to 0$$

by the same argument as (49). Then, we have

$$\left| \operatorname{tr}\left(A + s_0^{-1}\Phi_\ell - z\operatorname{Id}\right)^{-1}\Phi_\ell - \operatorname{tr}\left(A + s_0^{-1}\tilde{\Phi}_\ell - z\operatorname{Id}\right)^{-1}\Phi_\ell \right|$$
$$= \left| s_0^{-1} \operatorname{tr}\left(A + s_0^{-1}\Phi_\ell - z\operatorname{Id}\right)^{-1}(\tilde{\Phi}_\ell - \Phi_\ell)\left(A + s_0^{-1}\tilde{\Phi}_\ell - z\operatorname{Id}\right)^{-1}\Phi_\ell \right|$$
$$\leq \frac{C}{n}\|\tilde{\Phi}_\ell - \Phi_\ell\|_F \cdot \left\|(A + s_0^{-1}\tilde{\Phi} - z\operatorname{Id})^{-1}\Phi(A + s_0^{-1}\Phi - z\operatorname{Id})^{-1}\right\|_F$$
$$\leq \frac{C}{\sqrt{n}}\|\tilde{\Phi}_\ell - \Phi_\ell\|_F \cdot \|(A + s_0^{-1}\tilde{\Phi} - z\operatorname{Id})^{-1}\| \cdot \|\Phi\| \cdot \|(A + s_0^{-1}\Phi - z\operatorname{Id})^{-1}\| \to 0,$$

where the convergence to 0 follows from Lemma G.3. Finally, we have

$$\left| \operatorname{tr}\left(A + s_0^{-1}\Phi_\ell - z\operatorname{Id}\right)^{-1}\Phi_\ell - \operatorname{tr}\left(A + s_0^{-1}\Phi_\ell - z\operatorname{Id}\right)^{-1}\tilde{\Phi}_\ell \right|$$
$$\leq \frac{1}{n}\|(A + s_0^{-1}\Phi_\ell - z\operatorname{Id})^{-1}\|_F \cdot \|\Phi_\ell - \tilde{\Phi}_\ell\|_F \leq \frac{1}{\sqrt{n}}\|(A + s_0^{-1}\Phi_\ell - z\operatorname{Id})^{-1}\| \cdot \|\Phi_\ell - \tilde{\Phi}_\ell\|_F \to 0.$$

Applying these approximations to (62), we have almost surely along this sub-subsequence that

$$\left| s_0 - \frac{1}{\alpha} - \gamma_\ell \operatorname{tr}(A + s_0^{-1}\tilde{\Phi}_\ell - z\operatorname{Id})^{-1}\tilde{\Phi}_\ell \right| \to 0. \tag{63}$$

Now observe from the definitions of $A$, $\tilde{\Phi}_\ell$, and $z$ that

$$A + s_0^{-1}\tilde{\Phi}_\ell - z\operatorname{Id} = \left(z_{-1} + \frac{1 - b_\sigma^2}{s_0}\right)\operatorname{Id} + \sum_{k=0}^{\ell-2} z_k X_k^\top X_k + \left(z_{\ell-1} + \frac{b_\sigma^2}{s_0}\right)X_{\ell-1}^\top X_{\ell-1},$$
$$\tilde{\Phi}_\ell = (1 - b_\sigma^2)\operatorname{Id} + b_\sigma^2 X_{\ell-1}^\top X_{\ell-1}.$$

Then, applying (63) and the induction hypothesis that part (b) holds for $\ell - 1$, we obtain that the value $s_0$ must satisfy

$$s_0 = \frac{1}{\alpha} + \gamma_\ell t_{\ell-1}\left(\mathbf{z}_{\mathrm{prev}}(s_0, \mathbf{z}), (1 - b_\sigma^2, 0, \ldots, 0, b_\sigma^2)\right),$$

where $\mathbf{z}_{\mathrm{prev}}$ is defined in (14). This shows the existence of a solution (in $\mathbb{C}^+$) to the fixed-point equation (12). Notice that because $b_\sigma \neq 0$ and $s_0 \in \mathbb{C}^+$, the last entry of $\mathbf{z}_{\mathrm{prev}}(s_0, \mathbf{z})$ is in $\mathbb{C}^*$ and $(\mathbf{z}_{\mathrm{prev}}(s_0, \mathbf{z}), (1 - b_\sigma^2, 0, \ldots, 0, b_\sigma^2))$ is in the domain of function $t_{\ell-1}$.

To show uniqueness, we apply Corollary G.5: For any fixed $s \in \mathbb{C}^+$, defining

$$f_n(s) = \frac{1}{\alpha} + (n/d_\ell) \operatorname{tr}(A + s^{-1}\Phi_\ell - z\operatorname{Id})^{-1}\Phi_\ell,$$

the same arguments as above establish that

$$\lim_{n\to\infty} f_n(s) = f(s) \equiv \frac{1}{\alpha} + \gamma_\ell t_{\ell-1}\Big(\mathbf{z}_{\text{prev}}(s, \mathbf{z}), (1 - b_\sigma^2, 0, \dots, 0, b_\sigma^2)\Big).$$

Part (b) holding for $\ell - 1$ implies that both $A$ and $\Phi_\ell$ have deterministic spectral limits, where

$$\lim \operatorname{spec} \Phi_\ell = \lim \operatorname{spec} \tilde{\Phi}_\ell$$

by (47). This cannot be the point distribution at 0, because (28) implies that $\operatorname{tr} \Phi_\ell \geq 1/2$ for all large $n$, and $\|\Phi_\ell\| \leq C$ so at least $n/(2C)$ eigenvalues of $\Phi_\ell$ exceed $1/2$ for every $n$. Thus, Corollary G.5 implies that the fixed point $s = f(s)$ is unique. So the fixed point $s_\ell(\mathbf{z}) \in \mathbb{C}^+$ is uniquely defined by (12), and this shows part (a) for $\ell$.

By the uniqueness of this fixed point, we have also shown that $s_0 = s_\ell(\mathbf{z})$, where $s_0$ is the limit of $\bar{s}$ along the above sub-subsequence. Since for any subsequence in $n$, there exists a sub-subsequence for this which holds, this shows that $\lim_{n\to\infty} \bar{s} = s_\ell(\mathbf{z})$ almost surely.

Now, to show that part (b) holds for $\ell$, let us also fix any $\mathbf{w} = (w_{-1}, \dots, w_\ell) \in \mathbb{C}^{\ell+2}$. Using that $z_\ell \neq 0$, we may write

$$\mathbf{w} \cdot \mathbf{X}^\top \mathbf{X}(\ell) = \frac{w_\ell}{z_\ell} \cdot \mathbf{z} \cdot \mathbf{X}^\top \mathbf{X}(\ell) + \mathbf{w}_{\text{prev}} \cdot \mathbf{X}^\top \mathbf{X}(\ell - 1),$$

where $\mathbf{w}_{\text{prev}}$ is as defined in (15). Then

$$\Big(\mathbf{z} \cdot \mathbf{X}^\top \mathbf{X}(\ell)\Big)^{-1}\Big(\mathbf{w} \cdot \mathbf{X}^\top \mathbf{X}(\ell)\Big) = \frac{w_\ell}{z_\ell} \operatorname{Id} + \Big(\mathbf{z} \cdot \mathbf{X}^\top \mathbf{X}(\ell)\Big)^{-1}\Big(\mathbf{w}_{\text{prev}} \cdot \mathbf{X}^\top \mathbf{X}(\ell - 1)\Big). \quad (64)$$

We now apply Lemma E.2(a) conditional on $X_0, \dots, X_{\ell-1}$, with the same identifications as above and with

$$M = \mathbf{w}_{\text{prev}} \cdot \mathbf{X}^\top \mathbf{X}(\ell - 1).$$

Note that $M$ is indeed deterministic conditional on $X_0, \dots, X_{\ell-1}$, and $\|M\| \leq C$ for a constant $C > 0$ (depending on $\mathbf{z}$ and $\mathbf{w}$) since $X_0, \dots, X_{\ell-1}$ are $(\varepsilon_n, B)$-orthonormal. Then, applying Lemma E.2(a),

$$\operatorname{tr}\left[\Big(\mathbf{z} \cdot \mathbf{X}^\top \mathbf{X}(\ell)\Big)^{-1}\Big(\mathbf{w}_{\text{prev}} \cdot \mathbf{X}^\top \mathbf{X}(\ell-1)\Big)\right] - \operatorname{tr}\left[(A + \bar{s}^{-1}\Phi_\ell - z \operatorname{Id})^{-1}\Big(\mathbf{w}_{\text{prev}} \cdot \mathbf{X}^\top \mathbf{X}(\ell-1)\Big)\right] \to 0.$$

By the same arguments as above, we may replace $\bar{s}$ by $s_0 = s_\ell(\mathbf{z})$ and $\Phi_\ell$ by $\tilde{\Phi}_\ell$. Then, applying this to (64),

$$\operatorname{tr}\left[\Big(\mathbf{z} \cdot \mathbf{X}^\top \mathbf{X}(\ell)\Big)^{-1}\Big(\mathbf{w} \cdot \mathbf{X}^\top \mathbf{X}(\ell)\Big)\right] - \frac{w_\ell}{z_\ell} - \operatorname{tr}\left[(A + s_\ell(\mathbf{z})^{-1}\tilde{\Phi}_\ell - z \operatorname{Id})^{-1}\Big(\mathbf{w}_{\text{prev}} \cdot \mathbf{X}^\top \mathbf{X}(\ell-1)\Big)\right] \to 0.$$

Finally, applying that part (b) holds for $\ell - 1$, this yields

$$\lim_{n\to\infty} \operatorname{tr}\left[\Big(\mathbf{z} \cdot \mathbf{X}^\top \mathbf{X}(\ell)\Big)^{-1}\Big(\mathbf{w} \cdot \mathbf{X}^\top \mathbf{X}(\ell)\Big)\right] = \frac{w_\ell}{z_\ell} + t_{\ell-1}(\mathbf{z}_{\text{prev}}(s_\ell(\mathbf{z}), \mathbf{z}), \mathbf{w}_{\text{prev}}),$$

which is the definition of $t_\ell(\mathbf{z}, \mathbf{w})$. This establishes (60).

For any fixed $z_{-1}, \dots, z_\ell \in \mathbb{R}$ where $z_\ell \neq 0$, and any fixed $z \in \mathbb{C}^+$, this implies that the Stieltjes transform of $\mathbf{z} \cdot \mathbf{X}^\top \mathbf{X}(\ell)$ has the almost sure limit

$$m(z) = t_\ell\Big((-z + z_{-1}, z_0, \dots, z_\ell), (1, 0, \dots, 0)\Big).$$

So $m(z)$ defines the Stieltjes transform of a sub-probability distribution $\nu$, and the empirical eigenvalue distribution of $\mathbf{z} \cdot \mathbf{X}^\top \mathbf{X}(\ell)$ converges vaguely a.s. to $\nu$. Since $\|\mathbf{z} \cdot \mathbf{X}^\top \mathbf{X}(\ell)\|$ is bounded because $X_0, \dots, X_L$ are $(\varepsilon_n, B)$-orthonormal, this limit $\nu$ must in fact be a probability distribution, and the eigenvalue distribution converges weakly to $\nu$. This concludes the induction and the proof. $\qquad \square$

# H  Multi-dimensional outputs and rescaled parametrizations

In this section, we provide some motivation for the form of the NTK in (17) for networks with a $k$-dimensional output, and we prove Theorem 3.8 regarding its spectrum.

## H.1 Derivation of (17) from gradient flow training

Consider gradient flow training of the network (16), with training samples $(\mathbf{x}_\alpha, \mathbf{y}_\alpha)_{\alpha=1}^n$ where $\mathbf{x}_\alpha \in \mathbb{R}^{d_0}$ and $\mathbf{y}_\alpha \in \mathbb{R}^k$, under the general training loss

$$F(\theta) = \sum_{\alpha=1}^n \mathcal{L}(f_\theta(\mathbf{x}_\alpha), \mathbf{y}_\alpha).$$

Here, $\mathcal{L} : \mathbb{R}^k \times \mathbb{R}^k \to \mathbb{R}$ is the loss function. We denote by $\nabla \mathcal{L}(f_\theta(\mathbf{x}_\alpha), \mathbf{y}_\alpha) \in \mathbb{R}^k$ the gradient of $\mathcal{L}$ with respect to its first argument, and by $\nabla_{W_\ell} f_\theta(\mathbf{x}_\alpha) \in \mathbb{R}^{\dim(W_\ell) \times k}$ the Jacobian of $f_\theta(\mathbf{x}_\alpha)$ with respect to the weights $W_\ell$.

Consider a possibly reweighted gradient-flow training of $\theta$, where the evolution of weights $W_\ell$ is given by

$$\frac{d}{dt} W_\ell(t) = -\tau_\ell \cdot \nabla_{W_\ell} F(\theta(t)) = -\tau_\ell \sum_{\alpha=1}^n \nabla_{W_\ell} f_{\theta(t)}(\mathbf{x}_\alpha) \cdot \nabla \mathcal{L}(f_{\theta(t)}(\mathbf{x}_\alpha), \mathbf{y}_\alpha).$$

The learning rate for each weight matrix $W_\ell$ is scaled by a constant $\tau_\ell$—this may arise, for example, from reparametrizing the network (16) using $\widetilde{W}_\ell = \tau_\ell^{-1} \cdot W_\ell$ and considering gradient flow training for $\widetilde{W}_\ell$. Denoting the vectorization of all training predictions and its Jacobian by

$$f_\theta(X) = (f_\theta^1(X), \ldots, f_\theta^k(X)) \in \mathbb{R}^{nk}, \qquad \nabla_{W_\ell} f_\theta(X) \in \mathbb{R}^{\dim(W_\ell) \times nk},$$

and the corresponding vectorization of $(\nabla \mathcal{L}(f_\theta(\mathbf{x}_\alpha), \mathbf{y}_\alpha))_{\alpha=1}^n$ by $\nabla \mathcal{L}(f_\theta(X), \mathbf{y}) \in \mathbb{R}^{nk}$, this may be written succinctly as

$$\frac{d}{dt} W_\ell(t) = -\tau_\ell \cdot \nabla_{W_\ell} f_{\theta(t)}(X) \cdot \nabla \mathcal{L}(f_{\theta(t)}(X), \mathbf{y}).$$

Then the time evolution of in-sample predictions is given by

$$\begin{aligned}
\frac{d}{dt} f_{\theta(t)}(X) &= \left( \nabla_\theta f_{\theta(t)}(X) \right)^\top \cdot \frac{d}{dt} \theta(t) \\
&= -\sum_{\ell=1}^{L+1} \tau_\ell \left( \nabla_{W_\ell} f_{\theta(t)}(X) \right)^\top \left( \nabla_{W_\ell} f_{\theta(t)}(X) \right) \cdot \nabla \mathcal{L}(f_{\theta(t)}(X), \mathbf{y}) \\
&= -K^{\mathrm{NTK}}(t) \cdot \nabla \mathcal{L}(f_{\theta(t)}(X), \mathbf{y}),
\end{aligned}$$

where $K^{\mathrm{NTK}}$ is the matrix defined in (17). For $\tau_1 = \ldots = \tau_{L+1} = 1$, this matrix is simply

$$K^{\mathrm{NTK}} = \left( \nabla_\theta f_\theta(X) \right)^\top \left( \nabla_\theta f_\theta(X) \right) \in \mathbb{R}^{nk \times nk},$$

which is a flattening of the neural tangent kernel $K \in \mathbb{R}^{n \times n \times k \times k}$ (identified as a map $K : \mathbb{R}^{n \times n} \to \mathbb{R}^{k \times k}$) that is defined in [27].

## H.2 Proof of Theorem 3.8

The matrix $K^{\mathrm{NTK}}$ in (17) admits a $k \times k$ block decomposition

$$K^{\mathrm{NTK}} = \begin{pmatrix} K_{11}^{\mathrm{NTK}} & \cdots & K_{1k}^{\mathrm{NTK}} \\ \vdots & \ddots & \vdots \\ K_{k1}^{\mathrm{NTK}} & \cdots & K_{kk}^{\mathrm{NTK}} \end{pmatrix}, \qquad K_{ij}^{\mathrm{NTK}} = \sum_{\ell=1}^{L+1} \tau_\ell \left( \nabla_{W_\ell} f_\theta^i(X) \right)^\top \left( \nabla_{W_\ell} f_\theta^j(X) \right) \in \mathbb{R}^{n \times n}.$$

Writing

$$W_{L+1} = \begin{pmatrix} \mathbf{w}_1^\top \\ \vdots \\ \mathbf{w}_k^\top \end{pmatrix},$$

a computation using the chain rule similar to (54) verifies that

$$K_{ij}^{\mathrm{NTK}} = \mathbf{1}\{i = j\} \tau_{L+1} X_L^\top X_L + \sum_{\ell=1}^L \tau_\ell (S_\ell^{i \top} S_\ell^j) \odot (X_{\ell-1}^\top X_{\ell-1})$$

where $S_\ell^i \in \mathbb{R}^{d_\ell \times n}$ is the matrix with the same column-wise definition as in (53), replacing $\mathbf{w}$ by $\mathbf{w}_i$.

**Lemma H.1.** *Under the assumptions of Theorem 3.8, for any indices $i \neq j \in [k]$, almost surely as $n \to \infty$,*

$$\frac{1}{n}\|K_{ij}^{NTK}\|_F^2 \to 0.$$

*Furthermore, for a constant $C > 0$, almost surely for all large $n$, $\|K_{ij}^{NTK}\| \leq C$.*

*Proof.* By Corollary D.2, we may assume that each $X_0, \ldots, X_L$ is $(\varepsilon_n, B)$-orthonormal.

Let us fix $i, j, \ell$ and denote the columns of $S_\ell^i$ and $S_\ell^j$ by $\mathbf{s}_\alpha^{\ell,i}$ and $\mathbf{s}_\beta^{\ell,j}$ for $\alpha, \beta \in [n]$. We apply the Hanson-Wright inequality conditional on $W_1, \ldots, W_L$, which is similar to (57). However, since $\mathbf{w}_i$ and $\mathbf{w}_j$ are independent, there is no trace term, and we obtain instead

$$\left|\mathbf{s}_\alpha^{\ell,i\top}\mathbf{s}_\beta^{\ell,j}\right| \leq C\varepsilon_n \sqrt{n}\frac{1}{d_L}\|M_L\|_F$$

for both $\alpha = \beta$ and $\alpha \neq \beta$ with probability $1 - e^{-n^{0.01}}$, where $M_L$ is the same matrix as defined in (56). Applying the bound $\|M_L\|_F \leq C\sqrt{n}$ as in the proof of Corollary G.2, this yields

$$\left|\mathbf{s}_\alpha^{\ell,i\top}\mathbf{s}_\beta^{\ell,j}\right| \leq C\varepsilon_n$$

almost surely for all $\alpha, \beta \in [n]$ and all large $n$. Combining with the $(\varepsilon_n, B)$-orthonormality of $X_{\ell-1}$, we get for $\alpha \neq \beta$ that

$$\left|(S_\ell^{i\top}S_\ell^j) \odot (X_{\ell-1}^\top X_{\ell-1})[\alpha, \beta]\right| \leq C\varepsilon_n^2, \qquad \left|(S_\ell^{i\top}S_\ell^j) \odot (X_{\ell-1}^\top X_{\ell-1})[\alpha, \alpha]\right| \leq C\varepsilon_n.$$

Then

$$\|(S_\ell^{i\top}S_\ell^j) \odot (X_{\ell-1}^\top X_{\ell-1})\|_F^2 \leq Cn(n-1)\varepsilon_n^4 + Cn\varepsilon_n^2,$$

and the first statement follows from the assumption $\varepsilon_n n^{1/4} \to 0$. The second statement on the operator norm follows from the bound

$$\|(S_\ell^{i\top}S_\ell^j) \odot (X_{\ell-1}^\top X_{\ell-1})\| \leq \left(\max_{\alpha=1}^n \left|\mathbf{s}_\alpha^{\ell,i\top}\mathbf{s}_\alpha^{\ell,i}\right|\right)^{1/2} \left(\max_{\alpha=1}^n \left|\mathbf{s}_\alpha^{\ell,j\top}\mathbf{s}_\alpha^{\ell,j}\right|\right)^{1/2} \cdot \|X_{\ell-1}^\top X_{\ell-1}\|.$$

See [29, Eq. (3.7.9)] applied with $X = S_\ell^i$ and $Y = S_\ell^j$. The bound $\|K_{ij}^{NTK}\| \leq C$ then follows from the $(\varepsilon_n, B)$-orthonormality of $X_{\ell-1}$ and Corollary G.2, applied to $S_\ell^i$ and $S_\ell^j$. $\qquad \square$

Applying this lemma together with Proposition C.3, we obtain

$$\text{lim spec } K^{\text{NTK}} = \text{lim spec}\begin{pmatrix} K_{11}^{\text{NTK}} & & \\ & \ddots & \\ & & K_{kk}^{\text{NTK}} \end{pmatrix}$$

where the off-diagonal blocks $K_{ij}^{\text{NTK}}$ may be replaced by 0. Then the limit spectral distribution of $K^{\text{NTK}}$ is an equally weighted mixture of those of $K_{11}^{\text{NTK}}, \ldots, K_{kk}^{\text{NTK}}$. For each diagonal block $K_{ii}^{\text{NTK}}$, the argument of Lemma G.3 shows that

$$\text{lim spec } K_{ii}^{\text{NTK}} = \text{lim spec}\left(\tau \cdot r_+ \text{ Id} + \tau_{L+1} X_L^\top X_L + \sum_{\ell=0}^{L-1} \tau_{\ell+1} q_\ell X_\ell^\top X_\ell\right).$$

Then by Theorem 3.7, each diagonal block $K_{ii}^{\text{NTK}}$ has the same limit spectral distribution, whose Stieltjes transform is given by the function $m_{\text{NTK}}(z)$ in Theorem 3.8. Furthermore, since $\|K_{ii}^{\text{NTK}}\| \leq C$ by Lemma G.3 and $\|K_{ij}^{\text{NTK}}\| \leq C$ for $i \neq j$ by Lemma H.1, this shows $\|K^{\text{NTK}}\| \leq C$. This establishes Theorem 3.8.

Again, when $b_\sigma = 0$, the limit spectrum of each $K_{ii}^{\text{NTK}}$ reduces to $\text{lim spec}(\tau \cdot r_+ \text{ Id} + \tau_{L+1} X_L^\top X_L)$, which can be computed via the Stieltjes transform of $\rho_{\gamma_L}^{\text{MP}}$.

# I  Reduction to result of Pennington and Worah [46] for one hidden layer

Consider the one-hidden-layer conjugate kernel

$$K^{\mathrm{CK}} = X_1^\top X_1 = \frac{1}{d_1}\sigma(W_1 X)^\top \sigma(W_1 X) \in \mathbb{R}^{n\times n}.$$

Define an associated covariance matrix

$$M = \frac{1}{n}\sigma(W_1 X)\sigma(W_1 X)^\top \in \mathbb{R}^{d_1\times d_1}, \tag{65}$$

and observe that the eigenvalues of $K^{\mathrm{CK}}$ are those of $M$ multiplied by $n/d_1$ and padded by $n - d_1$ additional zeros (or with $d_1 - n$ zeros removed, if $n - d_1 < 0$). [46, Theorem 1] characterizes the limit spectral distribution of $M$ in terms of a quartic equation in its Stieltjes transform, under the additional assumptions that $X$ has i.i.d. $\mathcal{N}(0, 1/d_0)$ entries and $n/d_0 \to \gamma_0 \in (0,\infty)$.[4] By Theorem 3.4, this should be equivalent to the description

$$\lim \operatorname{spec} K^{\mathrm{CK}} = \rho_{\gamma_1}^{\mathrm{MP}} \boxtimes \left( (1 - b_\sigma^2) + b_\sigma^2 \mu_0 \right) \tag{66}$$

for the limit spectrum of $K^{\mathrm{CK}}$, if we specialize to $\mu_0 = \rho_{\gamma_0}^{\mathrm{MP}}$ being the Marcenko-Pastur limit of the input gram matrix $X^\top X$. We derive this equivalence in this section.

Let $m_K(z)$ and $m_M(z)$ be the *limit* Stieltjes transforms for $K^{\mathrm{CK}}$ and $M$. For any $z \in \mathbb{C}^+$, by the relation between the eigenvalues of $K^{\mathrm{CK}}$ and $M$,

$$\frac{1}{n}\operatorname{Tr}\left(K^{\mathrm{CK}} - \frac{n}{d_1}z\operatorname{Id}\right)^{-1} = \frac{n - d_1}{n}\left(-\frac{n}{d_1}z\right)^{-1} + \frac{1}{n}\operatorname{Tr}\left(\frac{n}{d_1}M - \frac{n}{d_1}z\operatorname{Id}\right)^{-1}$$

$$= -\left(1 - \frac{d_1}{n}\right)\frac{d_1}{n}\cdot\frac{1}{z} + \left(\frac{d_1}{n}\right)^2\cdot\frac{1}{d_1}\operatorname{Tr}(M - z\operatorname{Id})^{-1}.$$

Taking the limit on both sides, we obtain the relation between $m_K(z)$ and $m_M(z)$, which is

$$m_K(\gamma_1 z) = -\left(1 - \frac{1}{\gamma_1}\right)\frac{1}{\gamma_1 z} + \frac{1}{\gamma_1^2}m_M(z) = \frac{1}{\gamma_1^2}\left(m_M(z) + \frac{1 - \gamma_1}{z}\right). \tag{67}$$

Following the notation of [46], let us set

$$\phi = 1/\gamma_0, \quad \psi = \gamma_1/\gamma_0, \quad \eta = 1 = \mathbb{E}[\sigma(\xi)^2], \quad \zeta = b_\sigma^2. \tag{68}$$

[46, Theorem 1] characterizes $G(z) \equiv -m_M(z)$ as the root of a quartic equation. Defining three $z$-dependent quantities $P, P_\phi, P_\psi$ by

$$G(z) = \frac{\psi}{z}P + \frac{1 - \psi}{z}, \quad P_\phi = 1 + (P - 1)\phi, \quad P_\psi = 1 + (P - 1)\psi, \tag{69}$$

this quartic equation is expressed as

$$P = 1 + (1 - \zeta)tP_\phi P_\psi + \frac{\zeta t P_\phi P_\psi}{1 - \zeta t P_\phi P_\psi} \qquad \text{where} \qquad t = \frac{1}{z\psi}, \tag{70}$$

see [46, Equations (10–12)].

To verify that (66) is equivalent to this equation (70), note that (66) means the Stieltjes transform $m_K(z)$ is defined by the Marcenko-Pastur equation (6) as

$$m_K(z) = \int \frac{1}{[(1 - b_\sigma^2) + b_\sigma^2 x][1 - \gamma_1 - \gamma_1 z m_K(z)] - z}d\mu_0(x). \tag{71}$$

Applying the identity $1 - \gamma_1 - \gamma_1^2 z m_K(\gamma_1 z) = -z m_M(z)$ from rearranging (67), and applying also $\zeta = b_\sigma^2$ in (68),

$$m_K(\gamma_1 z) = \int \frac{1}{[(1 - \zeta) + \zeta x][-z m_M(z)] - \gamma_1 z}d\mu_0(x). \tag{72}$$

When $X$ has i.i.d. $\mathcal{N}(0, 1/d_0)$ entries, the limit spectral distribution of $X^\top X$ is the Marcenko-Pastur law $\mu_0 = \rho_{\gamma_0}^{\text{MP}}$. The Stieltjes transform $m(z)$ of this law $\mu_0 = \rho_{\gamma_0}^{\text{MP}}$ is characterized by the quadratic equation

$$1 = m(z)[1 - \gamma_0 - \gamma_0 z m(z) - z]$$

(which is the specialization of (6) when $\mu$ is the point distribution at 1). Defining

$$g(a,b) = \int \frac{1}{ax - b} d\mu_0(x) = \frac{1}{a} m\left(\frac{b}{a}\right),$$

we obtain then that $g(a,b)$ satisfies the quadratic equation

$$1 = g(a,b)[a - \gamma_0 a - \gamma_0 b m(b/a) - b]$$
$$= g(a,b)[(a - b) - \gamma_0 a - \gamma_0 ab \cdot g(a,b)].$$

Applying this with $a = -\zeta z m_M(z)$ and $b = (1 - \zeta)z m_M(z) + \gamma_1 z$, the quantity (72) is exactly $g(a,b)$. Thus this equation holds for $g(a,b) = m_K(\gamma_1 z)$ and these settings of $(a,b)$, i.e.

$$1 = m_K(\gamma_1 z)\Big(-zm_M(z) - \gamma_1 z + \gamma_0 \zeta z m_M(z) + \gamma_0 \zeta z m_M(z)[(1-\zeta)z m_M(z) + \gamma_1 z] m_K(\gamma_1 z)\Big).$$
(73)

From the relation (67), we see that this is a quartic equation in $m_M(z)$. Note that the definitions of $P_\psi$ and $P_\phi$ in (69) may be equivalently written as

$$P_\psi = \psi P + 1 - \psi = zG(z) = -zm_M(z),$$

$$P_\phi = 1 + \frac{\phi}{\psi}(zG(z) - 1) = \frac{1}{\gamma_1}(-zm_M(z) - 1 + \gamma_1) = -\gamma_1 z m_K(\gamma_1 z)$$

where we have used $G(z) = -m_M(z)$, $\psi/\phi = \gamma_1$ from (68), and the relation (67). Applying now $\gamma_1 z = (\psi/\phi)z = 1/(\phi t)$ and $\gamma_0 = 1/\phi$, the equation (73) becomes

$$1 = -\phi t P_\phi \left(P_\psi - \frac{1}{\phi t} - \frac{\zeta}{\phi}P_\psi + \frac{\zeta}{\phi}P_\psi\left[-(1-\zeta)P_\psi + \frac{1}{\phi t}\right]\phi t P_\phi\right)$$
$$= -\phi t P_\phi P_\psi + P_\phi + (1 - P_\phi)\zeta t P_\phi P_\psi + \zeta(1-\zeta)\phi(t P_\phi P_\psi)^2.$$

This may be rearranged as

$$(1 - P_\phi - \phi)(1 - \zeta t P_\phi P_\psi) = -\phi(1 - \zeta t P_\phi P_\psi) - \phi t P_\phi P_\psi + \zeta(1-\zeta)\phi(t P_\phi P_\psi)^2,$$

and dividing both sides by $-\phi(1 - \zeta t P_\phi P_\psi)$ yields

$$\frac{1}{\phi}(P_\phi - 1) + 1 = 1 + \frac{t P_\phi P_\psi - \zeta(1-\zeta)(t P_\phi P_\psi)^2}{1 - \zeta t P_\phi P_\psi} = 1 + (1-\zeta)t P_\phi P_\psi + \frac{\zeta t P_\phi P_\psi}{1 - \zeta t P_\phi P_\psi}.$$

Identifying the left side as $P$ by (69), we obtain (70) as desired.

## J  Additional simulation results

### J.1  Pairwise orthogonality of training samples

All pairwise inner-products $\{\mathbf{x}_\alpha^\top \mathbf{x}_\beta : 1 \leq \alpha < \beta \leq n\}$, for (a) 5000 CIFAR-10 training samples, (b) 5000 CIFAR-10 training samples with the first 10 PCs removed, and (c) i.i.d. Gaussian training data of the same dimensions. Results for (b) were reported in Section 4.2, and results for (a) are reported

below in Appendix J.2. CIFAR-10 training samples were mean-centered and normalized to satisfy $\mathbf{x}_\alpha^\top 1 = 0$ and $\|\mathbf{x}_\alpha\|^2 = 1$ in (a) and (b).

The pairwise inner-products in (a) span a typical range of $[-0.5, 0.5]$. Those in (b) span a range of about $[-0.2, 0.2]$, and those in (c) about $[-0.02, 0.02]$. Thus, with 10 PCs removed, these inner-products for CIFAR-10 are larger than for i.i.d. Gaussian inputs by a factor of 10. We found in Section 4.2 that the inner-products of (b) are sufficiently small for the observed spectra to match the theoretical limits of Theorems 3.4 and 3.7.

## J.2 CK and NTK spectra for CIFAR-10 without removal of leading PCs

Same plots as Figure 2 for CIFAR-10 training samples, without the removal of the 10 leading PCs. We observe a close agreement of the observed CK spectrum with the limit spectrum of Theorem 3.4. However, there is a greater discrepancy of the NTK spectrum with the limit spectrum of Theorem 3.7 in this setting.

## J.3 Example images of CIFAR-10 with/without leading PCs

Example CIFAR-10 training samples for each class. For each training sample, we compare the original image (above) and the corresponding normalized image upon removing the top 10 PCs (below). Most of the image details are preserved upon removing these 10 PCs.

## J.4 Observed and limit CK spectra for all layers

Simulated spectra of the initial CK matrices $X_\ell^\top X_\ell$ at all intermediate layers $\ell = 1, \ldots, 5$, corresponding to the i.i.d. Gaussian training data example of Figure 1. Numerical computations of the limit spectra from Theorem 3.4 are overlaid in red. We observe a merging of the two bulk spectral components and an extension of the spectral support with increase in layer number.

The same as above, corresponding to the CIFAR-10 training samples in Appendix J.2. (Results with 10 PCs removed look the same.) A close agreement with the limit spectrum described by Theorem 3.4 is observed at each layer.

Spectra of the CK matrices at all three layers, corresponding to the trained 3-layer network of Section 4.3. The limit spectra at random initialization of weights are depicted in red, and the two largest eigenvalues of each matrix are depicted by blue arrows.

### J.5 CK spectrum after training on a CIFAR-10 example

We train a binary classifier on $n = 10000$ training samples from CIFAR-10, corresponding to classes 0 (airplane) and 1 (automobile). The classifier is a fully-connected network with $L = 4$ hidden layers of dimensions $d_1 = \ldots = d_4 = 1000$, with bias terms and a normalized sigmoid activation at each hidden layer and also at the output layer. This network is given by

$$f_\theta(\mathbf{x}) = \sigma(\mathbf{w}^\top \mathbf{x}^L + b), \qquad \mathbf{x}^\ell = \frac{1}{\sqrt{d_\ell}} \sigma(W_\ell \mathbf{x}^{\ell-1} + \mathbf{b}_\ell) \quad \text{for} \quad \ell = 1, \ldots, L$$

where $b \in \mathbb{R}$ and $\mathbf{b}_\ell \in \mathbb{R}^{d_\ell}$ for each $\ell = 1, \ldots, L$ are the bias parameters. The activation function $\sigma(x) \propto (1 - e^{-x})/(1 + e^{-x})$ is scaled such that $\mathbb{E}[\sigma(\xi)^2] = 1$. Weights $\theta = (\text{vec}(W_1), \ldots, \text{vec}(W_4), \mathbf{w})$ are initialized to independent $\mathcal{N}(0,1)$ for each entry, and biases $(\mathbf{b}_1, \ldots, \mathbf{b}_4, b)$ are initialized to 0. Hence, $K^{\text{CK}}$ at random initialization has the same definition as in the main text.

We train the weights and biases using the Adam optimizer in Keras, with learning rate 0.01, batch size 128, and 60 training epochs. To ensure that the leading PCs of the *untrained* kernel matrix $K^{\text{CK}}$ are not too predictive of the training labels, and to better separate the original PCs from those that emerge after training, we remove the leading 5 PCs of the input data before training. The resulting 0–1 classification accuracy on the CIFAR-10 test set is 85.3%. (Training without removing these 5 PCs yields a slightly higher test accuracy of 90.7%, using the same network architecture.)

Panel (a) above shows the eigenvalue distribution of $K^{\text{CK}}$ at random initialization, with the largest eigenvalue being approximately 500. We observe a close agreement with the limit spectrum of Theorem 3.4. Panel (b) shows the eigenvalues of $K^{\text{CK}}$ after training. We observe an elongation of the bulk spectral support and the emergence of large outlier eigenvalues, analogous to the synthetic example of Section 4.3.

The above figure depicts the information about the training labels that is contained in the top 2 PCs of $K^{\text{CK}}$, (a) before training and (b) after training. Denoting by $\hat{X}_L$ the rank-2 approximation of $X_L$, with columns $\hat{\mathbf{x}}_1^L, \ldots, \hat{\mathbf{x}}_n^L$ (both before and after training), we re-fit a linear binary classifier $y_\alpha = \sigma(\mathbf{w}^\top \hat{\mathbf{x}}_\alpha^L + b)$ of the training labels to these columns. The in-sample 0–1 training accuracy of this classifier is 51.4% pre-training and 96.8% post-training, and the figure shows the linear predictions $\mathbf{w}^\top \hat{\mathbf{x}}_\alpha^L + b$ against the training labels $y_\alpha$. We observe that the leading principal components of $K^{\text{CK}}$ are not predictive of the training labels before training, but become highly predictive after training.

## Footnotes

[4]In [46], the $1/\sqrt{d_0}$ scaling is in $W_1$ rather than $X$, but these are clearly the same. We consider $\sigma_w = \sigma_x = 1$ and $\eta = 1$ in the results of [46].