[Reviews · NeurIPS 2020]

Review 1

Summary and Contributions: Update: I agree with the authors that applying their results to RF models in the high-dimensional asymptotics is an important application and encourage them to add some discussion on this to a future version of the paper. __________________________________________________________________________ The authors study the spectrum of the Conjugate Kernel and the Neural Tangent Kernel for a fully-connect neural network in the high-dimensional limit. Under an orthogonality condition on the data, they characterize the spectra using a recursive fixed-point equation. Finally, on synthetic data and CIFAR-10, they show a good agreement with their theoretical predictions.

Strengths: The high-dimensional limit has be fruitful for replicating phenomena associated with overparameterized models. However, these analyses have been limited to more unstructured random feature models, so extending these results to structured models like the NTK that more closely match NNs is significant. A first step in this for deep NNs, is understanding the spectra of their NTK, which this paper achieves. A priori this problem is technically challenging.

Weaknesses: The paper’s main weakness is that while studying the spectra of the NTK for deep NNs is an important step, it does not itself have immediate ML implications. This might limit the paper’s appeal to the NeurIPS community.

Correctness: Based on my reading, the results and teir proofs are correct.

Clarity: The paper is clear and not difficult to follow. The level of mathematical detail in the main text is appropriate and does not obscure the paper’s story.

Relation to Prior Work: The discussion of prior work is mostly very good, but the paper of d’Ascoli et al., “Double Trouble in Double Descent: Bias and Variance(s) in the Lazy Regime” should probably be cited. Also, Adlam and Pennington, “The Neural Tangent Kernel in High Dimensions: Triple Descent and a Multi-Scale Theory of Generalization” also seems to study this linear-width asymptotic regime for a single-hidden layer NN, getting results on its spectrum and generalization properties.

Reproducibility: Yes

Additional Feedback: When comparing the theoretical predictions with the empirical spectra in Figs. 1, 2, and 3, it would be good to have a quantitative measure of their agreement.


Review 2

Summary and Contributions: The article extends recent results on the NTK approximation of DNN training. For once, a realistic assumption of simultaneously growing network dimensions is made, instead of the simplistic assumption where only the number of neurons in each layer tends to infinity. The authors derive under these conditions the asymptotic measure of the eigenvalue spectrum of the NTK (and the CK) under near-orthogonal conditions for the input data. The contribution is interesting as it unveils (as confirmed by simulations) the actual "natural size" behavior of the NTK, at least under noisy/orthogonal inputs. The results, which are non trivial, are also neatly expressed in terms of free multiplicative convolutions of the Marcenko-Pastur measure. Simulations on synthetic data confirm the results, while other simulations on real data reveals extra phenomena, not yet accounted for by the theory but which open an interesting direction of research!

Strengths: The results are new, interesting, and for the first time use a natural dimension growth for the NTK size. This comes along with a non-trivial conclusion (as is so far the case for infinite-dimensional NTKs) but this remains tractable by random matrix theory. To me, the work opens the path to a much deeper exploration of the real learning capabilities of DNNs. Very few works tackle network learning in such a realistic setting; it just remains to include more structure in the input data and see how this is treated during learning... This is why, despite the several weaknesses I could spot (see below), I am strongly in favor of seeing the article accepted.

Weaknesses: The results in themselves are not easy to read, at least to non-experts in RMT. Some of the more marginal results (such as the last theorem) could have been discarded to leave much more room to interpretation. See my detailed comments and questions on these aspects. Something that remains unclear to me is that the authors seem to assume that K^{NTK}(t) remains "approximately" constant over training in order to derive the dynamics for theta(t) which is NOT constant. Plugged back into K^{NTK}(t), this gives a strong time-dependence to K. When the network size is large, convergence to expectation is claimed, but in a random matrix setting, this is far from valid. The authors do not at all discuss this apparent contradiction. I would be ready to buy it if some theoretical control on how much "time independent" the random matrix K is would be provided.

Correctness: As a specialist in RMT, I am fairly confident that the results, which cleanly use the random matrix framework, are correct.

Clarity: Aside from the aforementioned time-dependence problem at the onset of the work, the article is quite clear and, despite its technicalities, quite accessible. It would nonetheless gain in leaving more room to discussion and interpretation.

Relation to Prior Work: Appropriate. See also references [*1], [*2] below in "Additional feedback" which contradict/complement some of the statements in the text.

Reproducibility: Yes

Additional Feedback: Detailed comments: 1. - "Training occurs most rapidly along the eigenvectors of the largest eigenvalues" --> In [*1] below, it is shown that training arises both from isolated largest eigenvalues as well as from the "collective effect" of the bulk spectrum (even when no isolated eigenvalue is found, training occurs: there is no phase transition). - J is defined a bit late, as JJ' is introduced without knowing who's J - it is not clear from the introduction whether the work provides an understanding of the evolution of the NTK (the aforementioned "elongation" of the spectrum) during training. This deserves to be clarified (for instance, by clearly stating which are the actual contributions of the work). 2. - the scaling 1/sqrt(d_l) is not really NTK-related, but rather related to some bounded energy control when growing the network size? - 1/pi.Im[m_mu(z)] is an approximation of the density of mu only for x a continuity point of mu. - for (5), you must specify that Phi is deterministic (independent of X not being sufficient). 3. - among the examples satisfying Ass 3.2-(c), it is also interesting to see that it should also hold true for a "non-zero mean" and "non-identity covariance" Gaussian mixture (or mixture of concentrated random vectors) satisfying the "non-trivial" growth regime of [33]. So the setting becomes valid for more practical data than iid inputs!, especially for non-trivial classification tasks. - could the authors comment on the b_sigma=0 case? This essentially holds when sigma is an odd function (x, x^3, tanh(x), ...). Can one interpret the fact that these functions "do not propagate" the previous laws? It is known that these are nasty choices to exploit the non-linear discriminating features of the data (it focuses on their means, and kills their covariance aspects...), how do all these remarks relate? Again a result related to [33,*2]. - for the sake of interpretation/discussion, I would have discarded Section 3.4 which does not bring much to the work, and use this space for further discussion. 4. - in Fig 1-c), do we observe a bi-modal distribution for the NTK or an isolated mass + a bulk? Bi-modal bulks are interesting as, in a "spike extension", isolated eigenvalues may sneak in between, associated to relevant eigenvectors. - in 4.2, it is indeed quite unfair to remove the 10 leading PCs... This denatures the input data to essentially making it look noise-like. Possibly more acceptable solutions would be: (i) a collective centering and scaling of the data (possibly already done?), (ii) a per-class centering and scaling of the data (not a smart move in practice, but is could discard some of the isolated components more naturally, (iii) use only one (centred/scaled) class for training? - using the Adam optimizer is quite unfair too. The NTK is not derived under this assumption. Can the authors preferrably show the eigenvalue distribution output for an actual plain full-batch gradient descent mechanism as used to obtain the NTK? My own experiments seem to reveal a differing behavior. Refs: [*1] Z. Liao, R. Couillet, "The Dynamics of Learning: A Random Matrix Approach", International Conference on Machine Learning (ICML'18), Stockholm, Sweden, 2018. [*2] Z. Liao, R. Couillet, "On Inner-product Kernels of High Dimensional Data", IEEE International Workshop on Computational Advances in Multi-Sensor Adaptive Processing (CAMSAP'19), Guadeloupe, France, 2019. ****** After rebuttal, I confirm my overall grade. This is a very interesting article.


Review 3

Summary and Contributions: Under the assumptions that inputs are weakly correlated, the authors derived the limiting spectrum of the conjugate kernel (CK) and neural tangent kernel (NTK) for multi-layers feedforward networks. The results are novel and non-trivial.

Strengths: A rigorous deviation of the spectrum of multi-layers NTK and CK assuming the width and number of inputs goes to infinity with the same rate. The result is significant and non-trivial, given existing results are mainly focused on one hidden layer setting. The methods here could be interesting and useful to many researchers.

Weaknesses: As a conference paper, i don't see obvious weaknesses here.

Correctness: I haven't check the mathematical details in the appendix, but the approach seems correct. I am confident Theorem 3.4 and Lemma 3.5 are correct. I am not familiar with the details to derive Thm 3.7 from Lemma 3.5. But the approach is mathematically sound.

Clarity: I think so.

Relation to Prior Work: I think so.

Reproducibility: Yes

Additional Feedback: Is the approach to derive Thm 3.7 from Lemma 3.5 standard in the RMT literature? If so, could you provide a pointer?


Review 4

Summary and Contributions: ***post-rebuttal*** Thank you for your detailed answers, and especially the clarification regarding orthogonality and d_0 \to \infty. I'm happy with the proposed changes and am upgrading my score by one. *** This paper studies the spectrum of the Neural Tangent Kernel (NTK) and Conjugate Kernel (CK; also known as the NNGP kernel) of deep fully connected networks with twice differentiable activations as the input dimension, number of input points, and the width of all layers go to infinity. Assuming that the ratio of the number of points and each dimension converges to a finite constant, and that the input vectors are nearly orthogonal, the authors prove the distribution of the CK's spectrum converges weakly to a limit described by an iterative application of the Marchenko-Pastur map, and the limit of the NTK spectrum converges weakly to a linear combination of the weak limits of the spectra of the CK kernels of each layer. The paper is concluded by a series of numerical experiments validating the results when the authors' assumptions hold (in which case a good agreement is found), and studying what happens when they are violated (in which case especially the NTK spectrum distribution exhibits a behaviour distinct from the otherwise valid limit).

Strengths: - A solid technical contribution providing novel insights into the behaviour of neural networks in the "lazy" regime. - Sensible empirical evaluation showcasing both validity of the theoretical predictions when the assumptions do hold, and what happens when they do not (I especially appreciated the later as this is not very common, but provides valuable insights beyond what was theoretically proven). - An efficient algorithm for computation of both the CK and NTK spectral limit densities.

Weaknesses: - The paper would be more significant if the authors could remove the assumption that the inputs are nearly orthogonal which is often not the case (as the authors acknowledge in their CIFAR-10 experiments). While the regime the authors study is not the standard NTK/NNGP (i.e., a fixed number of inputs and only the widths of individual layers going to infinity), I still wonder whether this assumption is not too similar to just feeding in white noise (which is essentially what would happen in the standard NNGP/NTK setting). Such a setup is far from the practical scenarios NNs are typically deployed in. (Nevertheless, I do consider the submission to be substantial enough to warrant publication even with this limitation.) - There are other assumptions like the twice differentiable activations (excludes ReLU and similar) which limit the applicability of the provided theory. (Again, I do not consider these to be serious enough to outweigh the positive sides of the paper.)

Correctness: I did not check the proofs beyond high-level skimming.

Clarity: Yes, this is the best written submission of those allotted to me.

Relation to Prior Work: Yes.

Reproducibility: Yes

Additional Feedback: - Can you clarify whether the convergence in Theorem 3.4 holds marginally for each l, or jointly for all l please? - Can you please clarify the assumption that the dimension of the input spaces grows to infinity? Is it possible to still have a fixed size input dimension and then just project into an increasingly large dimension d_0, say by an appropriately scaled i.i.d. Gaussian matrix? - For the CIFAR-10 experiments, could you please include examples of images with and without the top components? It might help the reader to visually assess the difference between your nearly-orthogonal regime and the one that typically occurs in image data. - Besides the Lee et al. (2018) paper ([29] in your notation), you should cite the Matthews et al. (2018) "Gaussian process behaviour in wide deep neural networks" paper. Both came out and were published at the same time (ICLR 2018), and the latter has the benefit of proving (weak) convergence for neural networks with all layers finite but growing wide, whereas the former only proves convergence for a single hidden layer neural network with inputs sampled from a Gaussian process.

[Author Response · NeurIPS 2020]

Thanks for your careful reading and positive feedback! We address major comments here (and the rest in the revision).

**Further discussion/interpretation, and implications for ML (R1 and R2)** The $9^{\text{th}}$ content page allows us to greatly
expand our discussion throughout the paper, and to add a conclusion section highlighting the following implications:

(a) An increasingly large body of literature studies generalization in random features regression models derived from the
CK or NTK, and associated multiple-descent phenomena. In the linear-width regime, these results rely on asymptotic
approximations for the Stieltjes transforms and resolvents of these kernels. Such studies have largely been limited to
single-layer networks, and our results and techniques may enable their extension to deep networks with many layers.

(b) The linear-width asymptotic regime may provide a theoretically tractable setting for studying feature learning
and "non-lazy" network training, and it is arguably closer to the operating regimes of neural networks in practical
applications. Our experiments suggest an interesting possible mechanism of training in this regime, and our theoretical
analysis of the spectra for random weights may provide a first step towards understanding this phenomenon.

**Assumption of pairwise orthogonality (R2 and R4)** Thanks very much for these comments. As R2 points out, we
believe this assumption is significantly more general than white noise. In the revision, we will add a discussion that the
assumption encompasses many settings of independent samples with input dimension $d_0 \asymp n$, including:

(a) *Non-white* Gaussian inputs $\mathbf{x}_\alpha \sim \mathcal{N}(0, \Sigma)$, for any $\Sigma$ satisfying $\operatorname{Tr} \Sigma = 1$ and $\|\Sigma\| \le C/d_0$. Note that such data
can have spectral distribution very different from $\mathbf{x}_\alpha$ with i.i.d. entries (which would be the Marcenko-Pastur law).

(b) More generally, inputs that may be expressed as $\mathbf{x}_\alpha = f(\mathbf{z}_\alpha)/\sqrt{d_0}$, where $\mathbf{z}_\alpha \in \mathbb{R}^m$ has independent entries
satisfying a log-Sobolev inequality, and $f : \mathbb{R}^m \to \mathbb{R}^{d_0}$ is any Lipschitz function.

(c) Inputs $\mathbf{x}_\alpha$ drawn from certain multi-class Gaussian mixture models, satisfying the high-dimensional asymptotic
assumptions of [CBG '16], [LLC '18], [LC '18]. The mixture components can differ in both mean and covariance.

**Derivation of Theorem 3.7 from Lemma 3.5 (R3)** We believe this derivation is the main point of theoretical novelty
in our work, and is not standard. Each $X_\ell$ in $z_{-1}\operatorname{Id} + z_0 X_0^\top X_0 + \ldots + z_L X_L^\top X_L$ has a complicated dependence on
$X_0, \ldots, X_{\ell-1}$, so this is not a classical RMT model. We develop the new idea of analyzing the extended matrix model
$(z_{-1}\operatorname{Id} + z_0 X_0^\top X_0 + \ldots + z_L X_L^\top X_L)^{-1}(w_{-1}\operatorname{Id} + w_0 X_0^\top X_0 + \ldots + w_L X_L^\top X_L)$ in order to recursively characterize
the spectrum by induction on depth. The resulting fixed-point equations are also non-standard, and led to new challenges
in inductively showing uniqueness of their fixed points and providing a numerical algorithm for solving these equations.

**Removal of 10 leading PCs (R2 and R4)** This figure shows the NTK spectrum after mean-centering each CIFAR-10
class, rather than removing 10 PCs. The fit is OK but not perfect. Also shown are example images before (left) / after
(right) removing the 10 PCs. Differences are hard to discern, and we will add a page of such images to the appendix.

**Outliers and Adam optimizer (R2)** We agree that the role of Adam is unclear, and we will make our code publicly
available for further exploration. Training using full-batch gradient descent is slow—we tried based on R2's feedback,
but had difficulty producing results with comparable generalization in a short time. In our Adam experiments, we tested
various network depths, widths, and learning rates: Outlier eigenvalues emerged only in experiments that yielded good
generalization, and not in those where the learned function generalized poorly. Also, these phenomena for the CK are
perhaps more fundamental, and this then does not relate to the specific gradient flow derivation of the NTK.

**NTK remains constant over training (R2)** Our apologies for this confusion, and we will clarify in the revision: We
do not claim the NTK remains approximately constant in this regime. The training dynamics described in Section 2.1
hold regardless of whether $K^{\text{NTK}}(t)$ evolves or is fixed, and the eigenvalue $\lambda_\alpha(t)$ always determines the *instantaneous*
decay of the training error along $\mathbf{v}_\alpha(t)$ at the instant $t$. "Training occurs most rapidly along the eigenvectors of the
largest eigenvalues" is just an informal statement of this, with the understanding that the eigenvectors also evolve over
training. We will also clarify in the intro that our theory pertains only to random weights and not to this evolution.

**Miscellaneous** Scaling by $1/\sqrt{d_\ell}$ is specifically important for the NTK as it affects the scaling of the derivative in
deriving the NTK (R2). $b_\sigma = 0$ indeed has implications for classification and training, and we will add discussion
and references to [CBG '16], [PW '17] (R2). The NTK spectrum here has two non-point-mass bulk components (R2).
Convergence in Thm 3.4 holds marginally for each $\ell$ (R4). $d_0 \to \infty$ is necessary to ensure the approximate pairwise
orthogonality, but is not otherwise used in the proof (R4). Thanks very much for the missing references! (R1, R2, R4)

[Meta-Review · NeurIPS 2020]

The reviewers and I are all confident that this paper will be interesting to the NeurIPS community and should be accepted. In addition to the improvements suggested by the reviewers, I would encourage the authors to expand the description of how to unfold the recursion in Theorem 3.7. The discussion in Appendix A helps, but it is insufficient as it is missing crucial details that would clarify how to interpret some of the ambiguous notation. I think including a detailed worked example would be an important addition.